# The Bortoluzzi Mud Volcano (Ionian Sea, Italy) and its potential for tracking the seismic cycle of active faults

Marco Cuffaro[1], Andrea Billi[1], Sabina Bigi[2], Alessandro Bosman[1], Cinzia G. Caruso[3], Alessia Conti[2], Andrea Corbo[3], Antonio Costanza[4], Giuseppe D'Anna[4], Carlo Doglioni[2,5], Paolo Esestime[6], Gioacchino Fertitta[4], Luca Gasperini[6], Francesco Italiano[3], Gianluca Lazzaro[3], Marco Ligi[6], Manfredi Longo[3], Eleonora Martorelli[1], Lorenzo Petracchini[1], Patrizio Petricca[2], Alina Polonia[6], and Tiziana Sgroi[5]

[1]Istituto di Geologia Ambientale e Geoingegneria, CNR, Rome, Italy
[2]Dipartimento di Scienze della Terra, Sapienza Universitá di Roma, Rome, Italy
[3]Istituto Nazionale di Geofisica e Vulcanologia, Palermo, Italy
[4]Istituto Nazionale di Geofisica e Vulcanologia, Gibilmanna, Italy
[5]Istituto Nazionale di Geofisica e Vulcanologia, Rome, Italy
[6]Spectrum Geo Ltd, Woking, United Kingdom
[7]Consiglio Nazionale delle Ricerche, ISMAR, Bologna, Italy

**Correspondence:** Andrea Billi (andrea.billi@cnr.it)

**Abstract.** The Ionian Sea in southern Italy is at the centre of active interaction and convergence between the Eurasian and African-Adriatic plates in the Mediterranean. This area is seismically active with instrumentally/historically-recorded Mw > 7.0 earthquakes and it is affected by recently-discovered long strike-slip faults across the active Calabrian accretionary wedge. Many mud volcanoes occur on top of the wedge. A recently-discovered one (here named Bortoluzzi Mud Volcano, BMV) was surveyed during the Seismofaults 2017 cruise (May 2017). Bathymetric-backscatter surveys, seismic reflection profiles, geochemical and earthquake data as well as a gravity core are here used to geologically, geochemically, and geophysically characterize this structure. The BMV is a circular feature $\simeq$22 m high and $\simeq$1100 m in diameter with steep slopes (up to a dip of 22°). It sits atop the Calabrian accretionary wedge and a system of flower-like oblique-slip faults that are probably seismically active as demonstrated by earthquake hypocentral and focal data. Geochemistry of water samples from the seawater column on top of the BMV shows a significant contamination of the bottom waters from saline (evaporite-type) $CH_4$-dominated crustal-derived fluids similar to the fluids collected from a mud volcano located in the Calabria main land over the same accretionary wedge. These results attest for the occurrence of open crustal pathways for fluids through the BMV down to at least the Messinian evaporites at about -3000 m. This evidence is also substantiated by Helium isotope ratios and by comparison and contrast with different geochemical data from three sea water columns located over other active faults in the Ionian Sea area. One conclusion is that the BMV may be useful for tracking the seismic cycle of active faults through geochemical monitoring. Due to the widespread diffusion of mud volcanoes in seismically active settings, this study contributes to indicate a future path for the use of mud volcanoes in the monitoring and mitigation of natural hazards.

## 1 Introduction

Mud volcanoes are ubiquitous structures on the Earth's surface both in marine and in continental settings and form for a variety of causes, most of which amenable to clay- and fluid-rich subsurface levels, where overpressures contribute to drive the ascent of mixed and liquefied clay, water, and gases (e.g., $CO_2$, $CH_4$, and N), with consequent formation of the conical or conical-trunk landform (Etiope and Milkov, 2004; Planke et al., 2003; León et al., 2007; Ceramicola et al., 2014; Sella et al., 2014). Mud volcanoes can be therefore taken as markers of overpressure-related geofluid circulation, efflux, and ascent (Bertoni and Cartwright, 2015; Kirkham et al., 2017).

Many mud volcanoes occur on active accretionary/sedimentary prisms (Cita, 1981; Kopf, 2002; Panieri et al., 2013; Ceramicola et al., 2014), where their degassing activity is at least in part connected with active compression and associated faulting and fracturing (Camerlenghi et al., 1995; Kopf, 2002; Chamot-Rooke et al., 2005; Mazzini et al., 2007; Panieri et al., 2013). It is also noteworthy that, in these active tectonic settings, the longevity of mud volcanoes may even exceed 1 Ma (Cita et al., 1989; Camerlenghi et al., 1992, 1995; Robertson, 1996; Kopf et al., 1998; Praeg et al., 2009; Somoza et al., 2012). All together the known submarine mud volcanoes release about 27 Mt methane $a^{-1}$ (plus other gases; Milkov, 2004; Etiope and Milkov, 2004).

The Ionian Sea between eastern Sicily and southern Calabria (Italy; Fig. 1) hosts an active accretionary prism (the Calabrian Arc; Minelli and Faccenna, 2010; Polonia et al., 2011, 2016; Gutscher et al., 2016), where both prism-parallel thrusts and across-prism strike- and oblique-slip faults are active. Some of these faults are the source structures of some among the most devastating earthquakes and tsunamis of southern Italy and the entire Mediterranean (Polonia et al., 2012; Totaro et al., 2013; Doglioni et al., 2012). Among other devastating events in the Ionian Sea large area, we recall the 1908 (Mw 7.2), 1783 (Mw 6.9), 1693 (Mw 7.4), 1169 (Mw 6.6), and 362 (Mw 6.6) AD earthquakes and tsunamis, which caused damage, devastation, and death (more than 80,000 deaths in 1908) in eastern Sicily and southern Calabria (Locati et al., 2016). Although such events have been studied by many authors for the great impact they produced, their origin (zone and generation mechanism) is still heavily debated both for earthquakes and tsunamis. The faults that generated such earthquakes are not yet known as it is unknown whether the associated tsunamis were generated directly by earthquakes (i.e., fault dislocation of seabed) or indirectly by seismically-triggered submarine slides (Valensise and Pantosti, 1992; Billi et al., 2008, 2010; Polonia et al., 2016).

To better define the seismic behavior of active faults in the Ionian Sea, during the Seismofaults 2017 and 2018 marine surveys (May 2017 and 2018, respectively; www.seismofaults.it), we deployed eleven seabottom seismometers (OBS/H) and two seabottom multiparameric geochemical-geophysical sensors. At the time of writing of this paper, some of these devices are still operating on the seabottom, while data from the recovered devices are being processed and are not therefore available for publication. One of the aims of this project (Seismofaults) and related surveys was to contribute to the advancement of the science of earthquake forecasting. For this reason, we deployed the multiparameric sensors (i.e., to detect anomalous degassing prior to earthquakes), collected sea water column samples along and far away from (active) faults, and searched for new fluid

venting structures such as mud volcanoes. The activity of some of these latter structures may be partly connected with (seismic) faulting (Martinelli et al., 1995; Polonia et al., 2011; Capozzi et al., 2012; Panieri et al., 2013; Ceramicola et al., 2014). For instance, Martinelli and Ferrari (1991) and Martinelli et al. (1995) recorded radon anomalies ($^{222}$Rn) in the liquid phase of some mud volcanoes from the northern Apennines (Italy) before and during low magnitude earthquakes ($\leq 2.5$) occurred nearby in

1986 and 1987. Again in the northern Apennines, Lupi et al. (2015) recorded potential precursory seismological signals ten minutes before an Mw 4.7 earthquake in a monitored mud volcanic field. Discovering and monitoring such structures may therefore be the key, in the future, to better understand the seismic cycles and possible accompanying or precursory processes and phenomena such as anomalous fluid discharge from mud volcanoes (Martinelli and Ferrari, 1991; Martinelli et al., 1995; Kopf, 2002; Mazzini and Etiope, 2017). A link between the frequency of eruption of some mud volcanoes and the seismic

activity has been often documented (Deville and Guerlais, 2009; Manga et al., 2009). A nice example is the 5 km long mud flow emitted by the Kandewari mud volcano (Pakistan) following the 2001 Gujarat-Bhuj Mw 7.7 earthquake (India-Pakistan Delisle et al., 2002) It is also true, however, that many active mud volcanoes do not seem to manifestly react to nearby earthquakes. Our aim in this article is not finding a link between earthquakes and the activity of mud volcanoes but to understand whether even a mud volcano that seems inactive or poorly-active can constitute an open and preferential pathway for possible geochemical

precursors of earthquakes. Whether these precursors truly manifest before an earthquake is a different task that has to be accomplished with a specific monitoring as we will explain in the discussion section. Our aim is nonetheless relevant for many geoscientists as noteworthy examples of earthquake geochemical precursors are documented in many previous works (Wakita et al., 1988; Igarashi et al., 1995; Claesson et al., 2004; Huang and Ding, 2012; Inan et al., 2012; Skelton et al., 2014, 2019; Sano et al., 2016; Barberio et al., 2017; Petitta et al., 2018; Boschetti et al., 2019) and are very promising for the science of

earthquake forecasting.

Following the above-mentioned reasons and aims, during the Seismofaults 2017 scientific cruise in the Ionian Sea (Fig. 1), we surveyed a mud volcano recently signalled on the Calabrian accretionary prism by Gutscher et al. (2017) and Loher et al. (2018) but never analyzed in detail (Supplement, Movies S1 and S2). We performed a multidisciplinary analysis on this feature, including high-resolution multibeam bathymetry, high frequency chirp-sonar profiling, physical and chemical analyses of the

25 water column above the mud volcano, and gravity coring on one side of this structure. Based on these new data integrated with previous ones, we interpret the recently-discovered mud volcano within the framework of the (seismically) active accretionary prism of Calabria and eventually provide some implications for future marine researches and monitoring of the seismic cycle in marine areas. As mentioned above, our main aim is not only to characterize the mud volcano but also to provide a contribution toward a potential and feasible future path for the use of these ubiquitous structures in favour of the monitoring and mitigation

of natural hazards. As the mud volcano studied in this paper had already been observed years ago by our colleague Giovanni Bortoluzzi in one of his numerous marine surveys, we name this structure Bortoluzzi Mud Volcano (BMV) to recall his memory and fruitful life spent sailing for the science over the Mediterranean Sea and the oceans. The BMV was selected for this study mainly for its proximity to the coast, particularly to seismically hazardous regions (southern Calabria and eastern Sicily; Fig. 1), and for its location on top of a seismically active fault systems, as we will show below.

## 2 Geological Setting

The BMV is located on top of the Calabrian Arc (Ionian Sea; Fig. 1) that has developed along the Africa-Eurasia plate boundary in the center of the Mediterranean Sea. The arc belongs to the eastward retreating Apennines subduction system connecting the NW-SE-trending Apennines with the E-W-oriented Maghrebian thrust-fold belt (Patacca and Scandone, 2004). In particular, the Calabrian Arc has developed on top of a NW-dipping subduction system with the Ionian lithosphere sinking toward NW beneath the Tyrrhenian lithosphere. This subduction system is also characterized by an active volcanic arc (the Aeolian Islands in the southeastern Tyrrhenian) and a well-defined Wadati–Benioff zone (Wortel and Spakman, 2000), with earthquakes descending to nearly 500 km depth beneath the Aeolian Islands on the Tyrrhenian lithosphere. The Africa-Eurasia convergence is active in this area at a very slow rate (5 mm/yr or even less), as documented by recent GPS studies (Serpelloni et al., 2007; Billi et al., 2011; Palano et al., 2012, 2015). The external part of the Arc is represented by a 300 km wide accretionary complex bounded to the south by the outer deformation front and laterally by two major structural discontinuities: the Malta escarpment to the southwest and the Apulia escarpment to the northeast. To the northwest (i.e., toward the Calabria region), the accretionary wedge is significantly thickened (Cernobori et al., 1996), whereas it tapers away toward the southeast in the Mediterranean Sea. The wedge is segmented along strike in different structural domains by NW-trending structural discontinuities. The compartments are characterized by different rheologies and deformation styles (Polonia et al., 2011). In particular, three main morpho-structural domains can be identified in the subduction complex: (i) the post-Messinian accretionary wedge; (ii) the pre-Messinian accretionary wedge, and (iii) the inner plateau. Structural styles and seafloor morphologies vary in these four compartments in correlation with different tectonic processes that include frontal accretion, out-of-sequence thrusting, underplating, and complex faulting.

In the Calabrian Arc, the very low tapered (taper angle about 1.5°) outermost accretionary wedge is a salt-bearing complex developed during and after the Messinian salinity crisis (Fig. 1). Frontal accretion of the arc is active in this southern region over a shallow basal detachment located within or at the base of the Messinian evaporites. The inner wedge, which is located toward NW at the rear of the post-Messinian accretionary complex, consists of pre-Messinian clastic sediments. The basal thrust of the inner wedge is located on top of the Cretaceous sediments and/or at the transition with the basement. Moreover, the inner wedge is bounded toward NW by an inner deformation front, representing the transition between the strongly deformed accretionary wedge to the SE and a less deformed inner plateau to the NW. This plateau is however dissected by long normal and strike-slip fault zones mostly striking NW-SE. The inner plateau is characterized by chaotic units (Rossi and Sartori, 1981) and many mud volcanoes (Praeg et al., 2009; Ceramicola et al., 2014; Gutscher et al., 2017; Loher et al., 2018) including the BMV that is studied in this work.

As mentioned above, the Calabrian accretionary wedge is cut across (NW-SE) by a set of long faults or deformation zones (Fig. 1). Most of these tectonic features are active and characterized by strike-slip to transtensional or transpressional (oblique) kinematics. At a broad scale, the sense of horizontal shear along these zone is usually right-lateral (Minelli and Faccenna, 2010; Polonia et al., 2011, 2016, 2017; Gallais et al., 2012, 2013; Gutscher et al., 2016; Bortoluzzi et al., 2017). Below, we show that the BMV is located on top of one of these oblique-slip features crossing the Calabrian accretionary wedge.

## 3 Methods and Data

### 3.1 Rationale

As mentioned above, the main target of this work is the BMV located at 37° 53' 01" N and 16° 16' 50" E in the Ionian Sea (Fig. 1). During the Seismofaults 2017 cruise, in the BMV area, we acquired high-resolution multibeam bathymetry (bathymetry, backscatter and water column), chirp seismic profiles, physical and geochemical data of the sea water column, and a sediment gravity core. These data are described below together with the related methods of acquisition and with a re-processed and unpublished seismic reflection profile whose track is only about 300 m far away from the BMV. We integrated the above-mentioned geological-geophysical-geochemical evidence with previously-published data and with data from public databases (e.g., earthquake data). All used data are reported in the figures and tables of this paper and related supplementary material. Moreover, with this paper, we release 4970 km$^2$ of newly-acquired (during the Seismofaults 2017 cruise) high-resolution bathymetric data in the Ionian Sea including the BMV area (Fig. S1).

### 3.2 Earthquakes

To understand whether the study area (BMV) is seismically active, we analysed the crustal seismicity (depth ≤ 40 km) occurred in the study area and recorded by Osservatorio Nazionale Terremoti of the Istituto Nazionale di Geofisica e Vulcanologia (http://cnt.rm.ingv.it/) from 1985 to 2017. We collected the arrival time data from the Italian Seismic Catalogue (ISC 1985-2002, available online at: http://csi.rm.ingv.it/; Castello et al., 2006) and the Italian Seismic Bulletin (ISB 2003-2012, available online at http://bollettinosismico.rm.ingv.it/). Then, one of us (T.S.) manually picked the most recent earthquakes (2013-2017) following the same procedure adopted by the analysts of ISC 1985-2002 and ISB 2003-2012. Combining the picked arrival time data with the arrival time data from the analyzed bulletins, one of us relocated all earthquakes (Table S1). Moreover, to understand the kinematics of the seismic faulting, we collected earthquake focal mechanisms (Table S2) from the European-Mediterranean Regional Centroid-Moment Tensors (RCMT) catalog (http://rcmt2.bo.ingv.it/) and from previous papers by Orecchio et al. (2014) and Polonia et al. (2016). Earthquake data (Tables S1 and S2) are shown in map view in Figs. 1 and 2(a) and in vertical cross-sectional view in Fig. 2(b).

### 3.3 Bathymetry and Geomorphology

We carried out the high resolution multibeam bathymetric survey (Figs. 3,4 and S1) using a multibeam Teledyne Reson SeaBat 7160 (41-47 kHz) echosounder characterized by footprint size of 1°× 1°. We identified precise positioning through differential GPS (accuracy ±0.5 m), while we derived sound velocity profiles from multiple Conductivity-Temperature-Depth (CTD) casts (Seabird 911plus) to ray trace the acoustic wave along the water column. We processed multibeam data on board using Caris Hips & Sips hydrographic software (Bosman et al., 2015). We used backscatter images and observations of raw data scattering along the water column (Fig. 5) to verify anomalies of amplitude on the seafloor and along the water column.

Eventually, we compared some geometrical features of the BMV with the same geometrical features from previously published databases of mud volcanoes (Kioka and Ashi, 2015; Kirkham et al., 2017, Fig. 6).

### 3.4 Single-channel Chirp Seismic Reflection Profiles

During the Seismofaults 2017 survey, to define the local geological setting and a high resolution seismic stratigraphy of the BMV and surrounding area, we acquired a set of chirp seismic profiles (Figs. 7b-d, 8, and S2) using a frequency-modulated source operating in the frequency range of 2-7 kHz (Benthos Chirp III) and recorded with a 0.5-0.8 s sweep length. Maximum sub-bottom penetration is up to about 40 ms (TWT; corresponding to c. 30 m if considering a seismic velocity of 1500 m/s) and vertical resolution is about 0.7 ms (TWT; corresponding to c. 0.5 m if considering a seismic velocity of 1500 m/s). We processed the chirp profiles using the GeoSuite All Works software, applying Time Varied Gain, and the open source software Seisprho (Gasperini and Stanghellini, 2009). Thicknesses and depths of seismic profiles are described in two-way travel time (TWT; Figs. 7b-d and 8), with a seismic velocity of 1500 m/s being used to convert two-way travel time into depth. Seismic data interpretation was carried out through the Kingdom Suite software also integrating the bathymetric data and a gravity core (Fig. 9).

### 3.5 Seabottom Gravity Core

We collected the SF17-01 core (Fig. 9; see core location in Fig. 7a) during the Seismofaults 2017 survey using a 1.2 ton gravity corer with coring pipes 6 m long. Location of the core was controlled through the aforementioned onboard differential GPS system (accuracy ±0.5 m). We analyzed the core sections through a multi-proxy approach involving high-resolution digital photographs and determination of physical properties and sand content as deduced through the weight of selected samples. We also acquired high resolution magnetic susceptibility (MS) through a core log system (Bartinghton model MS2, 100 mm loop sensor) with a sampling interval of 1.0 cm.

### 3.6 Multi-Channel Seismic Reflection Profiles

In this paper, we present two multiple channel seismic reflection profiles and related interpretations (Fig. 10; see tracks in Fig. 1). The first profile (Fig. 10a) is the CROP M-31 (Scrocca et al., 2003a) in the version interpreted by Polonia et al. (2016). The second profile is the CA99-215 (Fig. 10b) both in the version interpreted by Polonia et al. (2016) and in a new version (Figs. 10c and 10d; see also Fig. S2) deriving from a recent reprocessing kindly provided by Spectrum Geo Ltd (http://www.spectrumgeo.com/) within the framework of a confidentiality agreement between Spectrum Geo and Sapienza University of Rome.

The CROP M-31 was acquired in 1994 in the framework of the CROP (CROsta Profonda) Project (Scrocca et al., 2003b). The streamer used was analogic, 4500 m long, composed by 180 channels with a group interval of 25 m, towed at a depth of 12-14 m. The total source volume was equal to 4882 in$^3$. The recording time length was 20 s, shot interval was 50 m, and coverage was 45 (Bertelli et al., 2003). In general, the applied processing sequences consisted in quality control, gain recovery,

trace sum (optional), deconvolution, CDP re-ordering, velocity analysis (CVS), NMO correction, muting, multiple reflection attenuation, array simulation (optional), weighted stack, F-K filtering (optional), horizontal mixing (optional), time-variant filter, and equalization (Bertelli et al., 2003).

The CA99-215 profile was acquired in 1999 and reprocessed in 2001. Source volume was 3410 in$^3$, located at 6 m depth, with a shot interval of 25 m. The used streamer had a length of 6000 m, towed at 8 m depth, with a group interval of 12.5 m, and a recording length of 8 s. Reprocessing mainly consisted in a standard PSTM (pre-stack time migration) processing sequence, resulted in an improved data quality.

Stratigraphic syntheses from the Fosca 1 and Floriana 1 wells (original well data are from the Videpi public database available online at http://unmig.sviluppoeconomico.gov.it/videpi/videpi.asp) are also presented (Fig. 10f) to help interpreting the seismic profiles (see well location in Fig. 1).

### 3.7  Geochemical Features of Four Sea Water Columns

During the Seismofaults 2017 survey, we sampled sea water columns at various depths in four localities of the Ionian Sea (Fig. 1 and Tables 1 and 2): namely, above the BMV and above the GeoC1, GeoC2, and GeoC3 localities that are along or nearby major presumably-active fault zones (Polonia et al., 2012, 2016). We carried out vertical casts by Rosette and Niskin bottles to determine the geochemical features and dissolved gases at the sea bottom and along the water columns (Fig. 11 and Tables 1 and 2). Samples are compared with the local air-saturated sea water (ASSW) used as benchmark.

The samples, specifically collected for the extraction of the whole gas phase for chemical and isotopic analyses, were stored in 240 ml pyrex bottles sealed on board using rubber/teflon septa and purpose-built pliers, and analyzed within two weeks. Details of the sampling methodology are reported in Italiano et al. (2009, 2014). During the cruise, the sampled bottles were stored upside down, keeping the necks with the rubber septa submerged in sea-water until they were transferred to the laboratory. The collected sea-water samples underwent laboratory procedures for both chemical (concentration of dissolved gas species) and isotopic (helium isotopes) determinations. In the laboratory, the dissolved gases were extracted after equilibrium was reached at constant temperature with a host-gas (high-purity argon) injected in the sample bottle (see Italiano et al., 2009, 2014, for further details). The chemical analyses of $O_2$, $N_2$, $CH_4$, and $CO_2$ were carried out by gas-chromatography (Agylent 7800B equipped with a double TCD-FID detector) using argon as carrier gas. Typical analytical uncertainties were within $\pm5\%$. He concentration was determined by mass spectrometry, during $^3$He/$^4$He analyses. Helium isotope and $^4$He/$^{20}$Ne ratios were carried out on gas fractions extracted following the same procedure described above and purified following the method described in the literature (Hilton, 1996; Sano and Wakita, 1988; Italiano et al., 2001). The isotopic analyses of the purified helium were performed using a static vacuum mass spectrometer (GVI5400TFT) that allows the simultaneous detection of $^3$He and $^4$He ion beams, thereby keeping the $^3$He/$^4$He error of measurement very low. The used analytical method also requires running alternatively one sample and one purified air shot used as internal $^3$He/$^4$He standard. Typical uncertainties in the range of atmospheric He-type samples are within $\pm1\%$. The $^4$He/$^{20}$Ne ratios were calculated by the relative peak heights measured on the same mass spectrometer.

Water samples of 100 ml were stored in PVC bottles for total alkalinity titration: 50 ml were filtered by a 0.45 $\mu$m filter and acidified by $HNO_3$ 0.1 N for cations (Ca, Mg, Na, and K) determination, whereas the non-acidified samples were collected for anion (Cl, F, and $SO_4$) determination. pH was measured by an electronic device calibrated in situ using buffer solutions. In the laboratory, chemical analyses of the major constituents were carried out by ion-chromatography (Dionex ICS-1100) both on filtered (0.45 mm) acidified ($HNO_3$ Suprapur) water samples (Na, K, Mg, and Ca), as well as on untreated samples (F, Cl, Br, $NO_3$, and $SO_4$). The $HCO_3$ content was measured by standard titration procedures with hydrochloric acid. Typical uncertainties are $\pm5\%$.

A few years ago (2003), one of us (F.I.) collected and analyzed fluid samples from a shallow well located in the Calabria main land nearby the Palizzi mud volcano (PMV in Figs. 1 and 2, located about 25 km to the WNW of the BMV area). Unfortunately, no documentation exists concerning the PMV except our original geochemical data reported in Tables 1 and 2. The PMV was indeed actively venting geofluids at the time of our sampling (2003) but it was then soon destroyed by human activity and cementation.

## 4   Results

### 4.1   Earthquakes

The earthquakes recorded in the BMV area (i.e. the area shown in Fig. 2a) between 1985 and 2017 at depths $\leq 40$ km are 178 (Mw $\leq 4.5$; Table S1). Focal mechanisms are from earthquakes that are $\leq 4.5$ in magnitude and $\leq 30$ km in depth (Table S2). During the considered time window, no larger magnitude earthquakes ($> $ Mw 4.5) are present for this area in the instrumental catalogs. These focal mechanisms are mostly characterized by strike-slip and transtensional faulting along NW-SE-striking planes or along the conjugate planes striking NE-SW.

Fig. 2(b) shows a transect-perpendicular projection (swath profile) of the above-described earthquake data (i.e., hypocenters and focal mechanisms; Tables S1 and S2) along a NE-SW vertical transect through the BMV (see the A-A' transect track in Fig. 2a). Projected seismic events were selected within an arbitrary distance (swath width) of 12 km from the transect track (Fig. 2a). Earthquake hypocenters are diffuse over the studied transect including the area beneath the BMV (Fig. 2b). A cluster of hypocenters is discernible near the northeastern tip of the transect (Fig. 2b). This same cluster is also visible as a NW-trending cluster of epicenters in the northeastern portion of the study area (i.e., 40 km to the northeast of the BMV; Fig. 2a).

### 4.2   Bathymetry and Geomorphology

The study area is located on the upper part of the continental slope of the Calabrian-Ionian margin (Figs. 1 and S1) between 120 m and 2000 m water depth. This area is characterized by a complex morphology due to the interaction between tectonically controlled escarpments and several small scale mass-wasting features, including slide scars (Ss), regional scarps (Rs), and gullies (G) or channels (Ch) (Fig. 3). The upper part of the continental slope is characterized by a very steep slope (about 15° in average dip) that can reach, in places, a maximum dip of even 28° along the main escarpments oriented NE-SW and long up

to 35 km (Fig. 3b). At the foot of the upper continental slope, a well-defined flat area of about 26 km$^2$ occurs encompassing the sub-circular morphological high of the BMV (Fig. 3). The flat area is located at 1350 m water depth and bordered by several small ridges that are elongated parallel to the continental slope. The BMV is characterized by a circular shape with a diameter of about 1100 m and a well-defined rim (Figs. 3 and 4). The BMV has an elevation of about 22 m from its base (i.e.,
the surrounding plain) and steep slopes (see the slope profile in Fig. 4a). The high resolution digital elevation model shows some complex structural and morphological features (Fig. 4). These features consist of concentric morphologies each with its perimetric topographic rim. The concentric morphologies are separated by some moats and encompassed by the outermost topographic rim of the BMV (Fig. 4). Furthermore, on top of the BMV, three minor sub-circular/arcuate bulges are present. These features have a relief of about 3 m with respect to the surrounding seafloor and are separated by small sub-circular
moats (Fig. 4b). The outermost topographic rim of the BMV is interrupted in the southern side (Fig. 4b). Low resolution backscatter data indicate low amplitude of acoustic signals whereas the absence of changes/anomalies in the amplitude indicates a clayey/pelitic sedimentary cover (Fig. 5a).

To better understand the morphology of the BMV, also in comparison with other known mud volcanoes on the Earth, we here consider recent compilations of mud volcano morphological data. We refer, in particular, to the works by Kirkham et al. (2017)
and Kioka and Ashi (2015), where relations between mud volcano height $H$ vs. diameter $D$, and volume $V$ ($= \pi R^2 H/3$) vs. $H/R$ ratio (where $R$ is the radius of the volcano base) are explored, respectively. Thus, the BMV volume $V = 6.9$ x $10^6$ m$^3$ is computed with the previous formula, after estimating its diameter and height through the use of high resolution multibeam bathymetry data ($D$ = 1100 m and $H$ = 22 m, Fig. 4). We used the same given formula to compute mud volcano volumes from the dataset provided by Kioka and Ashi (2015). However, some of these catalogued mud volcanoes are reported without
diameter. Hence, we selected only the 232 mud volcano having with both the diameter $D$ and the height $H$ (Fig. 6a). Fig. 6 shows $H$ vs. $D$ (Figs. 6b and 6c) and $V$ vs. $H/R$ (Fig. 6d) for the 232 selected mud volcanoes. When using the relation $V = \pi R^2 H/3$, volumes are calculated following the method proposed by Kioka and Ashi (2015), corresponding with the volume of the cone even though many volcanoes have a flat-topped summit and may be better approximated to a trunk cone (e.g., potentially the BMV). Figs. 6(b and c) show that some mud volcanoes display a large diameter but low relief, whereas
very few mud volcanoes exhibit small diameter but prominent topography. An approximately linear trend between increasing diameters and heights can be inferred. In this trend, the BMV stands close to the low $D$ and $H$ values. Fig. 6(d) shows that the $H/R$ ratio of all mud volcanoes is $\leq 0.4$. The same ratio for the BMV is $< 0.1$. There is a scattered distribution of volumes ($V$), mostly ranging in the $10^6$-$10^9$ m$^3$ interval. The BMV volume (6.9 x $10^6$ m$^3$) is highlighted in Fig. 6d.

### 4.3    Single-channel Chirp Seismic Reflection Profiles

In the chirp seismic profiles (253, 241, and 246 in Fig. 7b-d), we recognized the following seismostratigraphic units:

Unit U0 corresponds to the seafloor seismic unit of the BMV and is characterized by sharp bottom echoes with no sub-bottom reflections or little acoustic penetration with chaotic reflections and locally large hyperbolae. Adjacent to the BMV, we recognized the following four further seismic units (U1, U2, U3, and U4).

Unit U1 is mostly characterized by parallel to subparallel semi-continuous reflections with variable amplitude. The upper boundary is defined by a horizon (H1) generally characterized by a reflector with high-amplitude and high continuity. Locally (e.g., toward the BMV), amplitude and continuity of horizon H1 decrease and its reflection becomes indistinguishable. Furthermore, the H1 reflector is locally erosive in the sector adjacent to the BMV and tends to become a conformity surface far away from the BMV. Locally, toward the northern escarpment, reflections in U1 are truncated by an erosional surface. Adjacent to the BMV, normal faults displaced this unit.

Unit U2 is characterized by a quasi-transparent (reflection free) seismic facies and shows draping of the underlying relief. This unit has a wedge-shaped morphology, thickening toward the flank of the BMV (241 in Fig. 7c), where it reaches a thickness of at least 25 ms. The lower boundary of U2 is the horizon H1, whereas the upper boundary is defined by a horizon (H2) characterized by a reflector with high to medium amplitude and high continuity that locally (e.g., toward the BMV) become discontinuous. Horizon H2 is an erosional surface with local extent. U2 is distributed in the sector comprised between the northern escarpment and the BMV and is confined by the northern escarpment. Locally, U2 can be recognized also along the southern flank of the BMV edifice (246 in Fig. 7d).

Unit U3 is characterized by high amplitude reflections and by a limited thickness. The internal filling configuration of this unit is a parallel onlap. Locally, the seismic facies is semitransparent. The lower boundary of U3 is represented by horizon H2, whereas the upper boundary is defined by a horizon (H3) characterized by a reflector with high to medium amplitude and high continuity. In places, amplitude and continuity of horizon H3 decrease. Furthermore, horizon H3 is locally characterized by erosion.

Unit U4 is characterized by a transparent/chaotic seismic facies (241, 246, and 253 in Fig. 7b-d). It has a wedge-shaped morphology and thickening toward the northern escarpment (max. thickness of about 20 ms). The lower boundary of U4 is represented by horizon H3, locally characterized by erosion. The upper boundary is the seafloor. U4 is well discernible (max. thickness = c. 20 ms TWT) in the sector comprised between the northern escarpment and the BMV and less discernible but still present (max. thickness = c. 4 ms TWT) to the south of the BMV.

## 4.4 Seabottom Gravity Core

We collected the SF17-01 core at the toe of the peripheral rim marking the external slopes of the BMV (Fig. 7a). Chirp profiles collected during coring operations (Figs. 7b-d and 8) shows a chaotic unit (U4, Figs. 7b-d and 8) resting on the mud volcano. Chirp-core correlation suggests that the chaotic and transparent sediments are represented by the 2-m-thick fining-upward resedimented unit between 0.50 and 2.50 m depth (Fig. 9). The base of this unit is marked by an abrupt increase in sand content and by an erosional basal contact (Fig. 9).

Although the gravity core did not sample mud breccias, it shows indirect evidence of fluid/mud flow as pointed out by the presence of patchy/cloudy facies (*sensu* Staffini et al., 1993; Cita et al., 1996) where sediment disturbance is caused by fluid expulsion. According to Staffini et al. (1993), the main feature of a patchy/cloudy facies is that the mud breccia is characterized by patches and clouds of different colours. No visible clasts larger than sand-size are observed and the mud breccia shows typical colour changes. The sampled section below the base of the resedimented unit is characterized by silty-clay bulk

sediments containing irregularly clustered intervals of differently coloured sediment patches (from dark grey, olive grey and brownish clouds within grey matrix) with several fragmented and vertically dislocated thin silty turbidites and volcanoclastic layers (Fig. 9). Sediments contain several vertical or sub-vertical micro-pipes. The presence of the patchy/cloudy facies in the core is identified by changes in magnetic susceptibility, whereas the sand content does not show diagnostic changes (Fig. 9).

## 4.5   Multi-Channel Seismic Reflection Profiles

The CROP M-31 and CA99-215 profiles (Fig. 10) are located within the inner plateau of the Calabrian Arc, a morphologically flat area with forearc basins formed on the top of the continental basement (Polonia et al., 2011, 2016). The inner plateau of the Calabrian Arc is characterized by a generally thick portion of Plio-Quaternary deposits overlying the Messinian sediments sampled in several nearby boreholes (e.g., Floriana 1 and Fosca 1 wells; Figs. 1 and 10f) and recognized both in the CROP M-31 and in the CA99-215 seismic profiles (Polonia et al., 2016). Below the Messinian deposits, the upper Oligocene-upper Miocene turbidite deposits unconformably overlie the basement units.

The seismic profiles are characterized by different structural domains with peculiar deformation patterns (Polonia et al., 2016). In particular, the Ionian Fault system (see this system in map view in Fig. 1) is observed in the seismic profiles (between s.p. 200 and 1400 of the A99-215 profile; Fig. 10b). The Ionian Fault system, characterized by transtensive faulting, seems to re-activate pre-existing Miocene-Pliocene thrusts (Polonia et al., 2016) and it separates the Calabrian Arc into two sectors: the Western and the Eastern lobes (Figs. 1 and 10). The Eastern Lobe, where the BMV is located, is characterized by a more elevated accretionary wedge and by steeper topographic slopes than those of the Western Lobe, as shown in the sector between s.p. 1500 and 2600 of the CA99-215 profile (Fig. 10; Polonia et al., 2011).

## 4.6   Geochemical Features of Four Sea Water Columns

The analytical results in terms of chemical and isotopic composition of the water samples and the dissolved gases are listed in Tables 1 and 2, respectively. The fault zone data (GeoC1, 2, and 3) are from samples coming from vertical casts performed far from the BMV over the Ionian and Alfeo-Etna fault systems (Fig. 1). The typical composition of an Air Saturated Sea Water (ASSW) is reported for comparison (Tables 1 and 2).

The five sea-water samples collected along the water column over the BMV (except the sample collected near the seabed) show a chemical composition similar to the fault zone samples (GeoC1, 2, and 3; Fig. 11 and Tables 1 and 2). The concentration values for major cations (Na, K, Mg, Ca; Table 1) show similar values for the GeoC1, 2, and 3 samples and the ones from the BMV (including water column and bottom samples). The dissolved gases detected along the water column (Table 2) exhibit geochemical features with large differences from the sea water equilibrated with the atmosphere (ASSW). In particular, the composition of the dissolved gases (Table 2) shows the presence of air-derived gases ($N_2$ and $O_2$) along with non-atmospheric gases ($CO_2$ and $CH_4$). The analytical results clearly display a slight decrease in oxygen content (i.e., compare ASSW and the rest of data in Fig. 11) besides a significant increase in He, $CH_4$, and $CO_2$. The $CH_4$ content along the water column and at the sea bottom (BMV and GeoC1, 2, and 3) ranges over two to three orders of magnitude higher than the ASSW with the highest concentration recorded at the BMV depth (-1337 m). The $CO_2$ content above the BMV, in particular, is the double of

the ASSW (0.24 ccSTP/L) at the depth of 100 m and it increases up to 0.73 ccSTP/L at the sea bottom (-1337 m) (Fig. 11 and Table 1).

The PMV is characterized by a dissolved gas phase mainly composed by Nitrogen and $CH_4$ with a significant helium concentration and a slight amount of $CO_2$. Oxygen is below the detection limits (Tables 1 and 2).

The helium isotopic signature of samples coming from the GeoC1, 2, and 3 casts show an atmospheric signature from the surface to the depth of 1000 m. The bottom sample displays a lower ratio than the air with a higher $^4He/^{20}Ne$ ratio. Two samples from the BMV cast were analyzed to determine the helium isotopes. Results approximately match those from the GeoC1, 2, and 3 casts showing a slight but detectable difference with the respect to the atmospheric ratios (as expected for an ASSW) for both $^3He/^4He$ and $^4He/^{20}Ne$ ratios (Fig. 11 and Tables 1 and 2).

## 5  Interpretation

In this section, we interpret some of the presented data, particularly (but not only) the seismic ones (Figs. 7-10) that require specific interpretation. Further discussion on and synthesis of all data are reported in the next section.

First of all, earthquake epicenters and focal mechanisms (Figs. 1 and 2) show that the area surrounding the BMV is seismically active as it is populated by earthquakes. Concerning the activity of the BMV, consistently with the resolution of the multibeam equipment, multibeam water column data recorded during the Seismofaults 2017 survey does not highlight acoustic backscatter anomalies related to large amount of fluids/mud escaped from the seafloor (Fig. 5b; e.g., Römer et al., 2014). This evidence suggests that no paroxysmal activity is ongoing (or at least was ongoing at the time of the Seismofaults 2017 survey) from the BMV. However, our geochemical data (Fig. 11) show active fluid circulation through the BMV. Further discussion on this theme is proposed in the next section. Moreover, sediments from the gravity core contain several vertical or sub-vertical micro-pipes suggesting sediment reworking by fluid migration (Fig. 9). We associate this sediment structure to the patchy/cloudy facies (e.g., Staffini et al., 1993; Cita et al., 1996), which was already described in the surrounding areas in association with geofluid ascent and mud volcanism (Panieri et al., 2013).

Concerning the single-channel chirp profiles, based on their seismic characters, we interpret the five seismic units identified in Fig. 7 as follows (Fig. 8):

Unit U0 is interpreted as the BMV main edifice. The transition between the rim and the summit caldera is identified by the large hyperbolae. The floor of the summit caldera is not penetrated by seismic signal, possibly indicating the occurrence of mud breccias deposits that are typical of mud volcanoes (van der Meer, 1996; Gennari et al., 2013).

Unit U1 is interpreted as slope deposits including turbidite-hemipelagite intervals. Locally, slope deposits are eroded by U2 (i.e., mud volcano deposits).

Unit U2 is interpreted as a mud volcano deposit belonging to the BMV, due to its wedge-shaped quasi-transparent (reflection free) seismic facies thinning away from the BMV center. This deposit can be related to eruptive events or post eruptive instability of the following types:

(1) Buried mudflow deposits: gravity flow deposits related to slope instability of the mud volcano.

(2) Buried mud volcano sediment: similar wedge-shaped seismic units have been interpreted by Evans et al. (2008) as massive and structureless sediment extruded from the volcano centre.

Unit U3 is interpreted as ponded deposits, including turbidite-hemipelagite intervals and thin mass transport deposits. Locally, ponded deposits are eroded by U4 (i.e., mass transport deposits).

Unit U4 is interpreted as consisting of mass transport deposits originated by slope instability along the northern escarpment (i.e., toward the coast). This interpretation is based on the quasi-transparent (reflection free) seismic facies of this unit and on its wedge shape with reduced thickness moving away from the northern escarpment. Interpretation of U4 is also based on direct evidence from the gravity core of Fig. 9.

Concerning the multi-channel seismic reflection profiles, the CA99-215 profile displays numerous faults characterized by different kinematics and related to the complex evolution of the area, namely: thrust faults, which are the result of the post-Messinian shortening, and normal and strike-slip faults deriving from the extensional process acting in more recent times (from lower Pleistocene to present times) as shown by the faults reaching the sea bottom on the eastern side of the seismic profiles (Fig. 10). Earthquakes focal mechanisms located in the study area (Figs. 1 and 2) show that faults are mostly characterized by strike-slip and transtensional kinematics defining a clear flower-like structure on the western side and an oblique-slip deformation zone on the eastern side (Fig. 10b). In this area, one of the main reflective horizons, usually discernible in most seismic profiles, is the top of the Messinian deposits. The Messinian deposits cored in the nearby wells (in particular in the Squillace Gulf) consist of clay-dominated deposits with interbedded gypsum, anhydrite, and halite (Capozzi et al., 2012). The Floriana 1 well shows, for instance, a thick portion (more than 1300 m) of the Gessoso Solfifera Formation (consisting of crystalline gypsum, anhydrite, and clay layers) whereas, in the Fosca 1 well, the Messinian deposits are mostly formed by clay and silt for a total thickness of 110 m (Fig. 10f). The distribution and sedimentary pattern of the Messinian deposits suggest indeed that the basin was already articulated and tectonically controlled at the time of Messinian sedimentation (Capozzi et al., 2012). The occurrence of Messinian evaporites as well as clays and silts of the same age is very relevant for mud volcanism. Indeed, according to many studies on mud volcanoes in the Ionian Sea and in the adjacent Mediterranean Ridge (located off southern Peloponnesus and Crete), the disruption of the Messinian low-permeability layers concurred to or was the key process controlling the ascent of pressurized fluids entrapped below these sealing layers and the consequent outflow and mud volcanism (Capozzi et al., 2012).

The identification on seismic profiles of submarine mud volcanoes (i.e., their subsurface roots) can depend on and be limited by different factors, such as the complexity of the tectonic setting and/or the seismic characteristics of the hosting succession (Dimitrov, 2002; Bertoni et al., 2017). Identification of mud volcanoes located in complex tectonic settings, such as the accretionary complex of the present work, is not straightforward; however, as we will explain below, we were able to observe a few seismic features that could be interpreted as diagnostic of mud volcanism and fluid-rock interaction (e.g., Dimitrov, 2002). For instance, immediately beneath the sea floor, where a mud volcano is located, seismic reflectors are usually characterized by locally strong amplitudes. This seismic effect can be associated with fluid-rock interaction (Dimitrov, 2002). Furthermore, chaotically disrupted seismic pattern and short seismic horizons are typical features related to the presence of a conduit and fluid-rock interaction beneath many mud volcanoes. Concerning the BMV (Figs. 10d, 10e, and S2), we observed the following

main features: (1) the presence of top Messinian deposits reflectors; (2) the disruption of these reflectors; (3) the presence of normal faults; and (4) the presence of an area of fluid-rock interaction similar to many ones observed in seismic profiles below mud volcanoes (Capozzi et al., 2012; Dimitrov, 2002; Bertoni et al., 2017). In particular, the area just below the BMV, which is about 300 m far away from the CA99-215 profile (Fig. 1), shows (at a time-depth of 2 s TWT) a rock volume characterized by a chaotic seismic reflection signal, with disrupted and short reflections (Figs. 10d, 10e and S2). This chaotic pattern is located in sediments otherwise characterized by reflectors showing high continuity. It is relevant to note that this area is located right at the top of a series of faults. The Messinian evaporites are located between 2 and 3 s TWT. Considering a seismic velocity of 1500 m/s for the sea water column and 2000 m/s for the post-Messinian column of sediments (Gallais et al., 2012), the Messinian evaporites should occur at about -3000 m.

## 6 Discussion

### 6.1 Origin and Activity

#### 6.1.1 General Context

Concerning the origin of the BMV, we here reconsider all the above-described data. First of all, we mention the fact that the Ionian Sea, where two active accretionary prisms occur and obliquely-converge (i.e., the Calabrian Arc and the Mediterranean Ridge), is a region rich of active or recently-active mud volcanoes (Cita, 1981; Camerlenghi et al., 1992; Capraro et al., 2006; Serpelloni et al., 2007; Praeg et al., 2009; Billi et al., 2011). The BMV itself was previously signaled and hypothesized as a mud volcano by Gutscher et al. (2017) and Loher et al. (2018). The BMV occur on top of the Calabrian accretionary wedge in a rather isolated location with several further mud volcanoes occurring a few kilometers or a few tens of kilometers toward NE along the prism (Fig. 1; Loher et al., 2018).

#### 6.1.2 Morphology

From a morphological point of view, the main geometric features of the BMV are those typical of most mud volcanoes (i.e., Kioka and Ashi, 2015) and all its measured parameters (height $H$, diameter $D$, and volume $V$; Fig. 6) are well within the range typical of marine mud volcanoes (Kioka and Ashi, 2015; Kirkham et al., 2017). From the morphological analysis, the BMV could be primarily interpreted as a pie-type mud volcano; however, looking at the slope values highlighted by the acquired multibeam data (Fig. 4), the $< 5°$ slope angle criterion proposed by Kopf (2002) is not satisfied, being the BMV slope values $> 10°$ (Fig. 4).

The stratigraphic analyses realized through direct and indirect methods (Figs. 7-10) show the occurrence of lithological units consistent with the activity of a mud volcano, such as the U2 deposits in Figs. 7c and 8, the evidence of fluid/mud flow in the patchy/cloudy facies of the gravity core (Fig. 9), and the chaotic seismic reflection signal with disrupted and short reflections (typical of mud volcanoes for the host rock-geofluid interaction; e.g., Dimitrov, 2002; Capozzi et al., 2012) beneath the BMV (Figs. 10d). However, from the available data, we cannot infer any eruptive dynamics for the BMV due to the limited resolution

of the seismic imaging and to the BMV location with respect to the seismic lines (c. 300 m). The dimension of the conduit and paleoflows are not visible in the seismic profiles, as, for example, the Christmas-tree structures described by (Somoza et al., 2003). From previous literature (e.g., Kioka and Ashi, 2015), we can only infer that the BMV dimensions and its computed volume suggest a polygenetic behaviour, so that we argue that the main fluid conduit has been possibly utilized several times

to increase the volume of the volcano itself.

### 6.1.3  Sealing

Particularly in the Mediterranean Sea, the origin of most mud volcanoes has been linked in a cause-effect relationship with the sealing action exerted by the Messinian evaporites, causing fluid entrapment underneath and consequent fluid overpressure (Camerlenghi et al., 1995; Chamot-Rooke et al., 2005; Camerlenghi and Pini, 2009; Capozzi et al., 2012; Ceramicola et al.,

2014; Rovere et al., 2014; Bertoni and Cartwright, 2015). Also in the case of the BMV, the seismic cross-sections show the presence (and disruption underneath the BMV) of the Messinian evaporites (Fig. 10), which therefore may have been decisive in building the necessary overpressure of fluids to consequently form the mud volcano itself (Bertoni and Cartwright, 2015; Al-Balushi et al., 2016). The disruption of the Messinian layers suggests that the conduit for the ascent of geofluids through the BMV is presently open. This hypothesis is also substantiated by the geochemical data (Fig. 11) that are discussed below.

### 6.1.4  Ongoing Activity

We have very little evidence to discuss the ongoing activity of the BMV also because we collected data from this structure only in a single campaign in May 2017. From a morphological point of view, the BMV seems well structured (Fig. 4) and therefore its main edifice-building paroxysmal activity may have substantially ceased. Moreover, the flanks of the main volcanic edifice seems partly covered by younger products (i.e., over its flanks; Figs. 7c-d and 8) deriving from nearby gravity

instabilities (i.e., from the continental slope of the Calabrian-Ionian margin; Figs. 1 and S1). Also, the backscatter data show no extensive anomalies (in May 2017) related to large amount of mud and fluids escaped from the seafloor (e.g., Römer et al., 2014); however, this evidence does not detract from the fact that the volcano may currently be quiescent and therefore may erupt in the future. To this end, both the geochemical and the reflection seismic evidence show that some fluid activity below the BMV is probably ongoing. Fig. 11, in particular, shows a trend of $CH_4$ and $CO_2$ enrichment for all the collected samples

with respect of a sea water simply equilibrated with the atmosphere (ASSW). Although $CO_2$ and $CH_4$ may derive from degradation processes of organic matter, the geochemical composition of the sea water at the BMV depth and the composition of the geofluids from the PMV (Palizzi) onland area clearly indicate a $CH_4$ injection that is typical of most mud volcanoes. The isotopic composition of helium, although dominated by a typical atmospheric signature both at the GeoC1, 2, and 3 localities and at the BMV locality, displays a detectable increase of radiogenic $^4$He of typical crustal origin, with the associated decrease

of the isotopic ratio from about 0.93-1 Ra to 0.77 Ra detected in the GeoC1, 2 and 3 seabottom waters and 0,73Ra in the BMV seabottom water (Tables 1 and 2). Hence, we propose that the BMV is actually infiltrated by open pathways as shown by the release of fluids into the surrounding sea water (Fig. 11 and Tables 1 and 2). Fluids are composed by a two-phase system: a $CH_4$-dominated gas phase and hypersaline waters of evaporite type. The hypersaline waters are indeed probably generated

by the dissolution of anciently buried evaporites (Messinian) and create dense anoxic brines, which are separated from the overlying oxygenated deep-seawater column due to differential densities. The chaotic seismic reflection signal, with disrupted and short reflections (typical of active mud volcanoes; e.g., Dimitrov, 2002; Capozzi et al., 2012) recorded beneath the BMV (Fig. 10) as well as the disruption of the Messinian evaporite layers supports the above-proposed hypothesis of pathways open to geofluids down to the Messinian evaporite layers (c. 3000 m depth; Fig. 10). Also the gravity core bears evidence of recent fluid circulation (Fig. 9). Moreover, some structures on top of the BMV (i.e., the rim and some small ridges; Fig. 4a-b) are morphologically similar to structures related to extrusion activity of mud volcanoes (e.g., Evans et al., 2008) and are therefore probably connected with the eruptive processes. These structures are not substantially covered by young sediments, hence attesting for a recent but undetermined time for the eruptive process of the BMV.

### 6.1.5 Draining Processes

In accretionary prisms, the causes (i.e., the engine) of mud volcanoes and related fluid activity are often found or hypothesized to be either the prism contraction and related fluid squeezing or the fault activity and related fluid ascent along fault damage zones. In other words, mud volcanoes can be caused by a contraction-related local dewatering or by a deeper crustal draining driven by the activity of normal and strike-slip faults (Gamberi and Rovere, 2010; Capozzi et al., 2012; Panieri et al., 2013; Ceramicola et al., 2014; Rovere et al., 2014). In the case of the BMV, we cannot unambiguously discriminate between these two main engines (prism contraction vs. fault activity). We hypothesize that both engines may concur or may have concurred to originate the BMV. Overall contraction is indeed slightly active in the Calabrian accretionary prism (Serpelloni et al., 2007; Billi et al., 2011; Faccenna et al., 2014; Polonia et al., 2016) as well as fault activity, particularly along prism-across (NW-SE) strike- to oblique-slip faults (Polonia et al., 2016, 2017). It is also true, however, that the seismological and reflection seismic data show the occurrence of a seismically-active flower-like system of faults right beneath the BMV (Figs. 2 and 10). This evidence let us think that (seismic) faulting more than the prism contraction process may have played a decisive role in the origin and feeding of the BMV, thus ultimately driving the ascent of geofluids from crustal depths. To this end, Polonia et al. (2017) have recently documented that the prism-across strike-slip faults in the Ionian Sea region are active and have been even capable of exhuming or contributing to exhume serpentinite domes from the lower plate of the Ionian subduction zone up to the upper plate and the Earth's surface.

### 6.1.6 Synthesis

Collectively, the data presented in this paper provide evidence for the fact that the studied structure (i.e., the BMV) is actually a mud volcano, through which fault-related crustal fluid activity and circulation is ongoing, and beneath which (seismic) faulting is active.

## 6.2 Potential Use of the BMV in the Science of Seismic Precursors

### 6.2.1 Previous Useful Results

Concerning the use of the BMV and similar structures in favor of the monitoring and mitigation of natural hazards, we first refer the reader to a few recent studies on geochemical precursors of earthquakes. Transient hydrogeochemical anomalies are increasingly becoming commonly recorded before M≥5.0 earthquakes at distances between 20 and more than 200 km from the earthquake epicenters. To understand their relevance for earthquake forecasting, we here briefly recall a few recent instances from Italy, Iceland, India, Japan, and Turkey. The 2016 Amatrice and Norcia earthquakes (central Italy) as well as the related sequence involved significant pore pressure changes (since about one week before the Amatrice earthquake; De Luca et al., 2018) and fluid movements both at deep and shallow crustal levels (Petitta et al., 2018; Tung and Masterlark, 2018), and were anticipated by hydrogeochemical anomalies recorded since April 2016 in springs from the central Apennines. In particular, increases of As, V, and Fe contents were recorded in groundwaters from springs monitored in the Sulmona area, about 70 km to the southeast of the epicentral area. Similar anomalies (i.e., As, V, and Fe) were also recorded in groundwaters from the San Chiodo spring located within the epicentral area (Barberio et al., 2017; Boschetti et al., 2019). In 1995, eight months before the Mw 7.2 Kobe earthquake (Japan), the Cl and $SO_4$ concentrations in groundwater started to significantly and anomalously increase (20-30 km from the epicenter). Nine days before the earthquake, a peak in Rn concentration was also recorded (King et al., 1995). In 2002, anomalies in Cu, Zn, Mn, and Cr concentrations were recorded in groundwater 1, 2, 5, and 10 weeks, respectively, before a Mw 5.8 earthquake in northern Iceland (90 km from the epicenter; Claesson et al., 2004). In 2012, anomalous increases of Ca, Mg, K, and Cl concentrations in groundwater together with decreases of Na and $SO_4$ concentrations started between at least 20 and 30 days before the Mw 7.1 Van earthquake, Turkey (20 km from the epicenter; Inan et al., 2012). In 2012, significant increases in the Na, Si, and Ca concentrations were recorded in groundwater 4-6 months before two Mw≥5.5 earthquakes in northern Iceland (70-80 km from the epicenter; Skelton et al., 2014, 2019). In 2004 and 2005, transient hydrogeochemical anomalies were detected in aquifer located to north of the Shillong Plateau, Assam, India, before two Mw≥5.0 earthquakes (200 km from the epicenter). The [Na+K]/Si, Na/K, and [Na+K]/Ca ratios as well as conductivity, alkalinity, and Cl concentrations began increasing 3-5 weeks before the Mw 5.3 earthquake whereas the Ba/Sr ratio began decreasing 3-6 days before the Mw 5.0 earthquake (Skelton et al., 2008). In 2017, oxygen isotopic ratio anomalies of +0.24 $^0/_{00}$ relative to the local background were recorded in groundwater a few months before the Mw 6.6 Tottori earthquake in southwest Japan (5 km from the epicenter; Onda et al., 2018). Although the aforementioned scientific results are certainly encouraging, we must acknowledge that earthquake deterministic forecasting is still unfeasible, both through hydrogeochemical data and through other evidence.

### 6.2.2 The BMV

The geochemistry of the sea water column (sampled in May 2017) above the BMV compared with the sea water benchmark (ASSW; Fig. 11) shows a clear mineralization of the BMV-related waters together with an injection of $CO_2$ and $CH_4$, particularly in proximity of the BMV depth (Fig. 11). Moreover, the Helium isotope ratios (Table S4) shows a contribution by

crustal fluids, also in this case particularly in proximity of the BMV depth. The ion content of the BMV-related waters (Table 1) is consistent with evaporite-type waters and this notion, in turn, is consistent with the hypothesis that the fluids feeding the BMV and other mud volcanoes in the Mediterranean area are entrapped below and within the Messinian evaporites (Fig. 10d; Camerlenghi et al., 1995; Chamot-Rooke et al., 2005; Capozzi et al., 2012; Ceramicola et al., 2014; Rovere et al., 2014; Bertoni

and Cartwright, 2015). Below the BMV, these rocks occur at about -3000 m (Fig. 10d). The $CO_2$ and $CH_4$ (particularly $CH_4$) high content of the BMV-related waters is consistent with most mud volcanoes around the world (Etiope and Milkov, 2004). Moreover, the decreasing content of $CO_2$ and $CH_4$ moving (shallowing) from the BMV summit upward to the sea surface is a clear symptom that the source of these dissolved gases is the BMV itself. We therefore infer that, although the BMV is not likely undergoing full mud-volcanic activity (at least during our survey in May 2017; see water backscatter data in Fig. 5b)

open crustal pathways for geofluids through this structure exist and are actively venting. This hypothesis is also corroborated by the comparison with the geochemistry of three sea water columns above active fault zones in the Ionian Sea (GeoC1, 2, and 3; Figs. 1 and 11). These three sea water columns are indeed characterized by a very low content of $CH_4$ and by a content of $CO_2$ significantly lower than that obtained for the sea water right on top of the BMV (i.e., at the BMV depth; Fig. 11).

### 6.2.3  Synthesis

Collectively, the geochemical, geophysical, and geologic data presented in this paper show that the BMV, likewise other onshore monitoring stations previously realized (e.g., Claesson et al., 2004; Skelton et al., 2014, 2019; Barberio et al., 2017; Huang et al., 2017), could be a proper site where installing a cabled submarine multiparametric station (Fig. 12) to study possible relationships between the seismic cycle of the underlying active faults and geofluid emissions. Similar stations are active onshore in Italy, Iceland, China, and elsewhere, but, to the best of our knowledge, have never been installed in marine

settings. In the case of the BMV and other mud volcanoes, dissolved gases such as $CO_2$ and $CH_4$ may rather easily be monitored by submarine devices (Annunziatellis et al., 2009; Roberts et al., 2017; Boschetti et al., 2019). In particular for the seismically-active Ionian Sea, many other existing mud volcanoes (Gutscher et al., 2017; Loher et al., 2018) may host a monitoring station, but the BMV is so far the one characterized by the largest geological, geophysical, and geochemical dataset and its location seems connected with seismically-active faults (Fig. 2).

## 25  7  Conclusions

Mud volcanoes are ubiquitous structures both inland and offshore. Their occurrence is easily discernible mainly based on morphological-bathymetric evidence, but their geological significance is rather difficult to ascertain particularly offshore. With the Bortoluzzi Mud Volcano (BMV), we have demonstrated that an integrated geological, geophysical, and geochemical study can shed much light on the geological significance and ongoing activity of a mud volcano, even if located in marine settings.

We now know, for instance, that the BMV is truly a mud volcano, it sits atop seismically-active faults, and its inner pathways are actively used for the rise of saline geofluids towards the Earth's surface from a depth of at least -3000 m. Although the paroxysmal activity of the BMV seems substantially extinct or at least quiescent, these inner pathways are open and used for

the rise of geofluids more efficiently than nearby active fault zones that are usually considered efficient pathways for geofluids. This latter evidence constitutes a novelty, at least for offshore seismically-active areas, and indicates that mud volcanoes could be efficiently used to install cabled stations to monitor the relationship between the seismic cycle of faults and the rise of geofluids. In particular, evidence from our work indicates that this type of stations may be installed also where mud volcanoes

seem extinct for what concerns the main paroxysmal mud activity but still own efficient inner pathways for the circulation of geofluids.

*Data availability.* All data used for this paper are available in numerical and graphical forms in the tables and diagrams/images, respectively, in the paper itself or in the supplement associated to this paper. The gravity core collected from the Ionian seabottom is visible at ISMAR Bologna, Italy (http://www.ismar.cnr.it/organizzazione/sedi-secondarie/bologna) through the following contacts: alina.polonia@bo.ismar.cnr.it

and luca.gasperini@bo.ismar.cnr.it. With this paper, we release 4970 km$^2$ of newly-acquired (during the Seismofaults 2017 cruise) high-resolution bathymetric data in numerical form for the Ionian Sea including the BMV (Fig. S1). This dataset is externally hosted and indexed with doi: XXXX.

*Author contributions.* All authors actively participated in conceiving the experiment and the paper, in discussing all results, in contributing to the writing of the paper and to the drawing of all figures, and in drawing the conclusions. ABil and MC coordinated the experiment. MC

and Abos led the fieldwork. MC, ABos, ACor, CGC, ACon, ACos, GD, GL, LG, LP, and TS participated to the Seismofaults 2017 marine campaign on board the R/V Minerva Uno. Data were mostly processed by MC, ABos, CGC, ACon, ACor, EM, FI, GL, LP, ML, PE, and AP. The manuscript was mostly written by AB with contributions from all authors. Figures were mostly drawn by MC, Abos, CGC, ACon, EM, LP, and AP with contributions from all authors.

*Competing interests.* The authors declare that they have no conflict of interest.

*Acknowledgements.* All used data are reported in the figures and tables of this paper and related supplementary material. Moreover, with this paper, we release 4970 km$^2$ of newly-acquired (during the Seismofaults 2017 cruise) high-resolution bathymetric data in numerical form for the Ionian Sea including the BMV (Fig. S1). The raw bathymetric data are externally hosted and indexed with doi: XXXX. M. Barbieri, L. Beranzoli, F. Frugoni, S. Monna, and many other colleagues from CNR, INGV, and Sapienza University of Rome are thanked for help and constructive exchanges. The officers and the crew of the R/V Minerva Uno and the scientific party of the Seismofaults 2017

survey are thanked for their cooperation during fieldwork. Some of the figures were produced with the Generic Mapping Tools software (http://gmt.soest.hawaii.edu). We thank F. Rossetti, M. Allen, and two anonymous reviewers for their constructive comments and editorial support. The research described in this paper is dedicated to Giovanni Bortoluzzi.

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

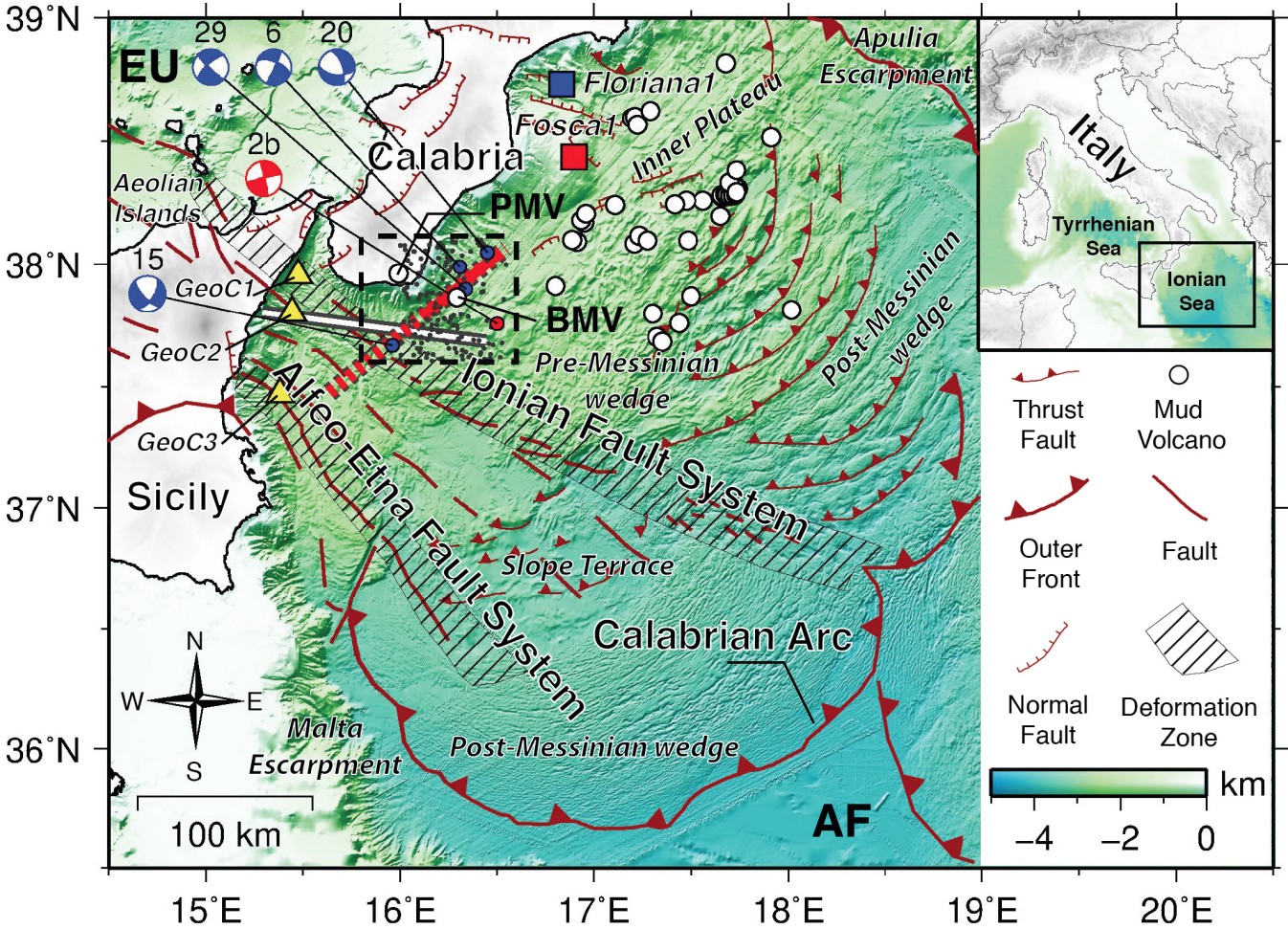

**Figure 1.** Bathymetric map the Ionian Sea (southern Italy) with main faults, mud volcanoes, earthquakes, tracks of multichannel seismic profiles, and location of the Fosca 1 (red square) and Floriana 1 (blue square) offshore wells. Location of mud volcanoes (except the Palizzi mud volcano, PMV, inland Calabria) are from Loher et al. (2018). The Bortoluzzi Mud Volcano is indicated with BMV.The thick white line corresponds to the CROP M-31 multichannel seismic profile (Fig. 10a). Dashed red line and solid red line are the CA99-215 multichannel profile (Fig. 10b) and its close up view reported in Fig. 10(c), respectively. Earthquake data (epicenters are indicated with thin grey dots whereas focal mechanisms with beach balls) surrounding the BMV area are from the European-Mediterranean Regional Centroid-Moment Tensors (RCMT) catalog (http://rcmt2.bo.ingv.it/) (red beach ball) and from previous papers (blue beach balls) by Orecchio et al. (2014) and Polonia et al. (2016). Faults are principally from Polonia et al. (2011), Polonia et al. (2016) and Bortoluzzi et al. (2017). The black box is the location of Fig. 2. AF stands for Africa Plate whereas EU for Eurasia Plate. GeoC1, GeoC2, and GeoC3 are locations of sea water column sampling.

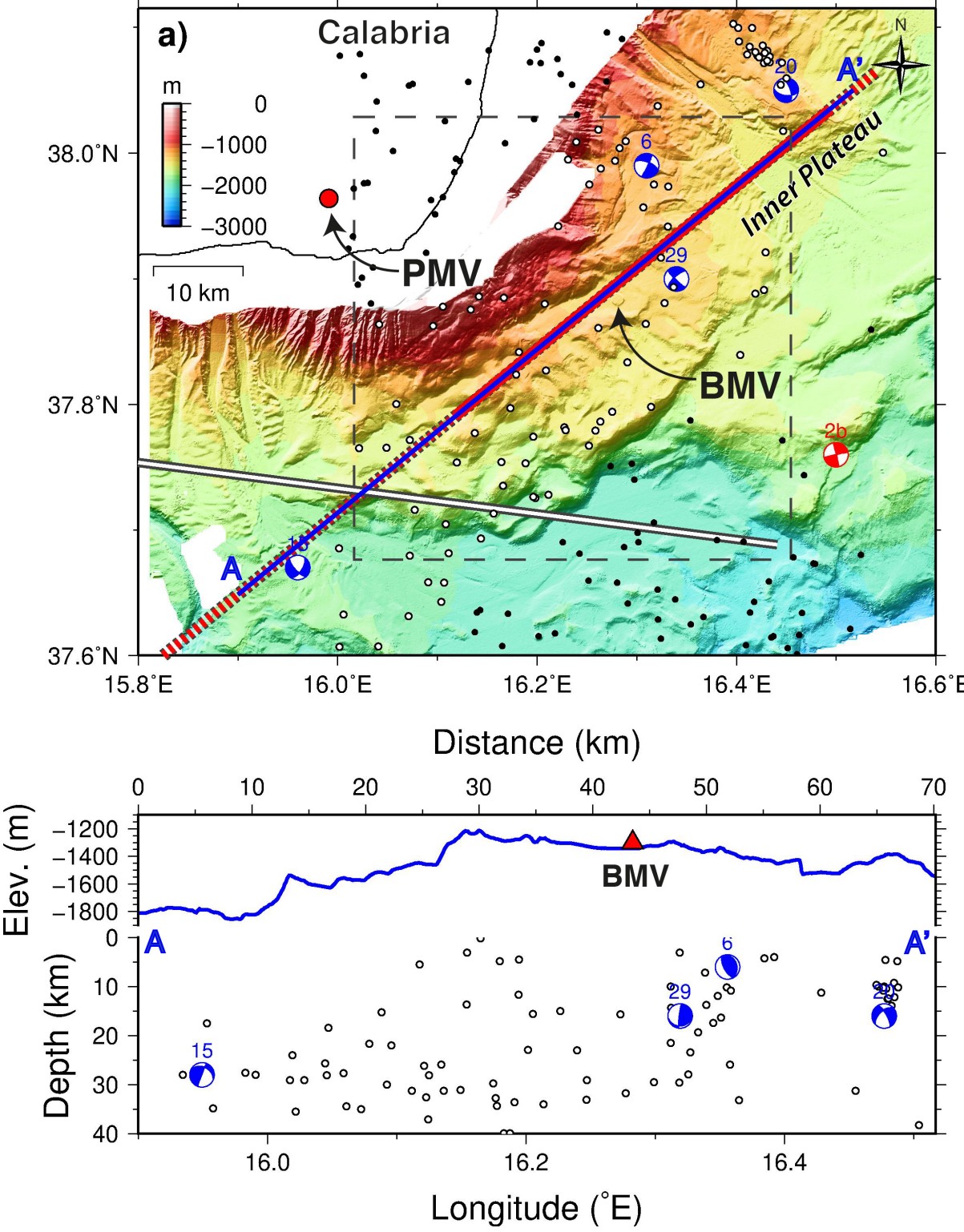

**Figure 2.** (Caption next page.)

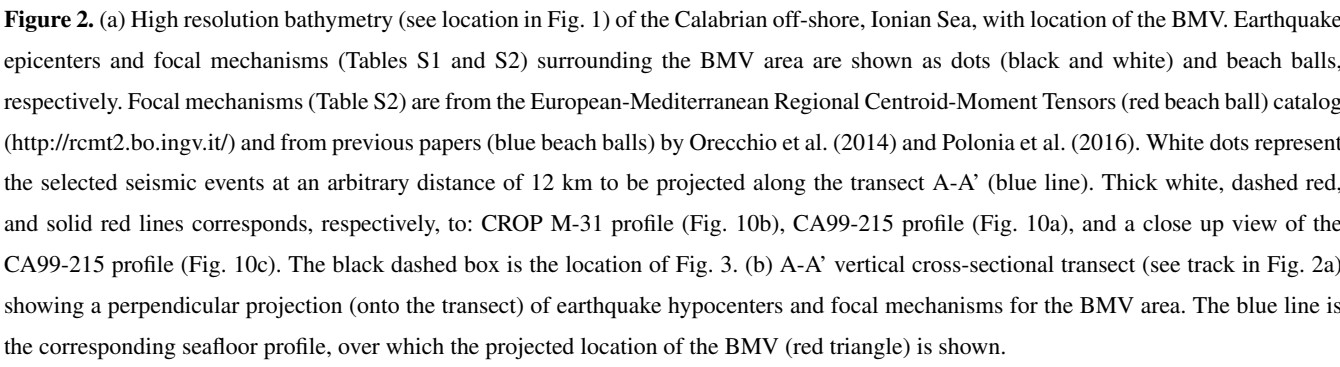

**Figure 2.** (a) High resolution bathymetry (see location in Fig. 1) of the Calabrian off-shore, Ionian Sea, with location of the BMV. Earthquake epicenters and focal mechanisms (Tables S1 and S2) surrounding the BMV area are shown as dots (black and white) and beach balls, respectively. Focal mechanisms (Table S2) are from the European-Mediterranean Regional Centroid-Moment Tensors (red beach ball) catalog (http://rcmt2.bo.ingv.it/) and from previous papers (blue beach balls) by Orecchio et al. (2014) and Polonia et al. (2016). White dots represent the selected seismic events at an arbitrary distance of 12 km to be projected along the transect A-A' (blue line). Thick white, dashed red, and solid red lines corresponds, respectively, to: CROP M-31 profile (Fig. 10b), CA99-215 profile (Fig. 10a), and a close up view of the CA99-215 profile (Fig. 10c). The black dashed box is the location of Fig. 3. (b) A-A' vertical cross-sectional transect (see track in Fig. 2a) showing a perpendicular projection (onto the transect) of earthquake hypocenters and focal mechanisms for the BMV area. The blue line is the corresponding seafloor profile, over which the projected location of the BMV (red triangle) is shown.

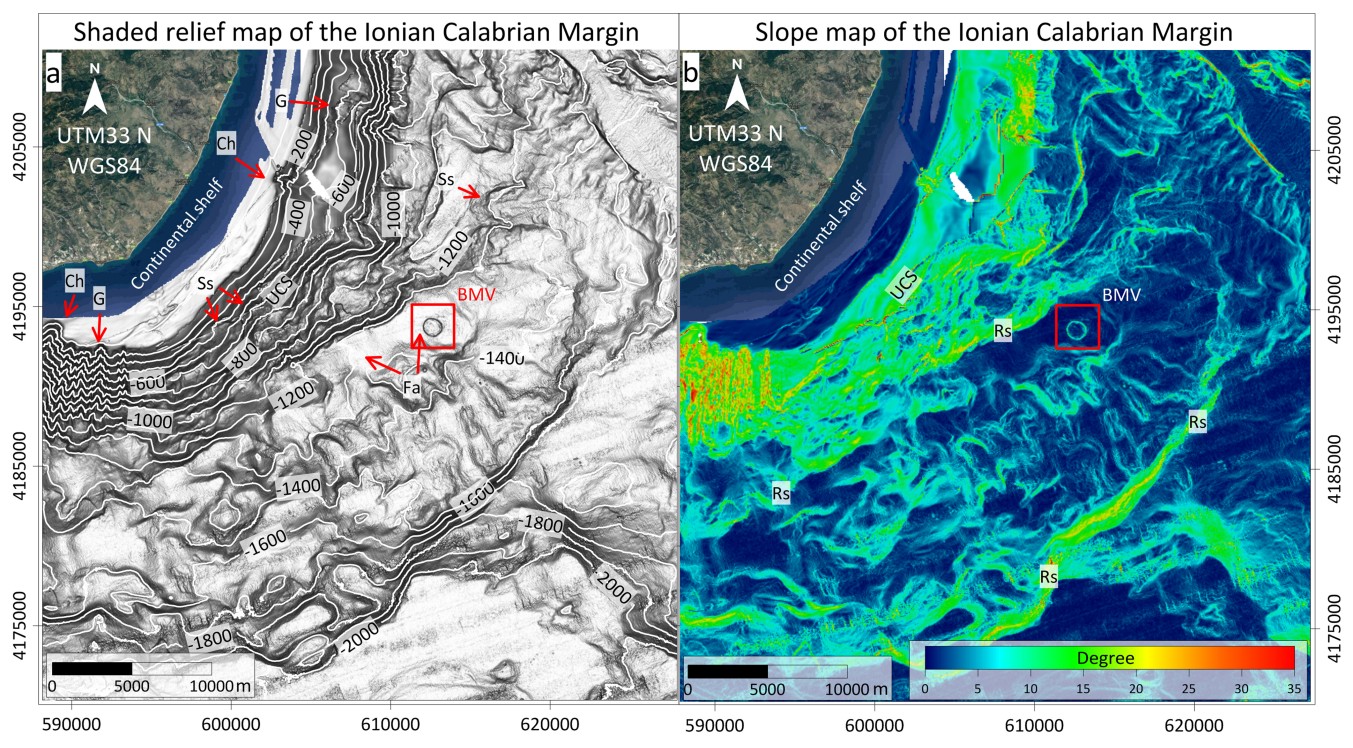

**Figure 3.** (a) Bathy-morphological map and main morphological features: canyon head (Ch), gullies (G), slide scars (Ss), regional scarps (Rs), and a 26 km$^2$ flat area (Fa) are indicated. (b) Slope map of the Upper portion of the Continental Slope (UCS) of the Calabrian-Ionian margin. Red square indicates the intraslope flat area hosting the circular high of the Bortoluzzi Mud Volcano (BMV). Location map is shown in Fig. 2a.

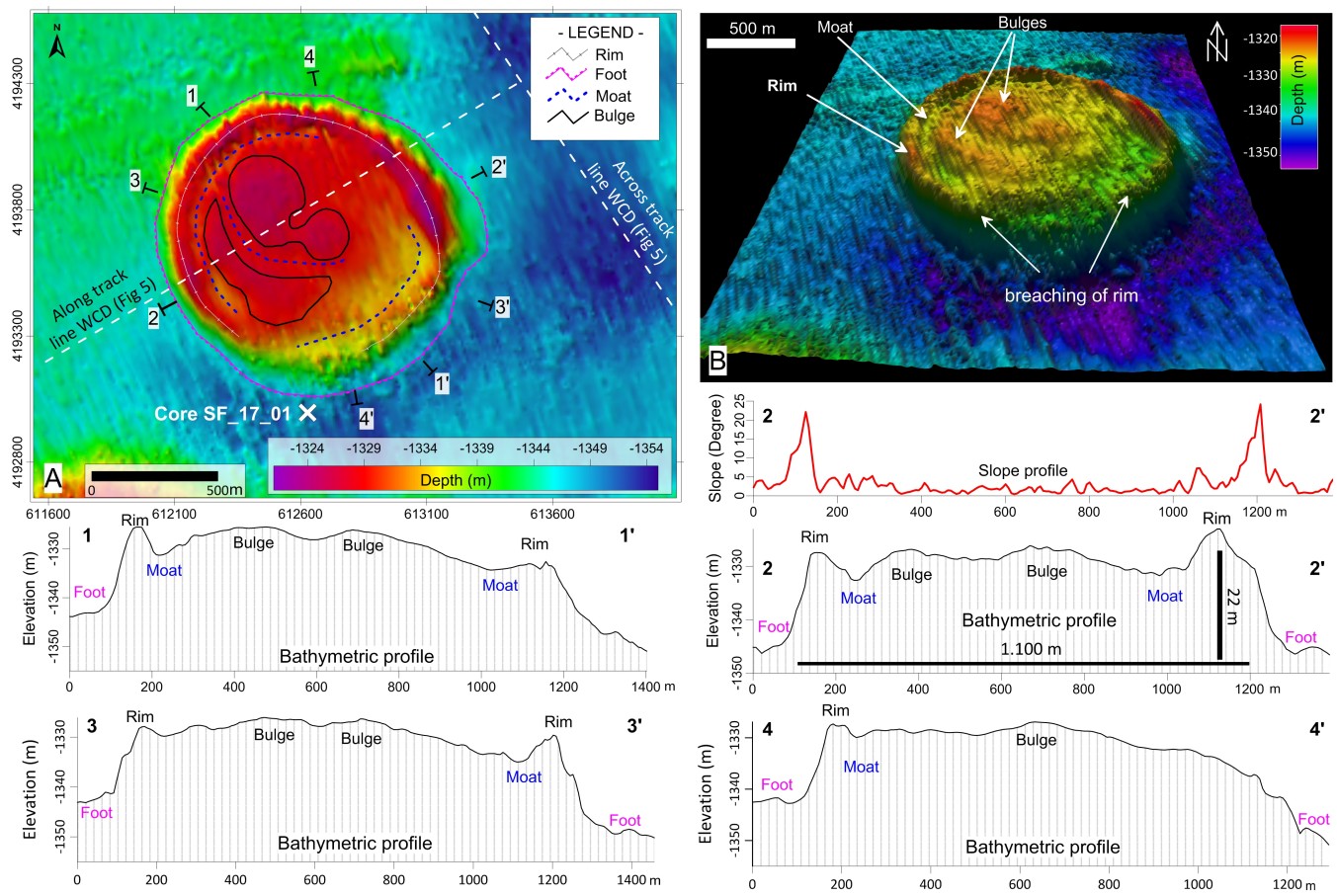

**Figure 4.** (a) Shaded relief map of the BMV with the main morphological features and location of bathymetric sections (1-1', 2-2', 3-3', and 4-4'). The bathymetric sections (vertical exaggeration 10 x) show the morphological features of the mud volcano with vertical slopes up to 28° and two main concentric bulges separated by moats. (b) Digital 3D perspective view of the BMV showing some main features and the breaching of the rim on the southern part of the BMV. In Fig. 4a, the track position (along and across) of the water column data (WCD) are shown (see Fig. 5). See also the Movies S1 and S2.

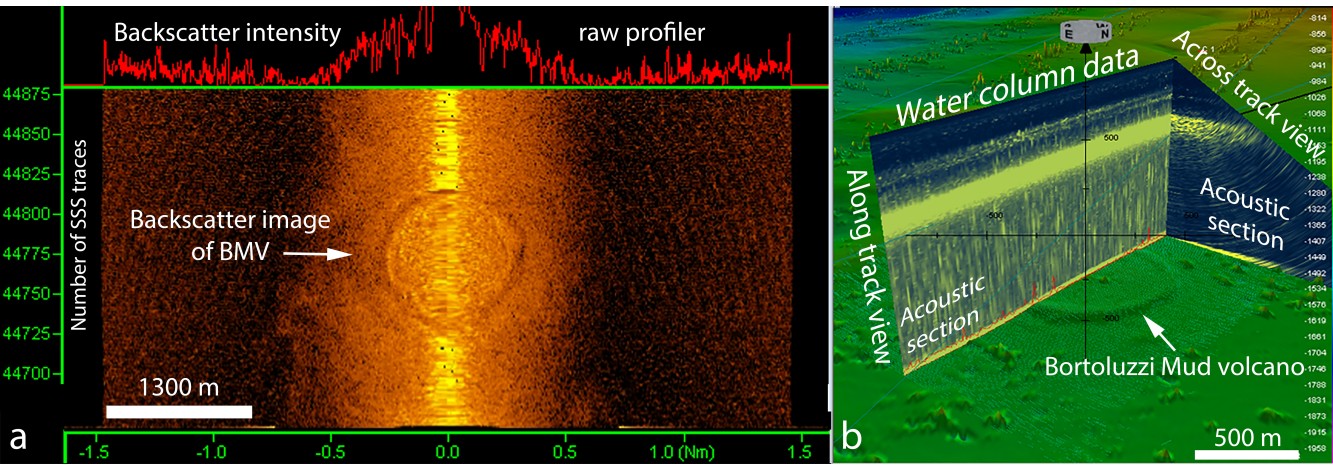

**Figure 5.** (a) Multibeam raw backscatter data. The raw backscatter data does not show amplitude anomalies on the top and around the BMW, indicating a mud homogeneous sedimentary cover. (b) Backscatter water column data recorded by multibeam system. The backscatter water column data does not show amplitude anomalies above the mud volcano excluding significant paroxysmal fluid/mud escape from the seafloor. Location tracks of the acoustic sections are shown in Fig. 4.

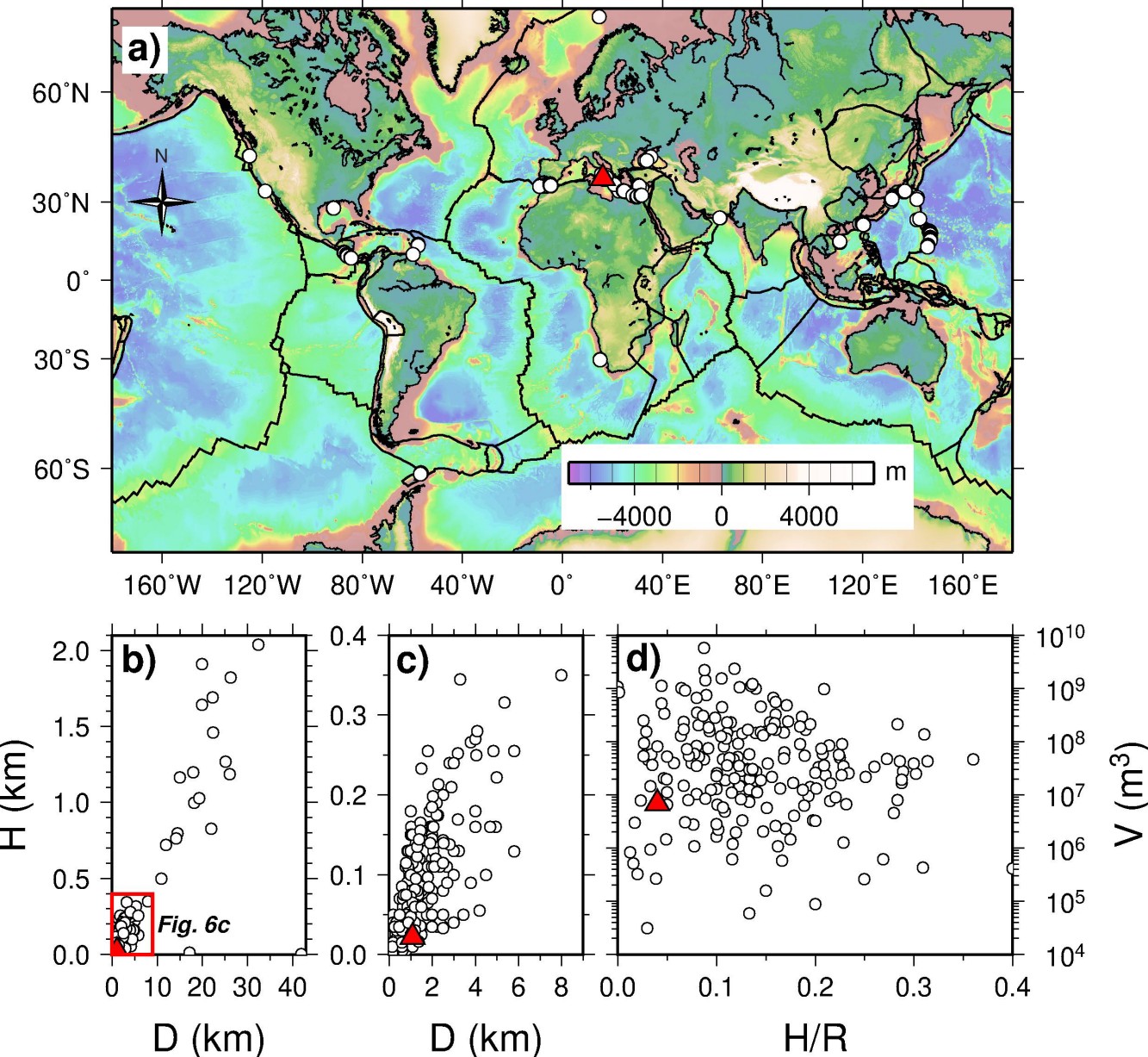

**Figure 6.** (a) Selection of worldwide distribution of mud volcanoes (white dots) from the dataset reported by Kioka and Ashi (2015). In particular, 232 volcanoes with available mean diameter $D$ and height $H$ are shown. The red triangle is the BMV location. (b) Compilation of mud volcanoes diameter $D$ vs. height $H$, showing an approximately linear trend between increasing mud volcano diameters ($D$) and heights ($H$), excluding some exceptions. Along this trend, the BMV (red triangle) stands close the lower values. (c) Close-up view of a portion of the diagram in b. (d) Compilation of volumes $V$ of mud volcanoes vs. $H/R$ (where $R$ is the radius of the volcano base), showing that the $H/R$ ratio is $\leq 0.4$ for all mud volcanoes, whereas there is a scatter distribution of mud volcano volumes, mostly ranging in the $10^6$-$10^9$ m$^3$ interval. The BMV volume (red triangle) corresponds to $6.9 \times 10^6$ m$^3$.

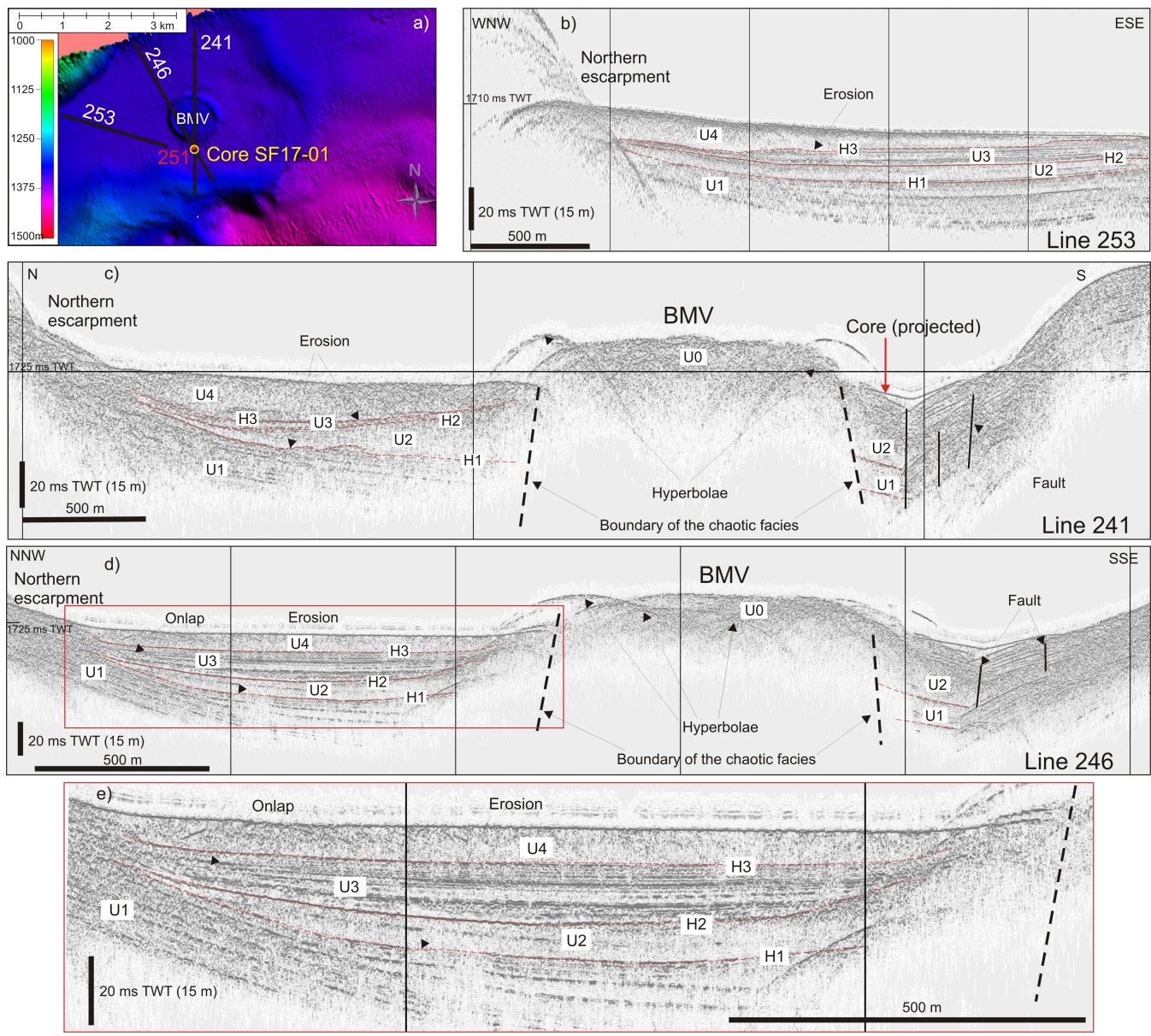

**Figure 7.** (a) Location map of three single-channel chirp profiles and one gravity core in the BMV area. The red dot (numbered 251) below the location of the gravity core indicates the location of the single-channel chirp profile 251 shown in Fig. 9. (b) Line 253. (c) Line 241. (d) Line 246. (e) Enlargement of Line 246 (see the red rectangle in Fig. 7d). Chirp profiles show main seismic units (U0-U4) and bounding horizons identified in the shallow subsurface. The high-resolution non-interpreted version of this figure is available in the Supplement (Fig. S1).

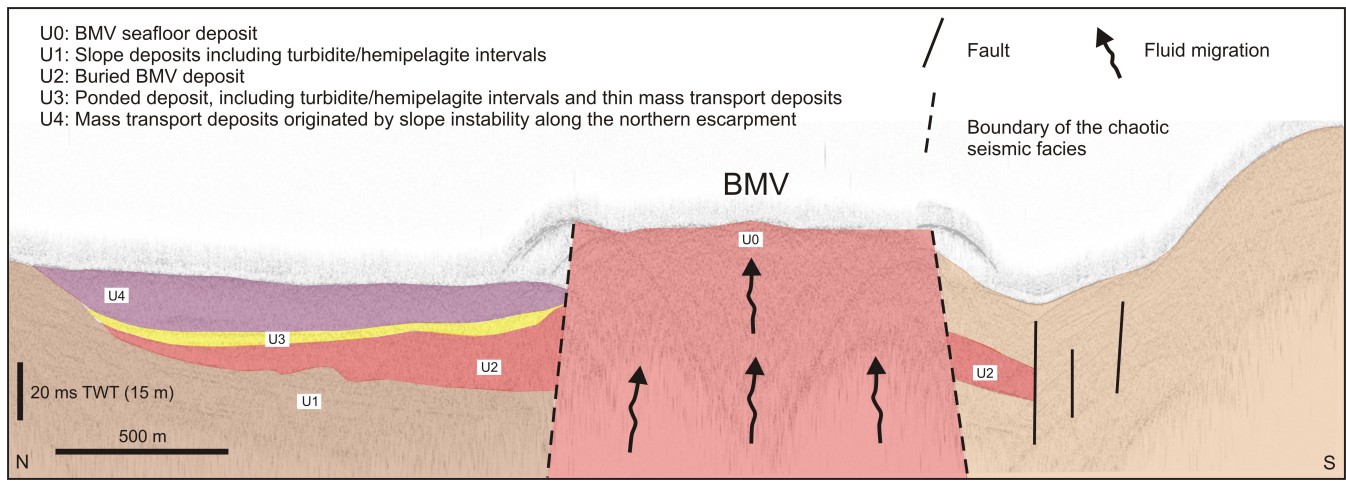

**Figure 8.** Interpretation of the chirp profile 241 shown in Fig. 7 (see text for details).

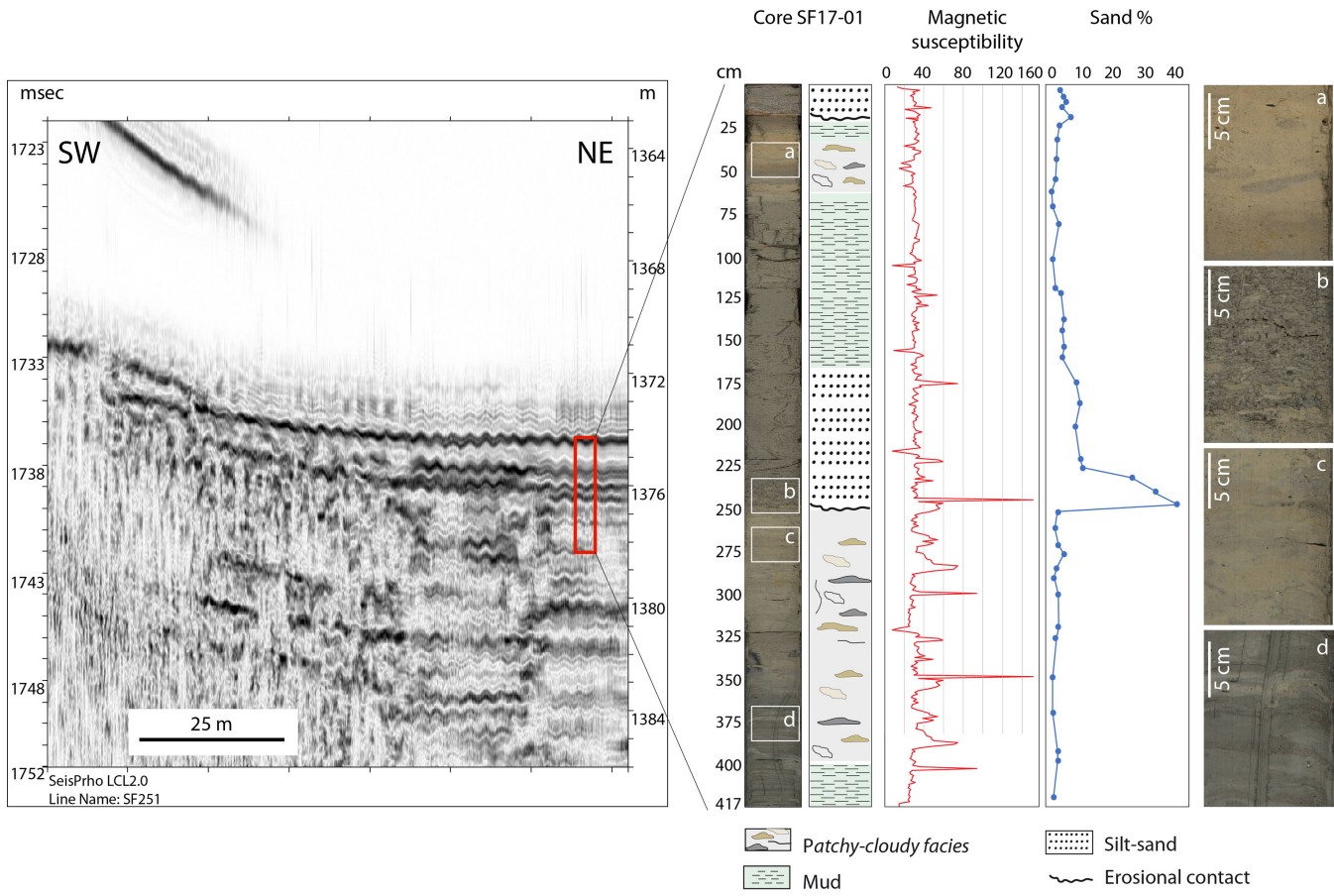

**Figure 9.** Correlation between the 251 chirp profile (left) and the SF17-01 gravity core (right). See locations in Fig. 7. Left: 251 chirp profile collected during coring operations. Right: photograph, lithological log, high-resolution magnetic susceptibility, and sand content of the SF17-01 gravity core. Magnetic susceptibility is rather constant throughout the core with the exception of some peaks in the lower part of the core, where the patchy-cloudy facies (*sensu* Staffini et al., 1993) is present. A peak in magnetic susceptibility marks the abrupt increase in sand content at the base of the resedimented deposit between 75 and 250 cm. a, b, c, and d represent close-ups of different core units: a, c, and d are patchy/cloudy facies whereas b is the base of the resedimented unit.

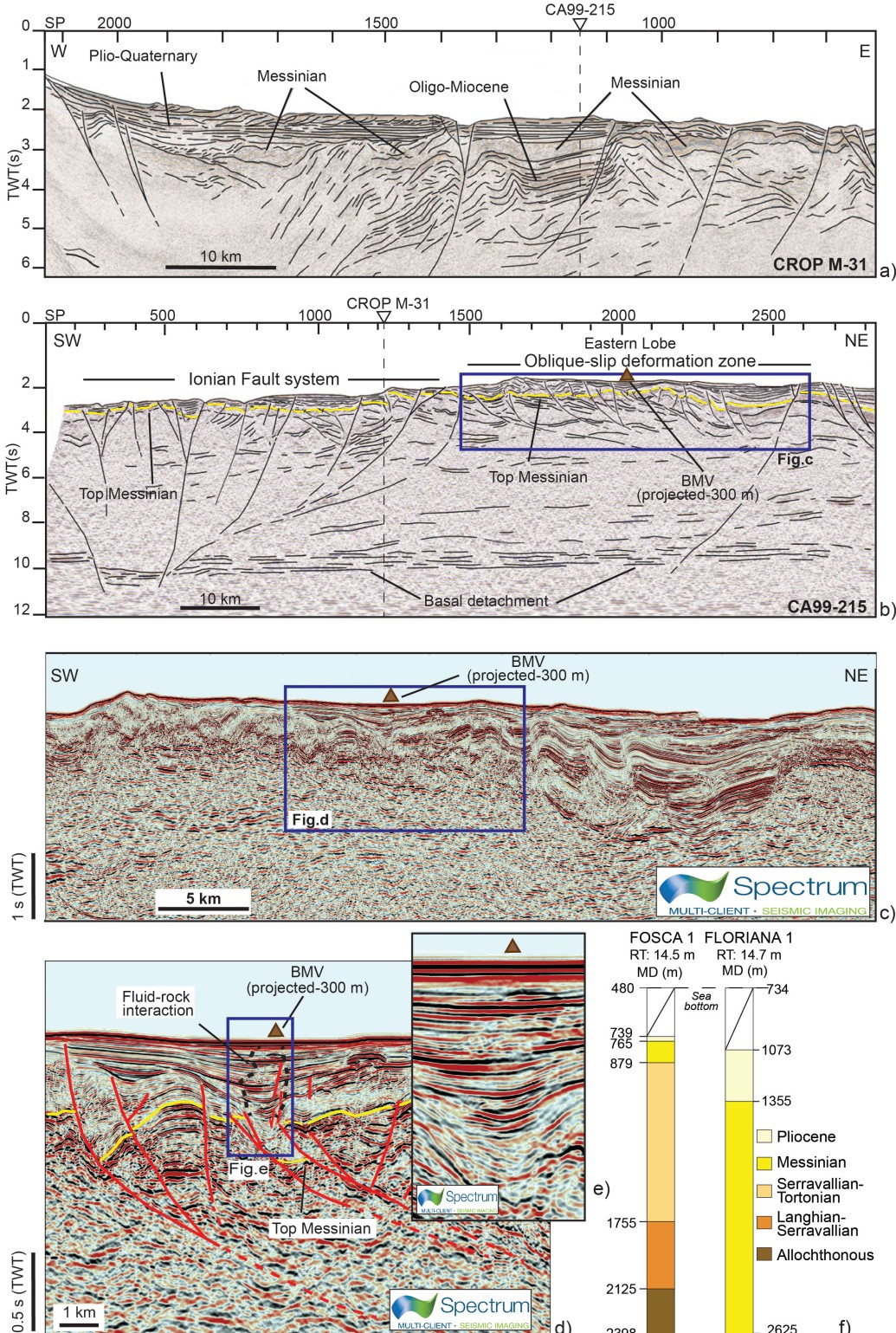

**Figure 10.** (Caption next page.)

**Figure 10.** (a) CROP M-31 seismic reflection profiles (track in Fig. 1) in the version interpreted by Polonia et al. (2016), showing the complex structural setting and the main interpreted deposits. The CROP M-31 profile crosses the CA99-215 profile at s.p. 1150. (b) CA99-215 seismic reflection profiles (track in Fig. 1) in the version interpreted by Polonia et al. (2016). The top of the Messinian deposits is reported in yellow. The CA99-215 profile crosses the CROP M-31 profile at s.p. 1210. (c) Portion of the CA99-215 profile reprocessed by Spectrum Geo Ltd (http://www.spectrumgeo.com/). The BMV is projected from a distance of $\simeq$300 m. (d) and (e) close up views of a portion of the reprocessed CA99-215 profile where the BMV is located. See Fig. S3 for a non-interpreted high-resolution version of Fig. 10(e). Note that the Messinian and Pliocene deposits are faulted by normal and thrust faults. The area just below the projection of the BMV is characterized by a chaotic seismic reflection signal as is typical beneath many mud volcanoes for the fluid-rock interaction. (f) Stratigraphic syntheses of the Fosca 1 and Floriana 1 wells (see location in Fig. 1). RT: Rotary Table. MD: Measure Depth. Raw well data are from the Videpi free database (available only at http://unmig.sviluppoeconomico.gov.it/videpi/videpi.asp).

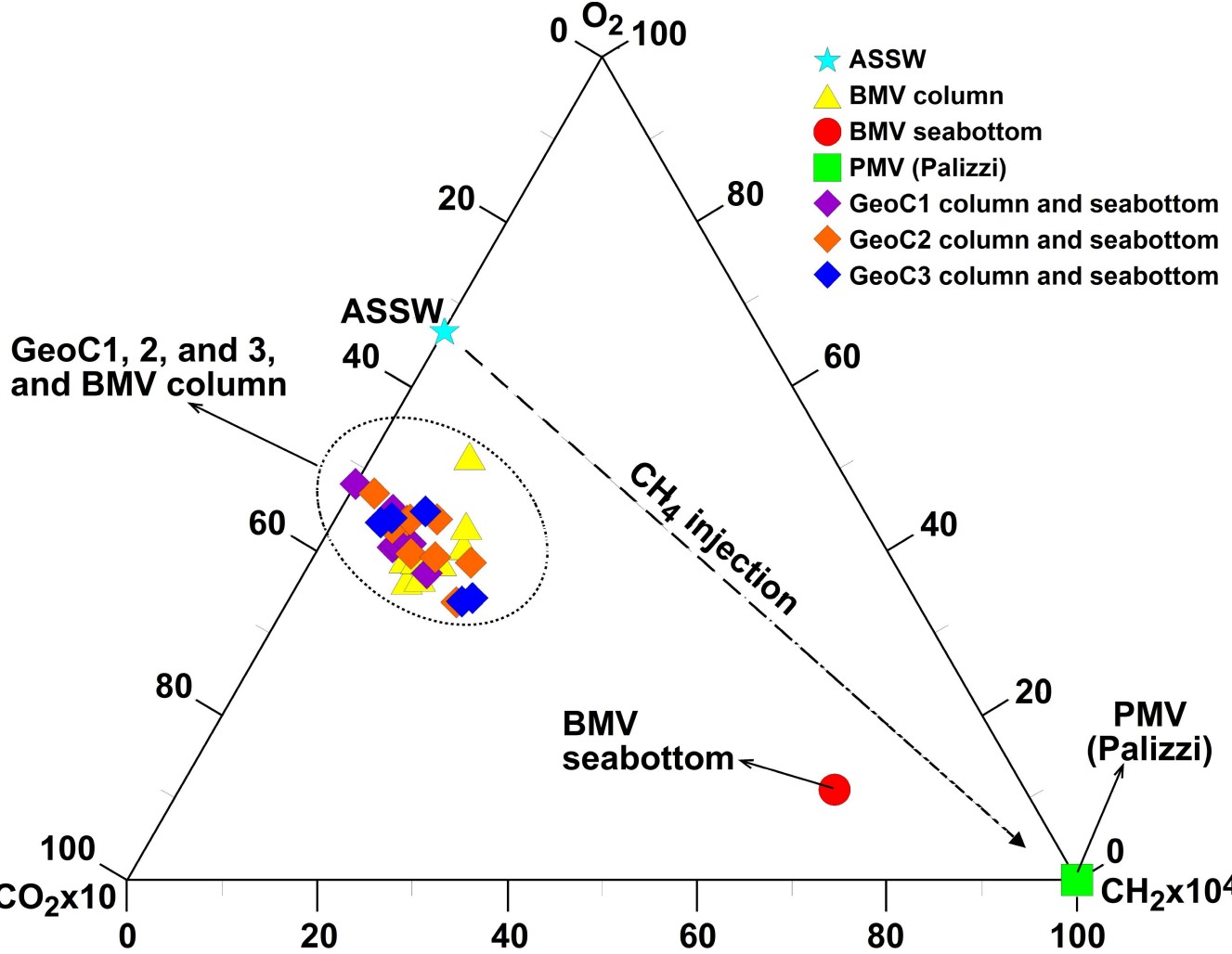

**Figure 11.** $CH_4$-$O_2$-$CO_2$ diagram. The ternary plot shows relative distribution of dissolved gases for the collected samples. Dissolved gases composition coming from the sampling site above the BMV are compared with the local air-saturated sea water (ASSW). The figure shows typical endogenic components ($CO_2$ and $CH_4$) versus the atmospheric component (here represented by $O_2$).

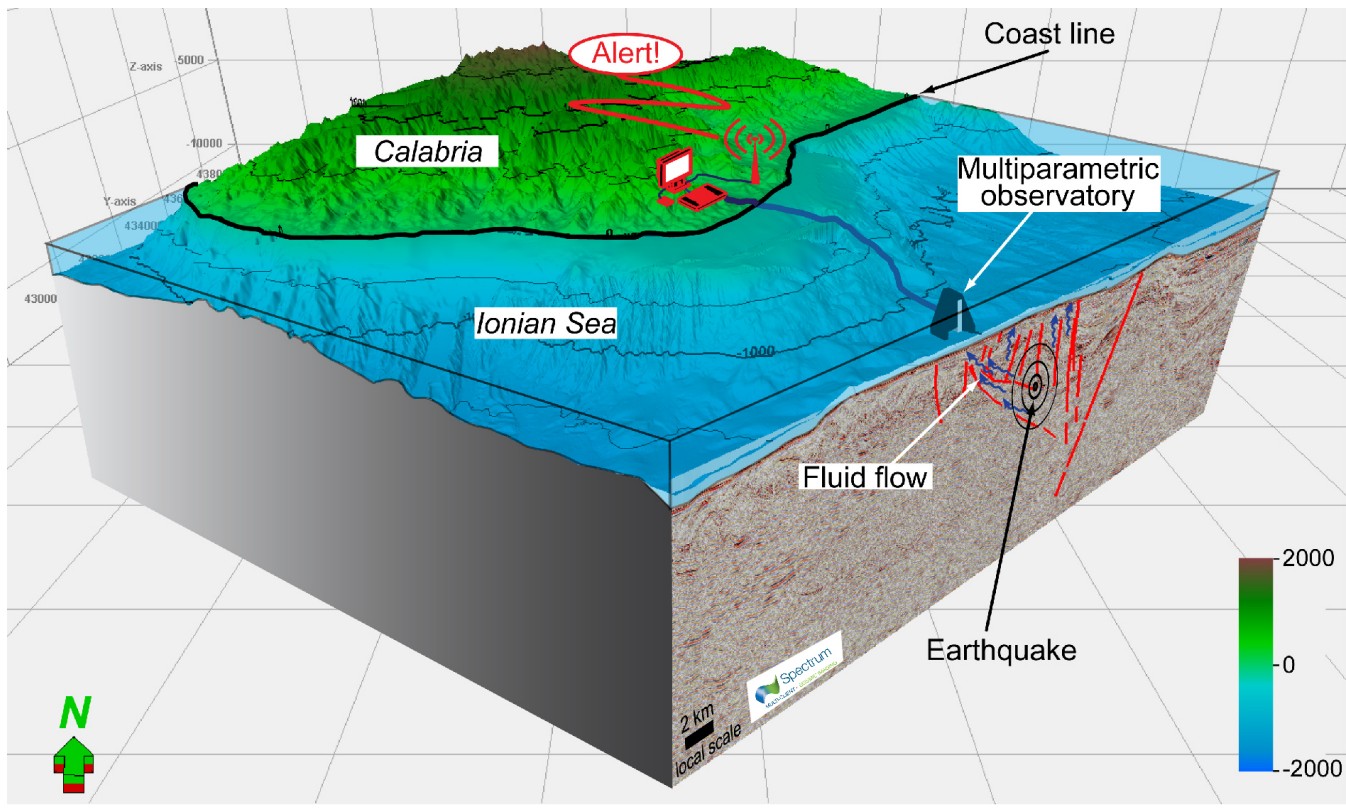

**Figure 12.** Conceptual scheme showing a future multiparametric cabled station on the BMV to geochemically tracking the seismic cycle of the underlying active faults and so to contribute to the mitigation of the seismic hazard.

| Sample | date | depth(m) | Latitude | Longitude | T°C | pH | Na(meq/L) | K(meq/L) | Mg(meq/L) | Ca(meq/L) | Cl(meq/L) | Br(meq/L) | SO$_4$(meq/L) | HCO$_3$(meq/L) |
|---|---|---|---|---|---|---|---|---|---|---|---|---|---|---|
| GeoC1 | 05.17.2017 | -1043 | 37°57'24.6" | 15°28'29.4" | 13.80 | 8.16 | 553 | 12.7 | 125 | 24.9 | 644 | 0.930 | 67.1 | 2.68 |
| GeoC2 | 05.16.2017 | -1649 | 37°43'10.8" | 15°26'40.8" | 13.79 | 8.37 | 552 | 12.7 | 125 | 25.2 | 637 | 0.999 | 66.5 | 2.68 |
| GeoC3 | 05.16.2017 | -2029 | 37°28'10.8" | 15°23'01.8" | 13.82 | 8.10 | 554 | 12.8 | 126 | 25.6 | 641 | bdl | 66.1 | 2.70 |
| BMV | 05.18.2017 | -1337 | 37°52'38.1" | 16°16'50.1" | 13.76 | 8.10 | 553 | 12.6 | 125 | 25.3 | 643 | 1.38 | 66.0 | 2.70 |
|  |  | -1000 |  |  | 13.79 | 8.09 | 551 | 12.6 | 125 | 25.5 | 633 | 1.04 | 63.5 | 2.72 |
|  |  | -500 |  |  | 14.25 | 8.16 | 555 | 12.6 | 125 | 25.5 | 645 | 1.04 | 65.8 | 2.70 |
|  |  | -200 |  |  | 14.88 | 8.06 | 555 | 12.8 | 126 | 25.5 | 643 | 1.75 | 65.9 | 2.66 |
|  |  | -50 |  |  | 15.91 | 8.07 | 552 | 12.7 | 125 | 25.5 | 638 | 1.09 | 66.0 | 2.71 |
| Palizzi PMV | 10.06.2003 | onland | 37°57'51.02" | 15°59'28.43" | na | na | 15.06 | 0.09 | 0.01 | 0.09 | 7.23 | bdl | 3.60 | na |

Table 1: Chemical composition of the sea water from the studied localities in the Ionian Sea (GeoC1, 2, and 3, and BMV). The chemical composition of the fluids sampled in a well adjacent to the Palizzi mud volcano (PMV) in the Calabria main land is also reported.

| Site | depth(m) | He(cc/L) | $O_2$(cc/L) | $N_2$(cc/L) | $CH_4$(meq/L) | $CO_2$(cc/L) | R/Ra | He/Ne |
|---|---|---|---|---|---|---|---|---|
| GeoC1 | 1000 | 7.14E-05 | 3.95 | 9.31 | 1.37E-04 | 5.28E-01 | 0.94 | 0.30 |
| | 900 | 6.11E-05 | 3.97 | 9.64 | 7.62E-05 | 5.10E-01 | 1.07 | 0.27 |
| | 800 | 7.74E-05 | 3.75 | 8.85 | 7.23E-05 | 4.73E-01 | 0.94 | 0.27 |
| | 700 | 6.86E-05 | 3.95 | 9.30 | 9.14E-05 | 4.80E-01 | 1.01 | 0.24 |
| | 500 | 5.51E-05 | 4.16 | 9.45 | 6.86E-05 | 4.73E-01 | 0.93 | 0.27 |
| | 300 | 5.56E-05 | 4.10 | 9.40 | 8.38E-05 | 4.95E-01 | 1.00 | 0.24 |
| | 200 | 6.35E-05 | 4.58 | 9.73 | 5.61E-05 | 5.04E-01 | 0.88 | 0.27 |
| | 100 | 6.61E-05 | 4.41 | 8.90 | bdl | 4.75E-01 | 0.93 | 0.30 |
| GeoC2 | 1649 | 6.01E-05 | 3.93 | 9.35 | 2.07E-04 | 5.64E-01 | 0.77 | 0.28 |
| | 1500 | 5.84E-05 | 4.23 | 8.98 | 1.07E-04 | 5.37E-01 | 0.92 | 0.26 |
| | 1300 | na | 3.77 | 8.98 | 1.24E-04 | 4.62E-01 | na | na |
| | 1000 | na | 4.24 | 9.62 | 2.30E-05 | 4.56E-01 | na | na |
| | 700 | na | 4.04 | 9.99 | 1.77E-04 | 4.67E-01 | na | na |
| | 600 | na | 3.94 | 9.21 | 6.90E-05 | 4.39E-01 | na | na |
| | 500 | na | 3.79 | 8.79 | 5.82E-05 | 4.50E-01 | na | na |
| | 300 | na | 4.24 | 9.48 | 7.67E-05 | 4.67E-01 | na | na |
| | 100 | na | 4.38 | 9.10 | 1.07E-04 | 4.54E-01 | na | na |
| GeoC3 | 2029 | 8.73E-05 | 3.98 | 9.20 | 5.33E-05 | 4.54E-01 | 0.766 | 0.422 |
| | 1500 | 6.75E-05 | 3.51 | 9.85 | 1.90E-04 | 4.97E-01 | 0.950 | 0.290 |
| | 1000 | 7.17E-05 | 3.99 | 9.34 | 4.57E-05 | 4.74E-01 | 0.927 | 0.366 |
| | 500 | 7.13E-05 | 3.61 | 9.33 | 2.02E-04 | 4.90E-01 | 0.976 | 0.289 |
| | 100 | na | 4.54 | 9.30 | 9.14E-05 | 4.68E-01 | na | na |
| BMV | 1337 | 8.27E-05 | 3.99 | 9.12 | 2.51E-03 | 7.31E-01 | 0.73 | 0.398 |
| | 1200 | na | 3.94 | 9.33 | 1.22E-04 | 5.67E-01 | nd | na |
| | 1000 | 7.70E-05 | 4.08 | 9.90 | 1.45E-04 | 5.03E-01 | 0.92 | 0.256 |
| | 800 | na | 3.97 | 9.37 | 9.90E-05 | 5.22E-01 | na | na |
| | 700 | na | 3.76 | 8.88 | 1.07E-04 | 4.85E-01 | na | na |
| | 500 | na | 3.87 | 8.60 | 1.45E-04 | 4.23E-01 | na | na |
| | 300 | na | 3.97 | 9.75 | 1.35E-04 | 5.50E-01 | na | na |
| | 200 | na | 4.61 | 9.58 | 1.52E-04 | 4.59E-01 | na | na |
| | 100 | na | 6.47 | 1.59 | 1.29E-04 | 4.80E-01 | na | na |
| PMV | onland | 2.62E-03 | bdl | 22.28 | 6.87E+00 | 1.23E-01 | 1.60E-01 | 17 |
| ASSW | | 4.80E-05 | 4.80 | 9.60 | 1.00E-06 | 2.40E-01 | 1 | 0.267 |

Table 2: Dissolved gas composition and helium isotopic ratio of the sea water from the studied localities in the Ionian Sea (GeoC1, 2, and 3, and BMV). Data for the fluids sampled in a well adjacent to the Palizzi mud volcano (PMV) in the Calabria main land are also reported.

# The Bortoluzzi Mud Volcano (Ionian Sea, Italy) and its potential for tracking the seismic cycle of active faults

Marco Cuffaro[1], Andrea Billi[1], Sabina Bigi[2], Alessandro Bosman[1], Cinzia G. Caruso[3], Alessia Conti[2], Andrea Corbo[3], Antonio Costanza[4], Giuseppe D'Anna[4], Carlo Doglioni[2,5], Paolo Esestime[6], Gioacchino Fertitta[4], Luca Gasperini[6], Francesco Italiano[3], Gianluca Lazzaro[3], Marco Ligi[6], Manfredi Longo[3], Eleonora Martorelli[1], Lorenzo Petracchini[1], Patrizio Petricca[2], Alina Polonia[6], and Tiziana Sgroi[5]

[1]Istituto di Geologia Ambientale e Geoingegneria, CNR, Rome, Italy
[2]Dipartimento di Scienze della Terra, Sapienza Universitá di Roma, Rome, Italy
[3]Istituto Nazionale di Geofisica e Vulcanologia, Palermo, Italy
[4]Istituto Nazionale di Geofisica e Vulcanologia, Gibilmanna, Italy
[5]Istituto Nazionale di Geofisica e Vulcanologia, Rome, Italy
[6]Spectrum Geo Ltd, Woking, United Kingdom
[7]Consiglio Nazionale delle Ricerche, ISMAR, Bologna, Italy

**Correspondence:** Andrea Billi (andrea.billi@cnr.it)

**Abstract.** The Ionian Sea in southern Italy is at the ~~center~~ centre of active interaction and convergence between the Eurasian and African-Adriatic plates in the Mediterranean. This area is seismically active with instrumentally/historically-recorded Mw > 7.0 earthquakes and it is affected by recently-discovered long strike-slip faults across the active Calabrian accretionary wedge. Many mud volcanoes occur on top of the wedge. A recently-discovered one (here named Bortoluzzi Mud Volcano, BMV) was surveyed during the Seismofaults 2017 cruise (May 2017). Bathymetric-backscatter surveys, seismic reflection profiles, geochemical and earthquake data as well as a gravity core are here used to geologically, geochemically, and geophysically characterize this structure. The BMV is a circular feature $\simeq$22 m high and $\simeq$1100 m in diameter with steep slopes (up to a dip of 22°). It sits atop the Calabrian accretionary wedge and a system of flower-like oblique-slip faults that are probably seismically active as demonstrated by earthquake hypocentral and focal data. Geochemistry of water samples from the seawater column on top of the BMV shows a significant contamination of the bottom waters from saline (evaporite-type) $CH_4$-dominated crustal-derived fluids similar to the fluids collected from a mud volcano located in the Calabria main land over the same accretionary wedge. These results attest for the occurrence of ~~an open crustal conduit~~ open crustal pathways for fluids through the BMV down to at least the Messinian evaporites at about -3000 m. This evidence is also substantiated by Helium isotope ratios and by comparison and contrast with different geochemical data from three sea water columns located ~~elsewhere~~ over other active faults in the Ionian Sea ~~. Conclusions are drawn on the origin of the BMV and on the potential of this type of structures~~ area. One conclusion is that the BMV may be useful for tracking the seismic cycle of active faults through geochemical monitoring. Due to the widespread diffusion of mud volcanoes in seismically active settings, this study ~~may contribute~~ contributes to indicate a ~~potential and feasible~~ future path for the use of ~~these ubiquitous structures in favor of the~~ mud volcanoes in the monitoring and mitigation of natural hazards.

## 1 Introduction

Mud volcanoes are ubiquitous structures on the Earth's surface both in marine and in continental settings and form for a variety of causes, most of which amenable to clay- and fluid-rich subsurface levels, where overpressures contribute to drive the ascent of mixed and liquefied clay, water, and gases (e.g., $CO_2$, $CH_4$, and N), with consequent formation of the conical or conical-trunk landform ~~(Milkov, 2000; Deville et al., 2003; Etiope and Milkov, 2004; Planke et al., 2003; León et al., 2007; Ceramicola et al., 201~~ Mud volcanoes can be therefore taken as markers of overpressure-related geofluid circulation, efflux, and ascent (Bertoni and Cartwright, 2015; Kirkham et al., 2017).

Many mud volcanoes occur on active accretionary~~prisms (Higgins, 1974; Cita, 1981; Barber, 1981; Dimitrov, 2002; Kopf, 2002; Rabaut~~ prisms (Cita, 1981; Kopf, 2002; Panieri et al., 2013; Ceramicola et al., 2014), where their degassing activity is at least in part connected with active compression and ~~subsequent~~ associated faulting and fracturing ~~(Camerlenghi et al., 1995; Robertson and Kopf, 1998;~~ It is also noteworthy that, in these active tectonic settings, the longevity of mud volcanoes may even exceed 1 Ma ~~(Cita et al., 1989; Camerler~~ ~~and they - all~~ (Cita et al., 1989; Camerlenghi et al., 1992, 1995; Robertson, 1996; Kopf et al., 1998; Praeg et al., 2009; Somoza et al., 2012 All together the known submarine mud volcanoes ~~-~~release about 27 Mt methane $a^{-1}$ ~~(plus other gases; Milkov et al., 2003; Etiope and Mill~~

The Ionian Sea between eastern Sicily and southern Calabria (Italy; Fig. 1) hosts an active accretionary prism ~~(the Calabrian Arc; Minelli~~ where both prism-parallel thrusts and across-prism strike- and oblique-slip faults are active~~(Polonia et al., 2012; Totaro et al., 2013; Doglion~~ Some of these faults ~~could even be~~ are the source structures of some among the most devastating earthquakes and tsunamis of southern Italy and the entire Mediterranean (Polonia et al., 2012; Totaro et al., 2013; Doglioni et al., 2012). Among other devastating events in the Ionian Sea large area, we recall the 1908 (Mw 7.2), 1783 (Mw 6.9), 1693 (Mw 7.4), 1169 (Mw 6.6), and 362 (Mw 6.6) AD earthquakes and tsunamis, which caused damage, devastation, and death (more than 80,000 deaths in 1908) in eastern Sicily and southern Calabria (Locati et al., 2016). Although such events have been studied by many authors for the great impact they produced, their origin (zone and generation mechanism) is still heavily debated both for earthquakes and tsunamis. ~~In other words, the~~ The faults that generated such earthquakes are not yet known as it is unknown whether the associated tsunamis were generated directly by earthquakes (i.e., fault dislocation of seabed) or indirectly by seismically-triggered submarine slides ~~(Valensise and Pantosti, 1992; Billi et al., 2008, 2010; Argnani et al., 2009; Polonia et al., 2016)~~(Valensise and Pantosti, 1992; I

To better define the seismic behavior of active faults in the Ionian Sea, during the Seismofaults 2017 and 2018 marine surveys (May 2017 and 2018, respectively; www.seismofaults.it), we deployed eleven seabottom seismometers (OBS/H) and two seabottom multiparameric geochemical-geophysical sensors. At the time of writing of this paper, some of these devices are still operating on the seabottom, while data from the recovered devices are being processed and are not therefore available for publication. One of the aims of this project (Seismofaults) and related surveys was to contribute to the advancement of the science of earthquake forecasting. For this reason, we deployed the multiparameric sensors (i.e., to detect anomalous degassing prior to earthquakes), collected sea water column samples along and far away from (active) faults, and searched for new fluid venting structures such as mud volcanoes. ~~We believe, in fact, that the~~ The activity of some of these latter structures

may be partly connected with (seismic) faulting (Martinelli et al., 1995; Polonia et al., 2011; Capozzi et al., 2012; Panieri et al., 2013; Ceramicola et al., 2014). For instance, Martinelli and Ferrari (1991) and Martinelli et al. (1995) recorded radon anomalies ($^{222}$Rn) in the liquid phase of some mud volcanoes from the northern Apennines (Italy) before and during low magnitude earthquakes ($\leq$ 2.5) occurred nearby in 1986 and 1987. Again in the northern Apennines, **?** recorded potential precursory seismological signals ten minutes before an Mw 4.7 earthquake in a monitored mud volcanic field. Discovering and monitoring such structures may therefore be the key, in the future, to better understand the seismic cycles and possible accompanying or precursory processes and phenomena such as anomalous fluid discharge from mud volcanoes ~~(Martinelli and Ferrari, 1991; Martinelli et al., 1995; Milkov, 2000; Dimitrov, 2002; Kopf, 2002; Mazzini and Etiope, 2017). Noteworthy~~ (Martinelli and Ferrari, 1991; Martinelli et al., 1995; Kopf, 2002; Mazzini and Etiope, 2017). A link between the frequency of eruption of some mud volcanoes and the seismic activity has been often documented (**??**). A nice example is the 5 km long mud flow emitted by the Kandewari mud volcano (Pakistan) following the 2001 Gujarat-Bhuj Mw 7.7 earthquake (India-Pakistan **?**). It is also true, however, that many active mud volcanoes do not seem to manifestly react to nearby earthquakes. Our aim in this article is not finding a link between earthquakes and the activity of mud volcanoes but to understand whether even a mud volcano that seems inactive or poorly-active can constitute an open and preferential pathway for possible geochemical precursors of earthquakes. Whether these precursors truly manifest before an earthquake is a different task that has to be accomplished with a specific monitoring as we will explain in the discussion section. Our aim is nonetheless relevant for many geoscientists as noteworthy examples of earthquake geochemical precursors are documented in many previous works ~~(Wakita et al., 1988; Igarashi et al., 1995; Claesson et al., 2004; Huang and Ding, 2012; Inan et al., 2012; Skelton et al., 2014; Sano et al.,~~ are very promising for the science of earthquake forecasting.

Following the above-mentioned reasons and aims, during the Seismofaults 2017 scientific cruise in the Ionian Sea (Fig. 1), we surveyed a ~~large~~ mud volcano recently ~~signaled~~ signalled on the Calabrian accretionary prism by Gutscher et al. (2017) and Loher et al. (2018) but never analyzed in detail (Supplement, Movies S1 and S2). We performed a multidisciplinary analysis on this feature, including high-resolution multibeam bathymetry, high frequency chirp-sonar profiling, physical and chemical analyses of the water column above the mud volcano, and gravity coring on one side of this structure. Based on these new data integrated with previous ones, we interpret the recently-discovered mud volcano within the framework of the (seismically) active accretionary prism of Calabria and eventually provide some implications for future marine researches and monitoring of the seismic cycle in marine areas. ~~Our~~ As mentioned above, our main aim is not only to characterize the mud volcano but also to provide a contribution toward a potential and feasible future path for the use of these ubiquitous structures in ~~favor of the~~ favour of the monitoring and mitigation of natural hazards. As the mud volcano studied in this paper had already been observed years ago by our colleague Giovanni Bortoluzzi in one of his numerous marine surveys, we name this structure ~~the~~ Bortoluzzi Mud Volcano (BMV) to recall his memory and fruitful life spent sailing for the science over the Mediterranean Sea and the oceans. The BMV was selected for this study mainly for its proximity to the coast, particularly to seismically hazardous regions (southern Calabria and eastern Sicily; Fig. 1), and for its location on top of a seismically active fault systems, as we will show below.

## 2 Geological Setting

The BMV is located on top of the Calabrian Arc (Ionian Sea; Fig. 1) that has developed along the Africa-Eurasia plate boundary in the center of the Mediterranean Sea. The arc belongs to the eastward retreating Apennines subduction system connecting the NW-SE-trending Apennines with the E-W-oriented Maghrebian thrust-fold belt (Patacca and Scandone, 2004). In particular, the Calabrian Arc has developed on top of a NW-dipping subduction system with the Ionian lithosphere sinking toward NW beneath the Tyrrhenian lithosphere. This subduction system is also characterized by an active volcanic arc (the Aeolian Islands in the southeastern Tyrrhenian) and a well-defined Wadati–Benioff zone (Wortel and Spakman, 2000), with earthquakes descending to nearly 500 km depth beneath the Aeolian Islands on the Tyrrhenian lithosphere. The Africa-Eurasia convergence is active in this area at a very slow rate (5 mm/yr or even less), as documented by recent GPS studies (Serpelloni et al., 2007; Billi et al., 2011; Palano et al., 2012, 2015). The external part of the Arc is represented by a 300 km wide accretionary complex bounded to the south by the outer deformation front and laterally by two major structural discontinuities: the Malta escarpment to the southwest and the Apulia escarpment to the northeast. To the northwest (i.e., toward the Calabria region), the accretionary wedge is significantly thickened (Cernobori et al., 1996), whereas it tapers away toward the southeast in the Mediterranean Sea. The wedge is segmented along strike in different structural domains by NW-trending structural discontinuities. The compartments are characterized by different rheologies and deformation styles (Polonia et al., 2011). In particular, three main morpho-structural domains can be identified in the subduction complex: (i) the post-Messinian accretionary wedge; (ii) the pre-Messinian accretionary wedge, and (iii) the inner plateau. Structural styles and seafloor morphologies vary in these four compartments in correlation with different tectonic processes that include frontal accretion, out-of-sequence thrusting, underplating, and complex faulting.

In the Calabrian Arc, the very low tapered (taper angle about 1.5°) outermost accretionary wedge is a salt-bearing complex developed during and after the Messinian salinity crisis (Fig. 1). Frontal accretion of the arc is active in this southern region over a shallow basal detachment located within or at the base of the Messinian evaporites. The inner wedge, which is located toward NW at the rear of the post-Messinian accretionary complex, consists of pre-Messinian clastic sediments. The basal thrust of the inner wedge is located on top of the Cretaceous sediments and/or at the transition with the basement. Moreover, the inner wedge is bounded toward NW by an inner deformation front, representing the transition between the strongly deformed accretionary wedge to the SE and a less deformed inner plateau to the NW. This plateau is however dissected by long normal and strike-slip fault zones mostly striking NW-SE. The inner plateau is characterized by chaotic units (Rossi and Sartori, 1981) and many mud volcanoes (Praeg et al., 2009; Ceramicola et al., 2014; Gutscher et al., 2017; Loher et al., 2018) including the BMV that is studied in this work.

As mentioned above, the Calabrian accretionary wedge is cut across (NW-SE) by a set of long faults or deformation zones (Fig. 1). Most of these tectonic features are active and characterized by strike-slip to transtensional or transpressional (oblique) kinematics. At a broad scale, the sense of horizontal shear along these zone is usually right-lateral ~~(Argnani and Bonazzi, 2005; Minelli and~~ Below, we show that the BMV is located on top of one of these oblique-slip features crossing the Calabrian accretionary wedge.

# 3 Methods and ~~Results~~Data

## 3.1 Rationale

~~The~~ As mentioned above, the main target of this work is the BMV located at 37° 53' 01" N and 16° 16' 50" E in the Ionian Sea (Fig. 1). During the Seismofaults 2017 cruise, in the BMV area, we acquired high-resolution multibeam bathymetry (bathymetry, backscatter and water column), chirp seismic profiles, physical and geochemical data of the sea water column, and a sediment gravity core. These data are described below together with the related methods of acquisition and with a re-processed and unpublished seismic reflection profile whose track is only about ~~200~~ 300 m far away from the BMV. We integrated the above-mentioned geological-geophysical-geochemical evidence with previously-published data and with data from public databases (e.g., earthquake data). All ~~these data are presented in the next sub-sections, where the description of each type of data and results is anticipated by a synthetic description of the acquisition and processing methods together with an account of the data source. Some data (i.e., the single- and multi-channels seismic reflection profiles as well as the gravity core) are also briefly interpreted to make the subsequent discussion section easier to read and understand. The discussion section is indeed devoted to a synthesis of results and to issues more general than the interpretation of the single dataset. All~~ used data are reported in the figures and tables of this paper and related supplementary material. Moreover, with this paper, we release 4970 km$^2$ of newly-acquired (during the Seismofaults 2017 cruise) high-resolution bathymetric data in the Ionian Sea including the BMV area (Fig. S1).

## 3.2 Earthquakes

### 3.2.1 ~~Method and Data~~

To understand whether the study area (BMV) is seismically active, we ~~collected earthquake data from a public database and from previous articles. In particular, we collected crustal~~ analysed the crustal seismicity (depth $\leq$ 40 km) ~~earthquake hypocentral-epicentral data (Table S1) from the public database of INGV Centro Nazionale Terremoti~~ occurred in the study area and recorded by Osservatorio Nazionale Terremoti of the Istituto Nazionale di Geofisica e Vulcanologia (http://cnt.rm.ingv.it/) ~~relocated in this study by~~ from 1985 to 2017. We collected the arrival time data from the Italian Seismic Catalogue (ISC 1985-2002, available the Italian Seismic Bulletin (ISB 2003-2012, available online at http://bollettinosismico.rm.ingv.it/). Then, one of us (T.S.) manually picked the most recent earthquakes (2013-2017) following the same procedure adopted by the analysts of ISC 1985-2002 and ISB 2003-2012. Combining the picked arrival time data with the arrival time data from the analyzed bulletins, one of us relocated all earthquakes (Table S1). Moreover, to understand the kinematics of the seismic faulting, we collected earthquake focal mechanisms (Table S2) from the European-Mediterranean Regional Centroid-Moment Tensors (RCMT) catalog (http://rcmt2.bo.ingv.it/) and from previous papers by Orecchio et al. (2014) and Polonia et al. (2016). Earthquake data (Tables S1 and S2) are shown in map view in Figs. 1 and 2(a) and in vertical cross-sectional view in Fig. 2(b).

### 3.2.1 ~~Results~~

### 3.3 Bathymetry and Geomorphology

~~Earthquake epicenters and focal mechanisms~~ We carried out the high resolution multibeam bathymetric survey (Figs. 3,4 and S1) using a multibeam Teledyne Reson SeaBat 7160 (41-47 kHz) echosounder characterized by footprint size of 1°×1°. We identified precise positioning through differential GPS (accuracy ±0.5 m), while we derived sound velocity profiles from multiple Conductivity-Temperature-Depth (CTD) casts (Seabird 911plus) to ray trace the acoustic wave along the water column. We processed multibeam data on board using Caris Hips & Sips hydrographic software (Bosman et al., 2015). We used backscatter images and observations of raw data scattering along the water column (Fig. 5) to verify anomalies of amplitude on the seafloor and along the water column. Eventually, we compared some geometrical features of the BMV with the same geometrical features from previously published databases of mud volcanoes (Kioka and Ashi, 2015; Kirkham et al., 2017, Fig. 6).

### 3.4 Single-channel Chirp Seismic Reflection Profiles

During the Seismofaults 2017 survey, to define the local geological setting and a high resolution seismic stratigraphy of the BMV and surrounding area, we acquired a set of chirp seismic profiles (Figs. 7b-d, 8, and S2) using a frequency-modulated source operating in the frequency range of 2-7 kHz (Benthos Chirp III) and recorded with a 0.5-0.8 s sweep length. Maximum sub-bottom penetration is up to about 40 ms (TWT; corresponding to c. 30 m if considering a seismic velocity of 1500 m/s) and vertical resolution is about 0.7 ms (TWT; corresponding to c. 0.5 m if considering a seismic velocity of 1500 m/s). We processed the chirp profiles using the GeoSuite All Works software, applying Time Varied Gain, and the open source software Seisprho (**?**). Thicknesses and depths of seismic profiles are described in two-way travel time (TWT; Figs. 7b-d and 8), with a seismic velocity of 1500 m/s being used to convert two-way travel time into depth. Seismic data interpretation was carried out through the Kingdom Suite software also integrating the bathymetric data and a gravity core (Fig. 9).

### 3.5 Seabottom Gravity Core

We collected the SF17-01 core (Fig. 9; see core location in Fig. 7a) during the Seismofaults 2017 survey using a 1.2 ton gravity corer with coring pipes 6 m long. Location of the core was controlled through the aforementioned onboard differential GPS system (accuracy ±0.5 m). We analyzed the core sections through a multi-proxy approach involving high-resolution digital photographs and determination of physical properties and sand content as deduced through the weight of selected samples. We also acquired high resolution magnetic susceptibility (MS) through a core log system (Bartinghton model MS2, 100 mm loop sensor) with a sampling interval of 1.0 cm.

### 3.6 Multi-Channel Seismic Reflection Profiles

In this paper, we present two multiple channel seismic reflection profiles and related interpretations (Fig. 10; see tracks in Fig. 1). The first profile (Fig. 10a) is the CROP M-31 (Scrocca et al., 2003) in the version interpreted by Polonia et al. (2016). The second profile is the CA99-215 (Fig. 10b) both in the version interpreted by Polonia et al. (2016) and in a new version (Figs. 10c and 10d; see also Fig. S2) deriving from a recent reprocessing kindly provided by Spectrum Geo Ltd (http://www.spectrumgeo.com/) within the framework of a confidentiality agreement between Spectrum Geo and Sapienza University of Rome.

The CROP M-31 was acquired in 1994 in the framework of the CROP (CROsta Profonda) Project (**?**). The streamer used was analogic, 4500 m long, composed by 180 channels with a group interval of 25 m, towed at a depth of 12-14 m. The total source volume was equal to 4882 in$^3$. The recording time length was 20 s, shot interval was 50 m, and coverage was 45 (**?**). In general, the applied processing sequences consisted in quality control, gain recovery, trace sum (optional), deconvolution, CDP re-ordering, velocity analysis (CVS), NMO correction, muting, multiple reflection attenuation, array simulation (optional), weighted stack, F-K filtering (optional), horizontal mixing (optional), time-variant filter, and equalization (**?**).

The CA99-215 profile was acquired in 1999 and reprocessed in 2001. Source volume was 3410 in$^3$, located at 6 m depth, with a shot interval of 25 m. The used streamer had a length of 6000 m, towed at 8 m depth, with a group interval of 12.5 m, and a recording length of 8 s. Reprocessing mainly consisted in a standard PSTM (pre-stack time migration) processing sequence, resulted in an improved data quality.

Stratigraphic syntheses from the Fosca 1 and Floriana 1 wells (original well data are from the Videpi public database available online at http://unmig.sviluppoeconomico.gov.it/videpi/videpi.asp) are also presented (Fig. 10f) to help interpreting the seismic profiles (see well location in Fig. 1).

### 3.7 Geochemical Features of Four Sea Water Columns

During the Seismofaults 2017 survey, we sampled sea water columns at various depths in four localities of the Ionian Sea (Fig. 1 and Tables 1 and 2)~~show that the area surrounding the BMV is seismically active as it is populated by earthquakes. In particular, the~~: namely, above the BMV and above the GeoC1, GeoC2, and GeoC3 localities that are along or nearby major presumably-active fault zones (Polonia et al., 2012, 2016). We carried out vertical casts by Rosette and Niskin bottles to determine the geochemical features and dissolved gases at the sea bottom and along the water columns (Fig. 11 and Tables 1 and 2). Samples are compared with the local air-saturated sea water (ASSW) used as benchmark.

The samples, specifically collected for the extraction of the whole gas phase for chemical and isotopic analyses, were stored in 240 ml pyrex bottles sealed on board using rubber/teflon septa and purpose-built pliers, and analyzed within two weeks. Details of the sampling methodology are reported in Italiano et al. (2009, 2014). During the cruise, the sampled bottles were stored upside down, keeping the necks with the rubber septa submerged in sea-water until they were transferred to the laboratory. The collected sea-water samples underwent laboratory procedures for both chemical (concentration of dissolved gas species) and isotopic (helium isotopes) determinations. In the laboratory, the dissolved gases were extracted after equilibrium was reached at constant temperature with a host-gas (high-purity argon) injected in the sample bottle (see Italiano et al., 2009, 2014, for furt The chemical analyses of $O_2$, $N_2$, $CH_4$, and $CO_2$ were carried out by gas-chromatography (Agylent 7800B equipped with a double TCD-FID detector) using argon as carrier gas. Typical analytical uncertainties were within $\pm 5\%$. He concentration

was determined by mass spectrometry, during $^3$He/$^4$He analyses. Helium isotope and $^4$He/$^{20}$Ne ratios were carried out on gas fractions extracted following the same procedure described above and purified following the method described in the literature (Hilton, 1996; Sano and Wakita, 1988; Italiano et al., 2001). The isotopic analyses of the purified helium were performed using a static vacuum mass spectrometer (GVI5400TFT) that allows the simultaneous detection of $^3$He and $^4$He ion beams, thereby keeping the $^3$He/$^4$He error of measurement very low. The used analytical method also requires running alternatively one sample and one purified air shot used as internal $^3$He/$^4$He standard. Typical uncertainties in the range of atmospheric He-type samples are within $\pm 1\%$. The $^4$He/$^{20}$Ne ratios were calculated by the relative peak heights measured on the same mass spectrometer.

Water samples of 100 ml were stored in PVC bottles for total alkalinity titration: 50 ml were filtered by a 0.45 $\mu$m filter and acidified by $HNO_3$ 0.1 N for cations (Ca, Mg, Na, and K) determination, whereas the non-acidified samples were collected for anion (Cl, F, and $SO_4$) determination. pH was measured by an electronic device calibrated in situ using buffer solutions. In the laboratory, chemical analyses of the major constituents were carried out by ion-chromatography (Dionex ICS-1100) both on filtered (0.45 mm) acidified ($HNO_3$ Suprapur) water samples (Na, K, Mg, and Ca), as well as on untreated samples (F, Cl, Br, $NO_3$, and $SO_4$). The $HCO_3$ content was measured by standard titration procedures with hydrochloric acid. Typical uncertainties are $\pm 5\%$.

A few years ago (2003), one of us (F.I.) collected and analyzed fluid samples from a shallow well located in the Calabria main land nearby the Palizzi mud volcano (PMV in Figs. 1 and 2, located about 25 km to the WNW of the BMV area). Unfortunately, no documentation exists concerning the PMV except our original geochemical data reported in Tables 1 and 2. The PMV was indeed actively venting geofluids at the time of our sampling (2003) but it was then soon destroyed by human activity and cementation.

# 4   Results

## 4.1   Earthquakes

The earthquakes recorded in the BMV area (i.e. the area shown in Fig. 2a) between 1985 and 2017 at depths $\leq 40$ km are 178 (Mw $\leq$ ~~4.4~~4.5; Table S1). Focal mechanisms are from earthquakes that are $\leq 4.5$ in magnitude and $\leq 30$ km in depth (Table S2). ~~Focal~~ During the considered time window, no larger magnitude earthquakes ($> $ Mw 4.5) are present for this area in the instrumental catalogs. These focal mechanisms are mostly characterized by strike-slip and transtensional faulting along NW-SE-striking planes or along the conjugate planes striking NE-SW.

Fig. 2(b) shows a transect-perpendicular projection (swath profile) of the above-described earthquake data (i.e., hypocenters and focal mechanisms; Tables S1 and S2) along a NE-SW vertical transect through the BMV (see the A-A' transect track in Fig. 2a). Projected seismic events were selected within ~~a distance~~ an arbitrary distance (swath width) of 12 km from the transect track (Fig. 2a). ~~The cross-sectional view of Fig. 2(b) shows that earthquake~~ Earthquake hypocenters are diffuse over the studied transect including the area beneath the BMV (Fig. 2b). A cluster of hypocenters is discernible near the northeastern tip of the transect (Fig. 2b). This same cluster is also visible as a NW-trending cluster of epicenters in the northeastern portion of the study area (i.e., 40 km to the northeast of the BMV; Fig. 2a).

## 4.2 Bathymetry and Geomorphology

### 4.2.1 ~~Method and Data~~

~~We carried out the high resolution multibeam bathymetric survey (Fig. S1) using a multibeam Teledyne Reson SeaBat 7160 (41-47 kHz) echosounder characterized by footprint size of 1°× 1°. We identified precise positioning through differential GPS (accuracy ±0.5 m), while we derived sound velocity profiles from multiple Conductivity-Temperature-Depth (CTD) casts (Seabird 911plus) to ray trace the acoustic wave along the water column. We processed multibeam data on board using Caris Hips & Sips hydrographic software with the following parameters and processing methods: (a) corrections for tidal height variations from the Catania harbor tide gauge (www.mareografico.it); (b) multibeam calibration (patch test) to measure the angular misalignment between the transducers, motion sensor, gyro, and the position latency; (c) statistical and geometrical (angle and distance) filters to remove coherent and incoherent noise in each swath; (d) manual removal of isolated fake soundings; and (e) generation of a high resolution digital marine model, with a resolution varying between 10 m in deep water (down to -1000 m water depth) and 25 m at greater depths (Bosman et al., 2015). In particular, we calibrated the multibeam data through a patch test in an area located at 1600 m water depth close to a morphological high and a sub-linear slope. The multibeam patch test conducted with PDS2000 and verified with Caris H&S provided the following values: Time delay: 0.0; Heading correction: -0.930°; Roll correction: 0.680°; and Pitch correction: -1.020°. We used backscatter images and observations of raw data scattering along the water column to verify anomalies of amplitude on the seafloor and along the water column. However, compatibly with the instrumental resolution and the depths of the BMV, the data did not show any significant anomaly of the amplitude of the signals or significant evidence of fluid escape.~~

### 4.2.1 ~~Results~~

The study area is located on the upper part of the continental slope of the Calabrian-Ionian margin (Figs. 1 and S1) between 120 m and 2000 m water depth. This area is characterized by a complex morphology due to the interaction between tectonically controlled escarpments and several small scale mass-wasting features, including ~~landslide scars and gullies/channels (~~slide scars (Ss), regional scarps (Rs), and gullies (G) or channels (Ch) (Fig. 3). The upper part of the continental slope is characterized by a very steep slope (about 15° ~~) that reaches a maximum of 22° or~~in average dip) that can reach, in places, a maximum dip of even 28° ~~,~~along the main escarpments oriented NE-SW and long up to 35 km (Fig. 3b). At the foot of the upper continental slope, a well-defined flat area of about 26 km$^2$ occurs encompassing the sub-circular morphological high of the BMV (Fig. 3). The flat area is located at 1350 m water depth and bordered by several small ridges that are elongated parallel to the continental slope. The BMV is characterized by a circular shape with a diameter of about 1100 m and a well-defined rim (Figs. 3 and 4). The BMV has an elevation of about 22 m from its base (i.e., the surrounding plain) and steep slopes ~~dipping up to 22°~~ (see the slope profile in Fig. 4a). The high resolution digital elevation model shows some complex structural and morphological features (Fig. 4). These features consist of concentric morphologies ~~with a~~ each with its perimetric topographic rim~~,~~. The concentric morphologies are separated by some moats and encompassed by the outermost topographic rim of the BMV (Fig. 4). Furthermore, on top of the BMV, three minor sub-circular/arcuate bulges are present. These features have a relief of about 3

m with respect to the surrounding seafloor and are separated by small sub-circular moats (Fig. 4b). The outermost topographic rim of the BMV is interrupted in the southern side (Fig. 4b). Low resolution backscatter data indicate low amplitude of acoustic signals whereas the absence of changes/anomalies in the amplitude indicates a clayey/pelitic sedimentary cover (Fig. 5a). ~~In~~

5 ~~addition, consistently with the resolution of the multibeam equipment, multibeam water column data recorded during the Seismofaults 2017 survey does not highlight acoustic backscatter anomalies related to large amount of fluids escaped from the seafloor (Fig. 5b; e.g., Römer et al., 2014).~~

To better understand the morphology of the BMV, also in comparison with other known mud volcanoes on the Earth, we here consider recent compilations of mud volcano morphological data ~~(e.g., Kioka and Ashi, 2015; Kioka et al., 2015)~~. We refer, in particular, to the works by Kirkham et al. (2017) and Kioka and Ashi (2015), where relations between mud volcano height ~~H~~

10 _H_ vs. diameter ~~D~~ _D_, and volume ~~V vs. H/R~~ _V ($= \pi R^2 H/3$)_ vs. _H/R_ ratio (where ~~R~~ _R_ is the radius of the volcano base) are explored, respectively. Thus, the BMV volume $V = 6.9 \times 10^6$ m$^3$ is computed with the previous formula, after estimating its diameter and height through the use of high resolution multibeam bathymetry data ($D = 1100$ m and $H = 22$ m, Fig. 4). We used the ~~mud volcano dataset reported by Kioka and Ashi (2015), selecting 232 mud volcanoes with available mean diameter D and height H~~ same given formula to compute mud volcano volumes from the dataset provided by Kioka and Ashi (2015). However,

some of these catalogued mud volcanoes are reported without diameter. Hence, we selected only the 232 mud volcano having with both the diameter _D_ and the height _H_ (Fig. 6a). Fig. 6 shows ~~H vs. D~~ _H_ vs. _D_ (Figs. 6b and 6c) and ~~V vs. H/R~~ _V_ vs. _H/R_ (Fig. 6d) for the 232 selected mud volcanoes. ~~Volumes~~ When using the relation $V = \pi R^2 H/3$, volumes are calculated following the method proposed by Kioka and Ashi (2015), ~~i.e., using the relation $V = \pi R^2 H/3$, that is~~ corresponding with the volume of the cone even though many volcanoes have a flat-topped summit and may be better approximated to a trunk cone

(e.g., potentially the BMV). Figs. 6(b and c) ~~shows~~ show that some mud volcanoes display a large diameter but low relief, whereas very few mud volcanoes exhibit small diameter but prominent topography. An approximately linear trend between increasing diameters and heights can be inferred. In this trend, the BMV stands close to the ~~lower D and H~~ low _D_ and _H_ values. Fig. 6(d) shows that the ~~H/R~~ _H/R_ ratio of all mud volcanoes is $\leq 0.4$. The same ratio for the BMV is $< 0.1$. There is a scattered distribution of volumes (~~V~~ _V_), mostly ranging in the ~~106-109~~ $10^6$-$10^9$ m$^3$ interval. The BMV volume ~~corresponds to~~

(6.9 x ~~106~~ $10^6$ m$^3$ ~~(Fig. 6d).~~

### 4.3 ~~Single-channel Chirp Seismic Reflection Profiles~~

#### 4.2.1 ~~Method and Data~~

~~During the Seismofaults 2017 survey, to define the local geological setting and a high resolution seismic stratigraphy of the BMV and surrounding area, we acquired a set of chirp seismic profiles (Figs. 7b-d, 8, and S2) using a frequency-modulated~~

30 ~~source operating in the frequency range of 2-7 kHz (Benthos Chirp III) and recorded with a 0.5-0.8 s sweep length. Maximum sub-bottom penetration is up to about 40 ms (TWT) and vertical resolution is about 0.7 ms (TWT). We processed the chirp profiles using the GeoSuite All Works software, applying Time Varied Gain. Thicknesses and depths of seismic profiles are described in two-way travel time (TWT; Figs. 7b-d and 8), with a seismic velocity of 1500 m/s being used to convert two-way~~

~~travel time into depth. Seismic data interpretation was carried out through the Kingdom Suite software also integrating the~~ ~~bathymetric data and a gravity core (Fig. 9)~~ is highlighted in Fig. 6d.

### 4.2.1 ~~Results~~

### 4.3 Single-channel Chirp Seismic Reflection Profiles

5   In the chirp seismic profiles (253, 241, and 246 in Fig. 7b-d), we recognized the following seismostratigraphic units:

Unit U0 corresponds to the seafloor seismic unit of the BMV and is characterized by sharp bottom echoes with no sub-bottom reflections or little acoustic penetration with chaotic reflections and locally large hyperbolae. Adjacent to the BMV, we recognized the following four further seismic units (U1, U2, U3, and U4).

Unit U1 is mostly characterized by parallel to subparallel semi-continuous reflections with variable amplitude. The up-

10  per boundary is defined by a horizon (H1) generally characterized by a reflector with high-amplitude and high continuity. Locally (e.g., toward the BMV), amplitude and continuity of horizon H1 decrease and its reflection becomes indistinguishable. Furthermore, the H1 reflector is locally erosive in the sector adjacent to the BMV and tends to become a conformity surface far away from the BMV. Locally, toward the northern escarpment, reflections in U1 are truncated by an erosional surface. Adjacent to the BMV, normal faults displaced this unit.

Unit U2 is characterized by a quasi-transparent (reflection free) seismic facies and shows draping of the underlying relief. This unit has a wedge-shaped morphology, thickening toward the flank of the BMV (241 in Fig. 7c), where it reaches a thickness of at least 25 ms. The lower boundary of U2 is the horizon H1, whereas the upper boundary is ~~an erosional surface~~ defined by a horizon (H2) characterized by a reflector with high to medium amplitude and high continuity that locally (e.g., toward the BMV) become discontinuous. Horizon H2 is an erosional surface with local extent. U2 is distributed in the sector

comprised between the northern escarpment and the BMV and is confined by the northern escarpment. Locally, U2 can be recognized also along the southern flank of the BMV edifice (246 in Fig. 7d).

Unit U3 is characterized by high amplitude reflections and by a limited thickness. The internal filling configuration of this unit is a parallel onlap. Locally, the seismic facies is semitransparent. The lower boundary of U3 is represented by horizon H2, whereas the upper boundary is defined by a horizon (H3) characterized by a reflector with high to medium amplitude and high

continuity. In places, amplitude and continuity of horizon H3 decrease. Furthermore, horizon H3 ~~that~~ is locally characterized by erosion.

Unit U4 is characterized by a transparent/chaotic seismic facies (241, 246, and 253 in Fig. 7b-d). It has a wedge-shaped morphology and thickening toward the northern escarpment (max. thickness of about 20 ms). The lower boundary of U4 is represented by horizon H3, locally characterized by erosion. The upper boundary is the seafloor. U4 is well discernible

(max. ~~thickenss~~ thickness = c. 20 ms TWT) in the sector comprised between the northern escarpment and the BMV and less discernible but still present (max. ~~thickenss~~ thickness = c. 4 ms TWT) to the south of the BMV.

~~Based on the seismic characters, we interpret the five seismic units as follows (Fig. 8):~~

Unit U0 is interpreted as the BMV main edifice. The transition between the rim and the summit caldera is identified by the large hyperbolae. The floor of the summit caldera is not penetrated by seismic signal, possibly indicating the occurrence of mud breccias deposits that are typical of mud volcanoes (van der Meer, 1996; Gennari et al., 2013).

Unit U1 is interpreted as slope deposits including turbidite-hemipelagite intervals. Locally, slope deposits are eroded by U2 (i.e., mud volcano deposits).

Unit U2 is interpreted as a mud volcano deposit belonging to the BMV, due to its wedge-shaped quasi-transparent (reflection free) seismic facies thinning away from the BMV center. This deposit can be related to eruptive events or post eruptive instability of the following types:

(1) Buried mudflow deposits: gravity flow deposits related to slope instability of the mud volcano.

(2) Buried mud volcano sediment: similar wedge-shaped seismic units have been interpreted by Evans et al. (2008) as massive and structureless sediment extruded from the volcano center (see, for comparison, figures 7 and 8 in Evans et al., 2008).

Unit U3 is interpreted as ponded deposits, including turbidite-hemipelagite intervals and thin mass transport deposits. Locally, ponded deposits are eroded by U4 (i.e., mass transport deposits).

Unit U4 is interpreted as consisting of mass transport deposits originated by slope instability along the northern escarpment (i.e., toward the coast). This interpretation is based on the quasi-transparent (reflection free) seismic facies of this unit and on its wedge shape with reduced thickness moving away from the northern escarpment. Interpretation of U4 is also based on direct evidence from the gravity core of Fig. 9 (see next subsection).

## 4.4 Seabottom Gravity Core

### 4.4.1 Method and Data

We collected the SF17-01 core using a 1.2 ton gravity corer with coring pipes 6 m long (see location in Fig. 7a). We analyzed the core sections through a multi-proxy approach involving high-resolution digital photographs and determination of physical properties and sand content as deduced through the weight of selected samples. We also acquired high resolution magnetic susceptibility (MS) through a core log system (Bartinghton model MS2, 100 mm loop sensor) with a sampling interval of 1.0 cm.

### 4.4.1 results

We collected the SF17-01 core at the toe of the peripheral rim marking the external slopes of the BMV (Fig. 7a). Chirp profiles collected during coring operations (Figs. 7b-d and 8) shows a chaotic unit (U4, Figs. 7b-d and 8) resting on the mud volcano flanks possibly related to re-sedimentation processes (e.g., slumping or turbidity currents). Chirp-core correlation suggests that the chaotic and transparent sediments are represented by the 2-m-thick fining-upward resedimented unit between 0.50 and 2.50 m depth (Fig. 9). The base of this unit is marked by an abrupt increase in sand content and by an erosional basal contact (Fig. 9).

Although the gravity core did not sample mud breccias, it shows indirect evidence of fluid/mud flow as pointed out by the presence of patchy/cloudy facies *(sensu* Staffini et al., 1993; Cita et al., 1996) where sediment disturbance is caused by fluid expulsion. According to Staffini et al. (1993), the main feature of a patchy/cloudy facies is that the mud breccia is characterized by patches and clouds of different colours. No visible clasts larger than sand-size are observed and the mud breccia shows typical colour changes. The sampled section below the base of the resedimented unit is characterized by silty-clay bulk sediments containing irregularly clustered intervals of differently colored coloured sediment patches (from dark grey, olive grey and brownish clouds within grey matrix) with several fragmented and vertically dislocated thin silty turbidites and volcanoclastic layers (Fig. 9). Sediments contain several vertical or sub-vertical micro-pipes suggesting sediment reworking by fluid migration. We associate this sediment structure to the patchy/cloudy facies (e.g., Staffini et al., 1993; Cita et al., 1996), which was already described in the surrounding areas in association with mud volcanism (Panieri et al., 2013). The presence of the patchy/cloudy facies in the core is identified by changes in magnetic susceptibility, whereas the sand content does not show diagnostic changes (Fig. 9).

## 4.5 Multi-Channel Seismic Reflection Profiles

### 4.5.1 Method and Data

In this section, we present two multiple channel seismic reflection profiles and related interpretations (Fig. 10; see tracks in Fig. 1). The first profile (Fig. 10a) is the CROP M-31 (Scrocca et al., 2003) in the version interpreted by Polonia et al. (2016). The second profile is the CA99-215 (Fig. 10b) both in the version interpreted by Polonia et al. (2016) and in a new version (Figs. 10c and 10d; see also Fig. S2) deriving from a recent reprocessing kindly provided by Spectrum Geo Ltd (http://www.spectrumgeo.co within the framework of a confidentiality agreement between Spectrum Geo and Sapienza University of Rome. The CA99-215 profile was acquired in 1999 and reprocessed in 2001. Reprocessing mainly consisted in a standard PSTM (pre-stack time migration) processing sequence, resulted in an improved data quality. Stratigraphic syntheses from the Fosca 1 and Floriana 1 wells (original well data are from the Videpi public database available online at http://unmig.sviluppoeconomico.gov.it/videpi/videpi.asp) are also presented (Fig. 10f) to help interpreting the seismic profiles (see well location in Fig. 1).

### 4.5.1 results

The CROP M-31 and CA99-215 profiles (Fig. 10) are located within the inner plateau of the Calabrian Arc, a morphologically flat area with forearc basins formed on the top of the continental basement (Polonia et al., 2011, 2016). The inner plateau of the Calabrian Arc is characterized by a generally thick portion of Plio-Quaternary deposits overlying the Messinian sediments sampled in several nearby boreholes (e.g., Floriana 1 and Fosca 1 wells; Figs. 1 and 10f) and recognized both in the CROP M-31 and in the CA99-215 seismic profiles (Polonia et al., 2016). Below the Messinian deposits, the upper Oligocene-upper Miocene turbidite deposits unconformably overlie the basement units.

The seismic profiles are characterized by different structural domains with peculiar deformation patterns (Polonia et al., 2016). In particular, the Ionian Fault system (see this system in map view in Fig. 1) is observed in the seismic profiles (between

s.p. ~~1100~~ 200 and 1400 of the ~~CROP M-31 profile and on the western side of the CA99-215 profile~~ A99-215 profile; Fig. 10b). The Ionian Fault system, characterized by transtensive faulting, seems to re-activate pre-existing Miocene-Pliocene thrusts (Polonia et al., 2016) and it separates the Calabrian Arc into two sectors: the Western and the Eastern lobes (Figs. 1 and 10).

The Eastern Lobe, where the BMV is located, is characterized by a more elevated accretionary wedge and by steeper topographic slopes than those of the Western Lobe, as shown in the sector between s.p. 1500 and 2600 of the CA99-215 profile (Fig. 10; Polonia et al., 2011).

~~The CA99-215 profile displays numerous faults characterized by different kinematics and related to the complex evolution of the area, namely: thrust faults, which are the result of the post-Messinian shortening, and normal faults deriving from the extensional process acting in more recent times (from lower Pleistocene to present times) as shown by the faults reaching the sea bottom on the eastern side of the seismic profiles (Fig. 10). Earthquakes focal mechanisms located in the study area (Figs. 1 and 2) show that faults are mostly characterized by strike-slip and transtensional kinematics defining a flower-like oblique-slip deformation zone (Fig. 10b). In this area, one of the main reflective horizons, usually discernible in most seismic profiles, is the top of the Messinian deposits. The Messinian deposits cored in the nearby wells (in particular in the Squillace Gulf) consist of clay-dominated deposits with interbedded gypsum, anhydrite, and halite (Capozzi et al., 2012). The Floriana 1 well shows, for instance, a thick portion (more than 1300 m) of the Gessoso Solfifera Formation (consisting of crystalline gypsum, anhydrite, and clay layers) whereas, in the Fosca 1 well, the Messinian deposits are mostly formed by clay and silt for a total thickness of 110 m (Fig. 10f). The distribution and sedimentary pattern of the Messinian deposits suggest indeed that the basin was already articulated and tectonically controlled at the time of Messinian sedimentation (Capozzi et al., 2012). The occurrence of Messinian evaporites as well as clays and silts of the same age is very relevant for mud volcanism. Indeed, according to many studies on mud volcanoes in the Ionian Sea and in the adjacent Mediterranean Ridge (located off southern Peloponnesus and Crete), the disruption of the Messinian low-permeability layers concurred to or was the key process controlling the ascent of pressurized fluids entrapped below these sealing layers and the consequent outflow and mud volcanism (Capozzi et al., 2012).~~

~~Zooming into the BMV area (Figs. 10d, 10e, and S2), we observe the following main features: (1) the presence of top Messinian deposits reflectors; (2) the disruption of these reflectors; (3) the presence of normal faults; and (4) the presence of an area of fluid-rock interaction similar to many ones observed in seismic profiles below mud volcanoes (Capozzi et al., 2012). In particular, the area just below the BMV, which is about 200 m far away from the CA99-215 profile (Fig. 1), shows (at a time-depth of 2 s TWT) a rock volume characterized by a chaotic and slightly transparent seismic reflection signal (Figs. 10d, 10e and S2). It is relevant to note that this area is located right at the top of a series of faults. The Messinian evaporites are located between 2 and 3 s TWT. Considering a seismic velocity of 1500 m/s for the sea water column and 2000 m/s for the post-Messinian column of sediments (Gallais et al., 2012), the Messinian evaporites should occur at about -3000 m.~~

## 4.6 Geochemical Features of Four Sea Water Columns

### 4.6.1 ~~Method and Data~~

We sampled sea water columns at various depths in four localities of the Ionian Sea (Fig. 1 and Tables 1 and 2): namely, above the BMV and above the GeoC1, GeoC2, and GeoC3 localities that are along or nearby major presumably-active fault zones (Polonia et al., 2012, 2016). We carried out vertical casts by Rosette and Niskin bottles to determine the geochemical features and dissolved gases at the sea bottom and along the water columns (Fig. 11 and Tables 1 and 2). Samples are compared with the local air-saturated sea water (ASSW) used as benchmark.

The samples, specifically collected for the extraction of the whole gas phase for chemical and isotopic analyses, were stored in 240 ml pyrex bottles sealed on board using rubber/teflon septa and purpose-built pliers, and analyzed within two weeks. Details of the sampling methodology are reported in Italiano et al. (2009, 2014). During the cruise, the sampled bottles were stored upside down, keeping the necks with the rubber septa submerged in sea-water until they were transferred to the laboratory. The collected sea-water samples underwent laboratory procedures for both chemical (concentration of dissolved gas species) and isotopic (helium isotopes) determinations. In the laboratory, the dissolved gases were extracted after equilibrium was reached at constant temperature with a host-gas (high-purity argon) injected in the sample bottle (see Italiano et al., 2009, 2014, for furt The chemical analyses of $O_2$, $N_2$, $CH_4$, and $CO_2$ were carried out by gas-chromatography (Agylent 7800B equipped with a double TCD-FID detector) using argon as carrier gas. Typical analytical uncertainties were within $\pm 5\%$. He concentration was determined by mass spectrometry, during 3He/4He analyses. Helium isotope and 4He/20Ne ratios were carried out on gas fractions extracted following the same procedure described above and purified following the method described in the literature (Hilton, 1996; Sano and Wakita, 1988; Italiano et al., 2001). The isotopic analyses of the purified helium were performed using a static vacuum mass spectrometer (GVI5400TFT) that allows the simultaneous detection of 3He and 4He ion beams, thereby keeping the 3He/4He error of measurement very low. The used analytical method also requires to alternatively run one sample and one purified air shot used as internal 3He/4He standard. Typical uncertainties in the range of atmospheric He-type samples are within $\pm 1\%$. The 4He/20Ne ratios were calculated by the relative peak heights measured on the same mass spectrometer.

Water samples of 100 ml were stored in PVC bottles for total alkalinity titration: 50 ml were filtered by a 0.45 $\mu$m filter and acidified by $HNO_3$ 0.1 N for cations (Ca, Mg, Na, and K) determination, whereas the non-acidified samples were collected for anions (Cl, F, and $SO_4$) determination. pH was measured by an electronic device calibrated in situ using buffer solutions. In the laboratory, chemical analyses of the major constituents were carried out by ion-chromatography (Dionex ICS-1100) both on filtered (0.45 mm) acidified ($HNO_3$ Suprapur) water samples (Na, K, Mg, and Ca), as well as on untreated samples (F, Cl, Br, $NO_3$, and $SO_4$). The $HCO_3$ content was measured by standard titration procedures with hydrochloric acid. Typical uncertainties are $\pm 5\%$.

### 4.6.1 Results

The analytical results in terms of chemical and isotopic composition of the water samples and the dissolved gases are listed in Tables 1 and 2, respectively. The fault zone data (GeoC1, 2, and 3) are from samples coming from vertical casts performed far from the BMV over the Ionian and Alfeo-Etna fault systems (Fig. 1). The typical composition of an Air Saturated Sea Water (ASSW) is reported for comparison (Tables 1 and 2).

The five sea-water samples collected along the water column over the BMV (except the sample collected near the seabed) show a chemical composition similar to the fault zone samples (GeoC1, 2, and 3; Fig. 11 and Tables 1 and 2). ~~Although the~~ The concentration values for major cations (Na, K, Mg, Ca; Table 1) show ~~comparable~~ similar values for the GeoC1, 2, and 3 samples and the ones from the BMV (including water column and bottom samples)~~, the Br content display small but detectable~~

5 ~~differences between the GeoC1, 2, and 3 and the BMV samples, respectively (Table 1). Along the BMV water column, the Br content is between 11 and 47% higher than the GeoC1, 2, and 3 samples. Contrastingly, the~~ . The dissolved gases detected along the water column (Table 2) exhibit geochemical features with large differences from the sea water equilibrated with the atmosphere (ASSW). In particular, the composition of the dissolved gases (Table 2) shows the presence of air-derived gases ($N_2$ and $O_2$) along with non-atmospheric gases ($CO_2$ and $CH_4$). The analytical results clearly display a slight decrease in

oxygen content (~~as expected~~i.e., compare ASSW and the rest of data in Fig. 11) besides a significant increase in He, $CH_4$, and $CO_2$. The $CH_4$ content along the water column and at the sea bottom (BMV and GeoC1, 2, and 3) ranges ~~from~~ over two to three orders of magnitude higher than the ASSW with the highest concentration recorded at the BMV depth (-1337 m). The $CO_2$ content above the BMV, in particular, is the double of the ASSW (0.24 ccSTP/L) at the depth of 100 m and it increases up to ~~0.71~~ 0.73 ccSTP/L at the sea bottom (-1337 m) (Fig. 11 and Table 1).

~~A few years ago (2003), one of us (FI) collected and analyzed fluid samples from a shallow well located in the Calabria main land nearby the Palizzi mud volcano (PMV in Fig. 1, located about 25 km to the WNW of the BMV area). Unfortunately, no documentation exists concerning the PMV except our original geochemical data reported in Tables 1 and 2. The PMV was indeed actively venting geofluids at the time of our sampling (2003) but it was then soon destroyed by human activity and cementation.~~ The PMV is characterized by a dissolved gas phase mainly composed by Nitrogen and $CH_4$ with a significant

helium concentration and a slight amount of $CO_2$. Oxygen is below the detection limits (Tables 1 and 2).

The helium isotopic signature of samples coming from the GeoC1, 2, and 3 casts show an atmospheric signature from the surface to the depth of 1000 m. The bottom sample displays a lower ratio than the air with a higher ~~4He~~$^4$He/~~20Ne~~ $^{20}$Ne ratio. Two samples from the BMV cast were analyzed to determine the helium isotopes. Results approximately match those from the GeoC1, 2, and 3 casts showing a slight but detectable difference with the respect to the atmospheric ratios (as expected for an

25 ASSW) for both ~~3He~~$^3$He/~~4He and 4He~~$^4$He and $^4$He/~~20Ne~~ $^{20}$Ne ratios (Fig. 11 and Tables 1 and 2).

## 5 ~~Discussion and conclusions~~Interpretation

### 5.1 ~~Focuses~~

In this section, we interpret some of the presented data, particularly (but not only) the seismic ones (Figs. 7-10) that require specific interpretation. Further discussion on and synthesis of all data are reported in the next section.

First of all, earthquake epicenters and focal mechanisms (Figs. 1 and 2) show that the area surrounding the BMV is seismically active as it is populated by earthquakes. Concerning the activity of the BMV, consistently with the resolution of the multibeam equipment, multibeam water column data recorded during the Seismofaults 2017 survey does not highlight acoustic backscatter anomalies related to large amount of fluids/mud escaped from the seafloor (Fig. 5b; e.g., Römer et al., 2014). This

evidence suggests that no paroxysmal activity is ongoing (or at least was ongoing at the time of the Seismofaults 2017 survey) from the BMV. However, our geochemical data (Fig. 11) show active fluid circulation through the BMV. Further discussion on this theme is proposed in the next section. Moreover, sediments from the gravity core contain several vertical or sub-vertical micro-pipes suggesting sediment reworking by fluid migration (Fig. 9). We associate this sediment structure to

5 the patchy/cloudy facies (e.g., Staffini et al., 1993; Cita et al., 1996), which was already described in the surrounding areas in association with geofluid ascent and mud volcanism (Panieri et al., 2013).

Concerning the single-channel chirp profiles, based on their seismic characters, we interpret the five seismic units identified in Fig. 7 as follows (Fig. 8):

Unit U0 is interpreted as the BMV main edifice. The transition between the rim and the summit caldera is identified by the

10 large hyperbolae. The floor of the summit caldera is not penetrated by seismic signal, possibly indicating the occurrence of mud breccias deposits that are typical of mud volcanoes (van der Meer, 1996; Gennari et al., 2013).

Unit U1 is interpreted as slope deposits including turbidite-hemipelagite intervals. Locally, slope deposits are eroded by U2 (i.e., mud volcano deposits).

Unit U2 is interpreted as a mud volcano deposit belonging to the BMV, due to its wedge-shaped quasi-transparent (reflection

free) seismic facies thinning away from the BMV center. This deposit can be related to eruptive events or post eruptive instability of the following types:

~~We focus our discussion on two main aspects of the BMV, that are:~~ (1) ~~its origin and activity, and~~ Buried mudflow deposits: gravity flow deposits related to slope instability of the mud volcano.

(2) Buried mud volcano sediment: similar wedge-shaped seismic units have been interpreted by Evans et al. (2008) as

massive and structureless sediment extruded from the volcano centre.

Unit U3 is interpreted as ponded deposits, including turbidite-hemipelagite intervals and thin mass transport deposits. Locally, ponded deposits are eroded by U4 (i.e., mass transport deposits).

Unit U4 is interpreted as consisting of mass transport deposits originated by slope instability along the northern escarpment (i.e., toward the coast). This interpretation is based on the quasi-transparent (reflection free) seismic facies of this unit and

25 on its wedge shape with reduced thickness moving away from the northern escarpment. Interpretation of U4 is also based on direct evidence from the gravity core of Fig. 9.

Concerning the multi-channel seismic reflection profiles, the CA99-215 profile displays numerous faults characterized by different kinematics and related to the complex evolution of the area, namely: thrust faults, which are the result of the post-Messinian shortening, and normal and strike-slip faults deriving from the extensional process acting in more recent times

30 (from lower Pleistocene to present times) as shown by the faults reaching the sea bottom on the eastern side of the seismic profiles (Fig. 10). Earthquakes focal mechanisms located in the study area (Figs. 1 and 2) ~~the potential for the use of this structure and similar ones in favor of the mitigation of natural hazards~~ show that faults are mostly characterized by strike-slip and transtensional kinematics defining a clear flower-like structure on the western side and an oblique-slip deformation zone on the eastern side (Fig. 10b). In this area, one of the main reflective horizons, usually discernible in most seismic profiles, is the

35 top of the Messinian deposits. The Messinian deposits cored in the nearby wells (in particular in the Squillace Gulf) consist of

clay-dominated deposits with interbedded gypsum, anhydrite, and halite (Capozzi et al., 2012). The Floriana 1 well shows, for instance, a thick portion (more than 1300 m) of the Gessoso Solfifera Formation (consisting of crystalline gypsum, anhydrite, and clay layers) whereas, in the Fosca 1 well, the Messinian deposits are mostly formed by clay and silt for a total thickness of 110 m (Fig. 10f). The distribution and sedimentary pattern of the Messinian deposits suggest indeed that the basin was already articulated and tectonically controlled at the time of Messinian sedimentation (Capozzi et al., 2012). The occurrence of Messinian evaporites as well as clays and silts of the same age is very relevant for mud volcanism. Indeed, according to many studies on mud volcanoes in the Ionian Sea and in the adjacent Mediterranean Ridge (located off southern Peloponnesus and Crete), the disruption of the Messinian low-permeability layers concurred to or was the key process controlling the ascent of pressurized fluids entrapped below these sealing layers and the consequent outflow and mud volcanism (Capozzi et al., 2012).

The identification on seismic profiles of submarine mud volcanoes (i.e., their subsurface roots) can depend on and be limited by different factors, such as the complexity of the tectonic setting and/or the seismic characteristics of the hosting succession (Dimitrov, 2002; ?). Identification of mud volcanoes located in complex tectonic settings, such as the accretionary complex of the present work, is not straightforward; however, as we will explain below, we were able to observe a few seismic features that could be interpreted as diagnostic of mud volcanism and fluid-rock interaction (e.g., Dimitrov, 2002). For instance, immediately beneath the sea floor, where a mud volcano is located, seismic reflectors are usually characterized by locally strong amplitudes. This seismic effect can be associated with fluid-rock interaction (Dimitrov, 2002). Furthermore, chaotically disrupted seismic pattern and short seismic horizons are typical features related to the presence of a conduit and fluid-rock interaction beneath many mud volcanoes. Concerning the BMV (Figs. 10d, 10e, and S2), we observed the following main features: (1) the presence of top Messinian deposits reflectors; (2) the disruption of these reflectors; (3) the presence of normal faults; and (4) the presence of an area of fluid-rock interaction similar to many ones observed in seismic profiles below mud volcanoes (Capozzi et al., 2012; Dimitrov, 2002; ?). In particular, the area just below the BMV, which is about 300 m far away from the CA99-215 profile (Fig. 1), shows (at a time-depth of 2 s TWT) a rock volume characterized by a chaotic seismic reflection signal, with disrupted and short reflections (Figs. 10d, 10e and S2). This chaotic pattern is located in sediments otherwise characterized by reflectors showing high continuity. It is relevant to note that this area is located right at the top of a series of faults. The Messinian evaporites are located between 2 and 3 s TWT. Considering a seismic velocity of 1500 m/s for the sea water column and 2000 m/s for the post-Messinian column of sediments (Gallais et al., 2012), the Messinian evaporites should occur at about -3000 m.

# 6 Discussion

## 6.1 Origin and Activity

### 6.1.1 General Context

Concerning the origin of the BMV, we here reconsider all the above-described data. First of all, we mention the fact that the Ionian Sea, where two active accretionary prisms occur and obliquely-converge (i.e., the Calabrian Arc and the Mediterranean

Ridge), is a region rich of active or recently-active mud volcanoes ~~(Cita, 1981; Camerlenghi et al., 1992; Limonov et al., 1996; Capraro et a~~
The BMV itself was previously signaled and hypothesized as a mud volcano by Gutscher et al. (2017) and Loher et al. (2018).
The BMV occur on top of the Calabrian accretionary wedge in a rather isolated location with several further mud volcanoes
occurring a few kilometers or a few tens of kilometers toward NE along the prism (Fig. 1; Loher et al., 2018).

### 6.1.2 Morphology

From a morphological point of view, the ~~shape~~ main geometric features of the BMV ~~is that~~ are those typical of most
mud volcanoes ~~(i.e., pie-type, Kioka and Ashi, 2015; Kioka et al., 2015)~~ (i.e., Kioka and Ashi, 2015) and all its measured pa-
rameters (height ~~, diameter~~ $H$, diameter $D$, and volume $V$; Fig. 6) are well within the range typical of marine mud vol-
canoes ~~(Kioka and Ashi, 2015; Kioka et al., 2015; Kirkham et al., 2017).~~ (Kioka and Ashi, 2015; Kirkham et al., 2017). From
the morphological analysis, the BMV could be primarily interpreted as a pie-type mud volcano; however, looking at the slope
values highlighted by the acquired multibeam data (Fig. 4), the < 5° slope angle criterion proposed by Kopf (2002) is not
satisfied, being the BMV slope values > 10° (Fig. 4).

The stratigraphic analyses realized through direct and indirect methods (Figs. 7-10) show the occurrence of lithological units
consistent with the activity of a mud volcano, such as the U2 deposits in Figs. 7c and 8, the evidence of fluid/mud flow in the
patchy/cloudy facies of the gravity core (Fig. 9), and the ~~semitransparent seismic facies (typical of mud volcanoes for the host rock-geofluid~~
seismic reflection signal with disrupted and short reflections (typical of mud volcanoes for the host rock-geofluid interaction; e.g., Dimitrov
the BMV (Figs. 10d). However, from the available data, we cannot infer any eruptive dynamics for the BMV due to the limited
resolution of the seismic imaging and to the BMV location with respect to the seismic lines (c. 300 m). The dimension of
the conduit and paleoflows are not visible in the seismic profiles, as, for example, the Christmas-tree structures described by
(Somoza et al., 2003). From previous literature (e.g., Kioka and Ashi, 2015), we can only infer that the BMV dimensions and
its computed volume suggest a polygenetic behaviour, so that we argue that the main fluid conduit has been possibly utilized
several times to increase the volume of the volcano itself.

### 6.1.3 Sealing

~~Moreover, particularly~~ Particularly in the Mediterranean Sea, the origin of most mud volcanoes has been linked in a cause-
effect relationship with the sealing action exerted by the Messinian evaporites, causing fluid entrapment underneath and conse-
quent fluid overpressure ~~(Camerlenghi et al., 1995; Chamot-Rooke et al., 2005; Rabaute and Chamot-Rooke, 2007; Camerlenghi and Pini,~~
Also in the case of the BMV, the seismic cross-sections show the presence (and disruption underneath the BMV) of the
Messinian evaporites (Fig. 10), which therefore may have been decisive in building the necessary overpressure of fluids to con-
sequently form the mud volcano itself (Bertoni and Cartwright, 2015; Al-Balushi et al., 2016). The disruption of the Messinian
layers suggests that the conduit for the ascent of geofluids through the BMV is presently open. This hypothesis is also substan-
tiated by the geochemical data (Fig. 11) that are discussed below.

### 6.1.4  Ongoing Activity

We have very little evidence to discuss the ongoing activity of the BMV also because we collected data from this structure only in a single campaign in May 2017. From a morphological point of view, the BMV seems well structured (Fig. 4) and therefore its main edifice-building paroxysmal activity may have substantially ceased. Moreover, the flanks of the main vol-
canic edifice seems partly covered by younger products (i.e., over its flanks; Figs. 7c-d and 8) deriving from nearby gravity instabilities (i.e., from the continental slope of the Calabrian-Ionian margin; Figs. 1 and S1). Also, the backscatter data show no extensive anomalies (in May 2017) related to large amount of mud and fluids escaped from the seafloor (e.g., Römer et al., 2014); however, this evidence does not detract from the fact that the volcano may currently be quiescent and therefore may erupt in the future. To this end, both the geochemical and the reflection seismic evidence show that some fluid activity below the
BMV is probably ongoing. Fig. 11, in particular, shows a trend of $CH_4$ and $CO_2$ enrichment for all the collected samples with respect of a sea water simply equilibrated with the atmosphere (ASSW). Although $CO_2$ and $CH_4$ may derive from degradation processes of organic matter, the geochemical composition of the sea water at the BMV depth and the composition of the ge-
ofluids from the PMV (Palizzi) onland area clearly indicate a $CH_4$ injection that is typical of most mud volcanoes. The isotopic composition of helium, although dominated by a typical atmospheric signature both at the GeoC1, 2, and 3 localities and at the
BMV locality, displays a detectable increase of radiogenic ~~4He~~ $^4$He of typical crustal origin, with the associated decrease of the isotopic ratio from about 0.93-1 Ra to 0.77 Ra detected in the GeoC1, 2 and 3 seabottom waters and 0,73Ra in the BMV seabottom water (Tables 1 and 2). ~~Combining the above-mentioned results with the enhanced Br content detected over the same site~~Hence, we propose that the BMV ~~has actually an open conduit~~ is actually infiltrated by open pathways as shown by the release of fluids into the surrounding sea water (Fig. 11 and Tables 1 and 2). Fluids are composed by a two-phase system: a $CH_4$-
dominated gas phase and hypersaline waters of ~~evaporate~~ evaporite type. The hypersaline waters are indeed probably generated by the dissolution of anciently buried evaporites (Messinian) and create dense anoxic brines~~with enhanced Br content, that~~, which are separated from the overlying oxygenated deep-seawater column due to ~~their density. The semitransparent seismic facies (typical of active mud volcanoes; e.g., Capozzi et al., 2012)~~ differential densities. The chaotic seismic reflection signal, with disrupted and short reflections (typical of active mud volcanoes; e.g., Dimitrov, 2002; Capozzi et al., 2012) recorded be-
neath the BMV (Fig. 10) as well as the disruption of the Messinian ~~evaporate~~ evaporite layers supports the above-proposed hypothesis of ~~an open conduit~~ pathways open to geofluids down to the Messinian ~~evaporate~~ evaporite layers (c. 3000 m depth; Fig. 10). Also the gravity core bears evidence of recent fluid circulation (Fig. 9). Moreover, some structures on top of the BMV (i.e., the rim and some small ridges; Fig. 4a-b) are morphologically similar to structures related to extrusion activity of mud volcanoes (e.g., Evans et al., 2008) and are therefore probably connected with the eruptive processes~~and~~. These structures are
not substantially covered by young sediments, hence attesting for a recent but undetermined time for the eruptive process of the BMV.

### 6.1.5 Draining Processes

In accretionary prisms, the causes (i.e., the engine) of mud volcanoes and related fluid activity are often found or hypothesized to be either the prism contraction and related fluid squeezing or the fault activity and related fluid ascent along fault damage zones. In other words, mud volcanoes can be caused by a contraction-related local dewatering or by a deeper crustal draining driven by the activity of normal and strike-slip faults (Rabaute and Chamot-Rooke, 2007; Gamberi and Rovere, 2010; Capozzi et al., 20 In the case of the BMV, we cannot unambiguously discriminate between these two main engines (prism contraction vs. fault activity). We hypothesize that both engines may concur or may have concurred to originate the BMV. Overall contraction is indeed slightly active in the Calabrian accretionary prism (Serpelloni et al., 2007; Billi et al., 2011; Faccenna et al., 2014; Polonia et al., 2016) as well as fault activity, particularly along prism-across (NW-SE) strike- to oblique-slip faults (Polonia et al., 2016, 2017). It is also true, however, that the seismological and reflection seismic data show the occurrence of a seismically-active flower-like system of faults right beneath the BMV (Figs. 2 and 10). This evidence let us think that (seismic) faulting more than the prism contraction process may have played a decisive role in the origin and feeding of the BMV, thus ultimately driving the ascent of geofluids from crustal depths. To this end, Polonia et al. (2017) have recently documented that the prism-across strike-slip faults in the Ionian Sea region are active and have been even capable of exhuming or contributing to exhume serpentinite domes from the lower plate of the Ionian subduction zone up to the upper plate and the Earth's surface.

### 6.1.6 Synthesis

Collectively, the data presented in this paper provide evidence for the fact that the studied structure (i.e., the BMV) is actually a mud volcano, through which fault-related crustal fluid activity and circulation is ongoing, and beneath which (seismic) faulting is active.

## 6.2 Potential Use of the BMV in the Science of Seismic Precursors

### 6.2.1 Previous Useful Results

Concerning the use of the BMV and similar structures in favor of the monitoring and mitigation of natural hazards, we first refer the reader to a few recent studies on geochemical precursors of earthquakes. Transient hydrogeochemical anomalies are increasingly becoming commonly recorded before M≥5.0 earthquakes at distances between 20 and more than 200 km from the earthquake epicenters. To understand their relevance for earthquake forecasting, we here briefly recall a few recent instances from Italy, Iceland, India, Japan, and Turkey. The 2016 Amatrice and Norcia earthquakes (central Italy) as well as the related sequence involved significant pore pressure changes (since about one week before the Amatrice earthquake; ?) and fluid movements both at deep and shallow crustal levels (Petitta et al., 2018; Tung and Masterlark, 2018), and were anticipated by hydrogeochemical anomalies recorded since April 2016 in springs from the central Apennines. In particular, increases of As, V, and Fe contents were recorded in groundwaters from springs monitored in the Sulmona area, about 70 km to the southeast of the epicentral area. Similar anomalies (i.e., As, V, and Fe) were also recorded in groundwaters from the San Chiodo spring

located within the epicentral area (Barberio et al., 2017)(Barberio et al., 2017; **?**). In 1995, eight months before the Mw 7.2 Kobe earthquake (Japan), the Cl and $SO_4$ concentrations in groundwater started to significantly and anomalously increase (20-30 km from the epicenter). Nine days before the earthquake, a peak in Rn concentration was also recorded (King et al., 1995). In 2002, anomalies in Cu, Zn, Mn, and Cr concentrations were recorded in groundwater 1, 2, 5, and 10 weeks, respectively, before a Mw 5.8 earthquake in northern Iceland (90 km from the epicenter; Claesson et al., 2004). In 2012, anomalous increases of Ca, Mg, K, and Cl concentrations in groundwater together with decreases of Na and $SO_4$ concentrations started between at least 20 and 30 days before the Mw 7.1 Van earthquake, Turkey (20 km from the epicenter; Inan et al., 2012). In 2012, significant increases in the Na, Si, and Ca concentrations were recorded in groundwater 4-6 months before two Mw$\geq$5.5 earthquakes in northern Iceland (70-80 km from the epicenter; Skelton et al., 2014)(70-80 km from the epicenter; Skelton et al., 2014; **?**). In 2004 and 2005, transient hydrogeochemical anomalies were detected in aquifer located to north of the Shillong Plateau, Assam, India, before two Mw$\geq$5.0 earthquakes (200 km from the epicenter). The [Na+K]/Si, Na/K, and [Na+K]/Ca ratios as well as conductivity, alkalinity, and Cl concentrations began increasing 3–5 3-5 weeks before the Mw 5.3 earthquake whereas the Ba/Sr ratio began decreasing 3–6 3-6 days before the Mw 5.0 earthquake (Skelton et al., 2008). In 2017, oxygen isotopic ratio anomalies of +0.24 $^0/_{00}$ relative to the local background were recorded in groundwater a few months before the Mw 6.6 Tottori earthquake in southwest Japan (5 km from the epicenter; Onda et al., 2018). Although the aforementioned scientific results are certainly encouraging, we must acknowledge that earthquake deterministic forecasting is still unfeasible, both through hydrogeochemical data and through other evidence.

### 6.2.2   The BMV

The geochemistry of the sea water column (sampled in May 2017) above the BMV compared with the sea water benchmark (ASSW; Fig. 11) shows a clear mineralization of the BMV-related waters together with an injection of $CO_2$ and $CH_4$, particularly in proximity of the BMV depth (Fig. 11). Moreover, the Helium isotope ratios (Table S4) shows a contribution by crustal fluids, also in this case particularly in proximity of the BMV depth. The ion content of the BMV-related waters (Table 1) is consistent with evaporite-type waters and this notion, in turn, is consistent with the hypothesis that the fluids feeding the BMV and other mud volcanoes in the Mediterranean area are entrapped below and within the Messinian evaporites (Fig. 10d; Camerlenghi et al., 1995; Chamot-Rooke et al., 2005; Capozzi et al., 2012; Ceramicola et al., 2014; Rovere et al., 2014; Bertoni and Cartwright, 2015). Below the BMV, these rocks occur at about -3000 m (Fig. 10d). The $CO_2$ and $CH_4$ (particularly $CH_4$) high content of the BMV-related waters is consistent with most mud volcanoes around the world (Milkov, 2000; Deville et al., 2003; Etiope and Milkov, 2004)(Etiope and Milkov, 2004). Moreover, the decreasing content of $CO_2$ and $CH_4$ moving (shallowing) from the BMV summit upward to the sea surface is a clear symptom that the source of these dissolved gases is the BMV itself. We therefore infer that, although the BMV is not likely undergoing full mud-volcanic activity (at least during our survey in May 2017; see water backscatter data in Fig. 5b) a crustal fluid conduit open crustal pathways for geofluids through this structure is open and exist and are actively venting. This hypothesis is also corroborated by the comparison with the geochemistry of three sea water columns above presumably active active fault zones in the Ionian Sea (GeoC1, 2, and 3; Figs. 1 and 11). These three sea water columns are indeed characterized by a very low content of $CH_4$ and

by a content of $CO_2$ significantly lower than that obtained for the sea water right on top of the BMV (i.e., at the BMV depth; Fig. 11).

### 6.2.3 Synthesis

Collectively, the geochemical, geophysical, and geologic data presented in this paper show that the BMV, likewise other on-shore monitoring stations previously realized ~~(e.g., Claesson et al., 2004; Skelton et al., 2014; Barberio et al., 2017; Huang et al., 2017)~~(e.g. could be a proper site where installing a cabled submarine multiparametric station (Fig. 12) to study possible relationships between the seismic cycle of the underlying active faults and geofluid emissions. Similar stations are active onshore in Italy, Iceland, China, and elsewhere, but, to the best of our knowledge, have never been installed in marine settings. In the case of the BMV and other mud volcanoes, dissolved gases such as $CO_2$ and $CH_4$ may rather easily be monitored by submarine devices ~~(Annunziatellis et al., 2009; Roberts et al., 2017)~~(Annunziatellis et al., 2009; Roberts et al., 2017; **?**). In particular for the seismically-active Ionian Sea, many other existing mud volcanoes (Gutscher et al., 2017; Loher et al., 2018) may host a monitoring station, but the BMV is so far the one characterized by the largest geological, geophysical, and geochemical dataset and its location seems connected with seismically-active faults (Fig. 2).

### 7 Conclusions

~~(1) The BMV on the Ionian Sea is , as previously hypothesized,~~ Mud volcanoes are ubiquitous structures both inland and offshore. Their occurrence is easily discernible mainly based on morphological-bathymetric evidence, but their geological significance is rather difficult to ascertain particularly offshore. With the Bortoluzzi Mud Volcano (BMV), we have demonstrated that an integrated geological, geophysical, and geochemical study can shed much light on the geological significance and ongoing activity of a mud volcano.~~ ~~

~~(2) Although evidence to support an ongoing robust and paroxysmal fluid circulation through the BMV are missing, the geochemistry of the water column above the BMV shows a clear contribution from crustal geofluids.~~

~~(3) This geochemical evidence support the hypothesis of a cause-effect relationship between the BMV ,~~ , even if located in marine settings. We now know, for instance, that the BMV is truly a mud volcano, it sits atop seismically-active faults, and its inner pathways are actively used for the rise of saline geofluids towards the Earth's surface from a depth of at least -3000 m. Although the paroxysmal activity of the ~~underlying active faults, and crustal-scale fluid circulation.~~

~~(4) It follows that the BMV and perhaps similar structures elsewhere could be selected in the future to geochemically tracking~~ BMV seems substantially extinct or at least quiescent, these inner pathways are open and used for the rise of geofluids more efficiently than nearby active fault zones that are usually considered efficient pathways for geofluids. This latter evidence constitutes a novelty, at least for offshore seismically-active areas, and indicates that mud volcanoes could be efficiently used to install cabled stations to monitor the relationship between the seismic cycle of ~~active faults as already done elsewhere in onshore localities.~~

~~(5) For what concerns the study area (Ionian Sea), on a first approximation, the presented geochemical data indicate that the BMV is a structure and locality for a potential monitoring station of the seismic cycle more suitable than fault zones, where the geochemical anomalies are significantly milder~~faults and the rise of geofluids. In particular, evidence from our work indicates that this type of stations may be installed also where mud volcanoes seem extinct for what concerns the main paroxysmal mud activity but still own efficient inner pathways for the circulation of geofluids.

*Data availability.* All data used for this paper are available in numerical and graphical forms in the tables and diagrams/images, respectively, in the paper itself or in the supplement associated to this paper. The gravity core collected from the Ionian seabottom is visible at ISMAR Bologna, Italy (http://www.ismar.cnr.it/organizzazione/sedi-secondarie/bologna) through the following contacts: alina.polonia@bo.ismar.cnr.it and luca.gasperini@bo.ismar.cnr.it. With this paper, we release 4970 km$^2$ of newly-acquired (during the Seismofaults 2017 cruise) high-resolution bathymetric data in numerical form for the Ionian Sea including the BMV (Fig. S1). This dataset is externally hosted and indexed with doi: XXXX.

*Author contributions.* All authors actively participated in conceiving the experiment and the paper, in discussing all results, in contributing to the writing of the paper and to the drawing of all figures, and in drawing the conclusions. ABil and MC coordinated the experiment. MC and Abos led the fieldwork. MC, ABos, ACor, CGC, ACon, ACos, GD, GL, LG, LP, and TS participated to the Seismofaults 2017 marine campaign on board the R/V Minerva Uno. Data were mostly processed by MC, ABos, CGC, ACon, ACor, EM, FI, GL, LP, ML, PE, and AP. The manuscript was mostly written by AB with contributions from all authors. Figures were mostly drawn by MC, Abos, CGC, ACon, EM, LP, and AP with contributions from all authors.

*Competing interests.* The authors declare that they have no conflict of interest.

*Acknowledgements.* ~~With~~ All used data are reported in the figures and tables of this paper and related supplementary material. Moreover, with this paper, we release 4970 km$^2$ of newly-acquired (during the Seismofaults 2017 cruise) high-resolution bathymetric data in numerical form for the Ionian Sea including the BMV (Fig. S1). The raw bathymetric data are externally hosted and indexed with doi: XXXX. M. Barbieri, L. Beranzoli, F. Frugoni, S. Monna, and many other colleagues from CNR, INGV, and Sapienza University of Rome are thanked for help and constructive exchanges. The officers and the crew of the R/V Minerva Uno and the scientific party of the Seismofaults 2017 survey are thanked for their cooperation during fieldwork. Some of the figures were produced with the Generic Mapping Tools software (http://gmt.soest.hawaii.edu). We thank F. Rossetti, M. Allen, and two anonymous reviewers for their constructive comments and editorial support. The research described in this paper is dedicated to Giovanni Bortoluzzi.

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

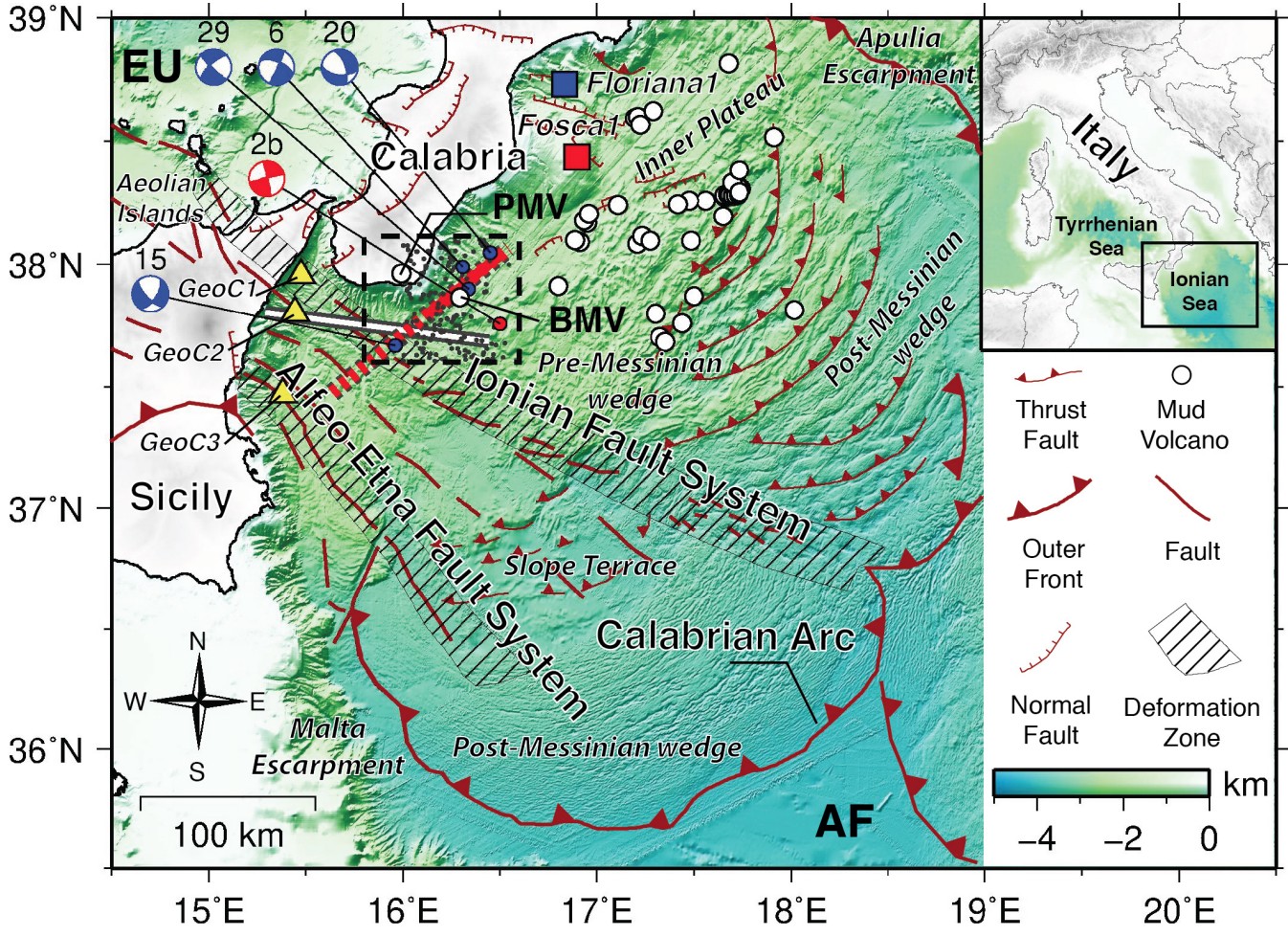

**Figure 1.** Bathymetric map the Ionian Sea (southern Italy) with main faults, mud volcanoes, earthquakes, tracks of multichannel seismic profiles, and location of the Fosca 1 (red square) and Floriana 1 (blue square) offshore wells. Location of mud volcanoes (except ~~the Palizzi mud volcano,~~ PMV, inland Calabria) are from Loher et al. (2018). The Bortoluzzi Mud Volcano is indicated with BMV.The ~~Palizzi mud volcano (onland Calabria) is indicated with PMV. Grey~~ thick white line corresponds to the CROP M-31 multichannel seismic profile (Fig. 10a). ~~Grey dashed~~ Dashed red line and ~~thick~~ solid red line are the CA99-215 multichannel profile (Fig. 10b) and its close up view reported in Fig. 10(c), respectively. Earthquake data (epicenters are indicated with thin grey dots whereas focal mechanisms with beach balls) surrounding the BMV area are from the European-Mediterranean Regional Centroid-Moment Tensors (RCMT) catalog (http://rcmt2.bo.ingv.it/) (red beach ball) and from previous papers (blue beach balls) by Orecchio et al. (2014) and Polonia et al. (2016). Faults are principally from Polonia et al. (2011), Polonia et al. (2016) and Bortoluzzi et al. (2017). The black box is the location of Fig. 2. AF stands for Africa Plate whereas EU for Eurasia Plate. GeoC1, GeoC2, and GeoC3 are locations of sea water column sampling.

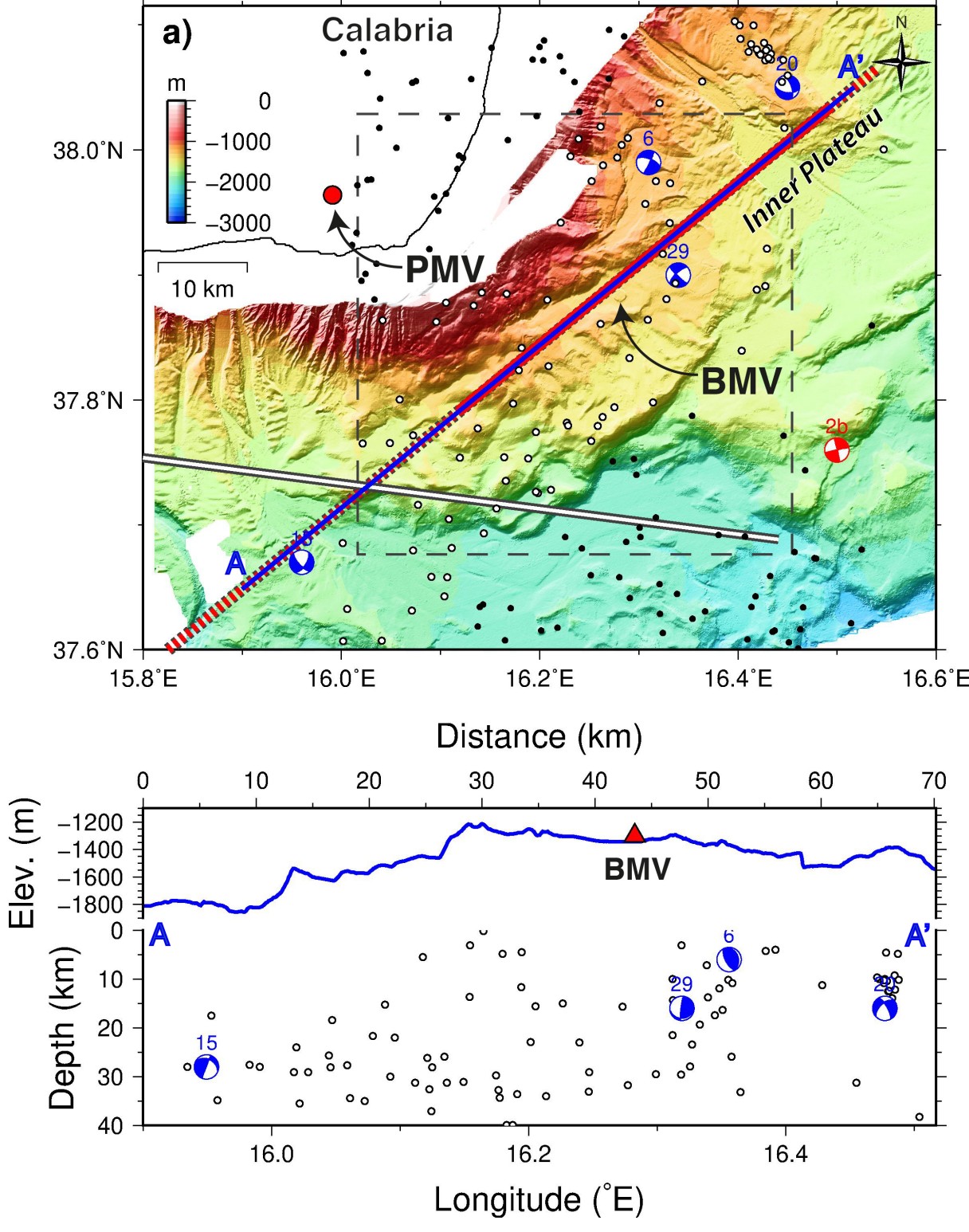

**Figure 2.** (a) High resolution bathymetry (see location in FigCaption next page.1) of the Calabrian off-shore, Ionian Sea, with location of the BMV. Earthquake epicenters and focal mechanisms (Tables S1 and S2) surrounding the BMV area are shown as dots

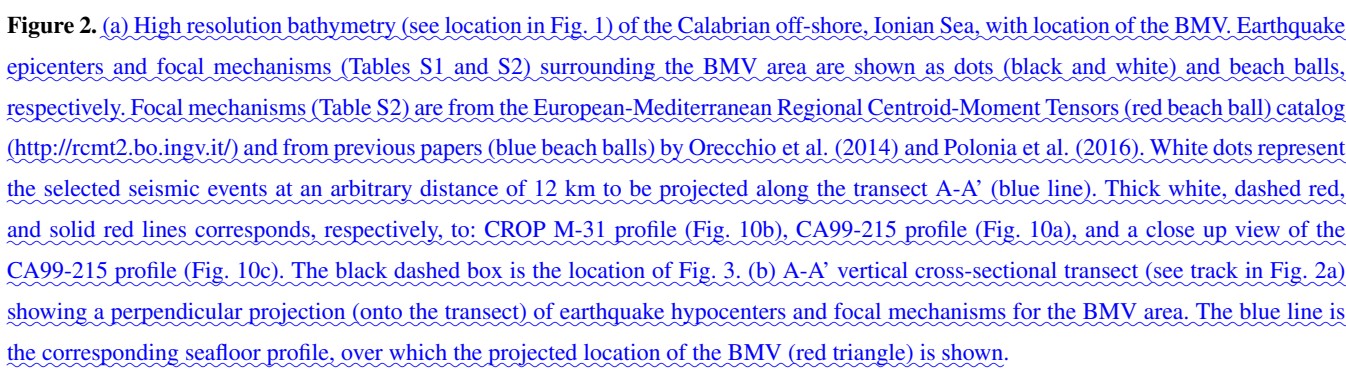

**Figure 2.** (a) High resolution bathymetry (see location in Fig. 1) of the Calabrian off-shore, Ionian Sea, with location of the BMV. Earthquake epicenters and focal mechanisms (Tables S1 and S2) surrounding the BMV area are shown as dots (black and white) and beach balls, respectively. Focal mechanisms (Table S2) are from the European-Mediterranean Regional Centroid-Moment Tensors (red beach ball) catalog (http://rcmt2.bo.ingv.it/) and from previous papers (blue beach balls) by Orecchio et al. (2014) and Polonia et al. (2016). White dots represent the selected seismic events at an arbitrary distance of 12 km to be projected along the transect A-A' (blue line). Thick white, dashed red, and solid red lines corresponds, respectively, to: CROP M-31 profile (Fig. 10b), CA99-215 profile (Fig. 10a), and a close up view of the CA99-215 profile (Fig. 10c). The black dashed box is the location of Fig. 3. (b) A-A' vertical cross-sectional transect (see track in Fig. 2a) showing a perpendicular projection (onto the transect) of earthquake hypocenters and focal mechanisms for the BMV area. The blue line is the corresponding seafloor profile, over which the projected location of the BMV (red triangle) is shown.

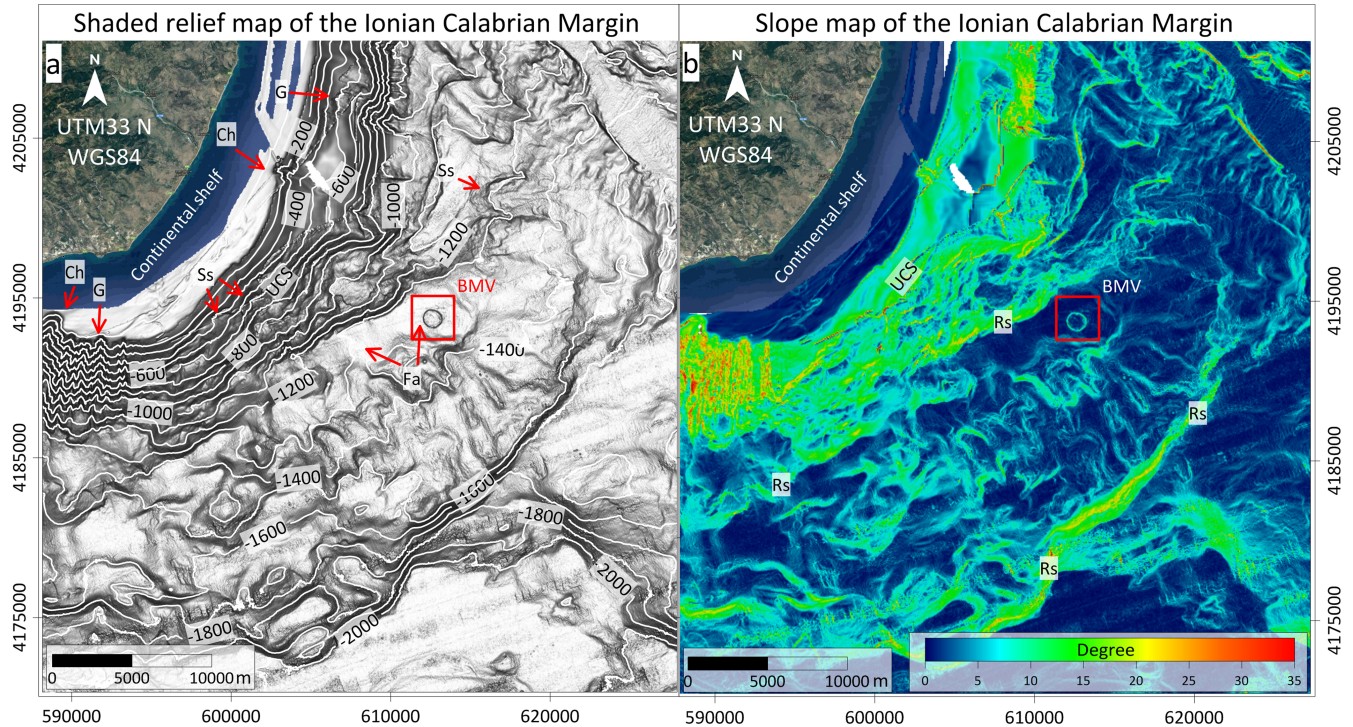

(a) Bathy-morphological map and (b) ~~slope~~ Slope map of the ~~upper~~ Upper portion of the ~~continental slope~~ Continental Slope (UCS) of the Calabrian-Ionian margin. Red square indicates the intraslope flat area hosting the circular high of the Bortoluzzi Mud Volcano (BMV). Location map is shown in Fig. 2a.

(a) Bathy-morphological map and (b) ~~slope~~ Slope map of the ~~upper~~ Upper portion of the ~~continental slope~~ Continental Slope (UCS) of the Calabrian-Ionian margin. Red square indicates the intraslope flat area hosting the circular high of the Bortoluzzi Mud Volcano (BMV). Location map is shown in Fig. 2a.

**Figure 3.** (~~black and white~~a) Bathy-morphological map and ~~beach balls, respectively. Focal mechanisms (Table S2) are from the European-Mediterranean Regional Centroid-Moment Tensors (red beach ball) catalog (http~~main morphological features: ~~//rcmt2.bo.ingv.it/)~~ ~~and from previous papers (blue beach balls) by Orecchio et al. (2014) and Polonia et al. (2016). White dots represent the selected seismic events at distance of 12 km to be projected along the transect A-A' (blue line). Grey thick and grey dashed lines corresponds to the CROP M-31 (Fig. 10b) and the CA99-215~~ canyon head ~~(Fig. 10aCh)multichannel profiles, respectively. The black dashed box is the location of Fig. 3.~~ gullies (~~bG)A-A' vertical cross-sectional transect~~, slide scars (~~see track in Fig. 2aSs)showing a perpendicular projection~~, regional scarps (~~onto the transectRs)of earthquake hypocenters~~, and ~~focal mechanisms for the BMV~~ a 26 km² flat area ~~. The blue line is the corresponding seafloor profile, over which the projected location of the BMV~~ (~~red triangleFa) is shown~~are indicated.

(a) Bathy-morphological map and (b) ~~slope~~ Slope map of the ~~upper~~ Upper portion of the ~~continental slope~~ Continental Slope (UCS) of the Calabrian-Ionian margin. Red square indicates the intraslope flat area hosting the circular high of the Bortoluzzi Mud Volcano (BMV). Location map is shown in Fig. 2a.

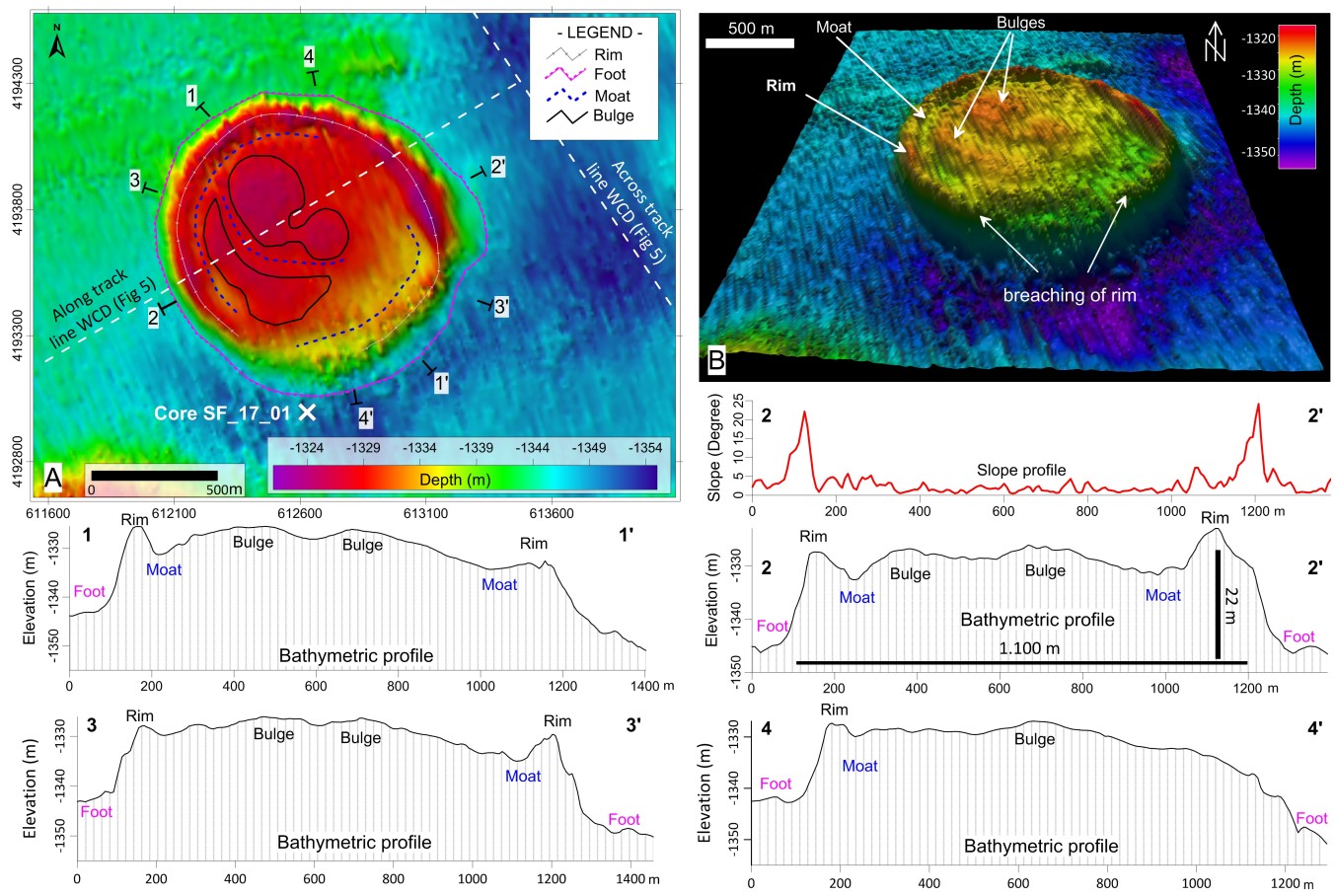

**Figure 4.** (a) Shaded relief map of the BMV with the main morphological features and location of bathymetric sections (~~a-a~~1-1', 2-2', 3-3', and ~~b-b~~4-4'). The ~~map is accompanied by two~~ bathymetric ~~profiles~~ sections (~~bottom left and bottom right~~vertical exaggeration 10 x) ~~along show~~ the ~~a-a' and b-b' tracks~~ morphological features of the mud volcano with vertical slopes up to 28° and two main concentric bulges separated by ~~one slope profile (middle right) along the b-b' track~~moats. (b) ~~Digital~~ 3D perspective view of the BMV showing some main ~~morphological~~ features and the breaching of the rim on the southern part of the BMV. In Fig. 4a, the track position (along and across) of the water column data (WCD) are shown (see Fig. 5). See also the Movies S1 and S2.

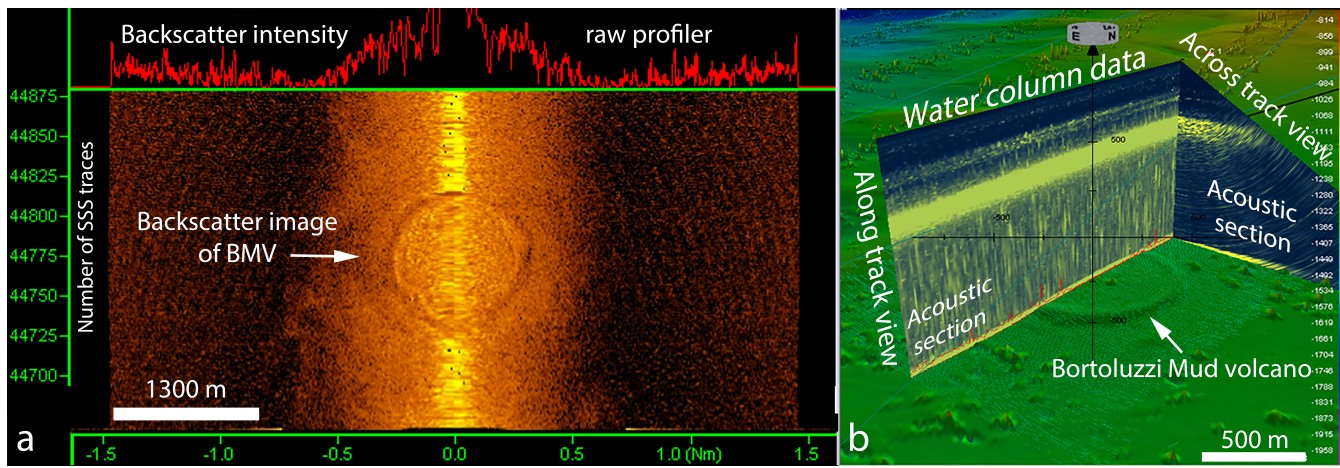

**Figure 5.** (a) ~~Raw profiler~~ Multibeam raw backscatter ~~of~~ data. The raw backscatter data does not show amplitude anomalies on the ~~BMV area~~top and around the BMW, indicating a mud homogeneous sedimentary cover. (b) ~~Acoustic backscatter of the~~ Backscatter water column ~~above the BMV~~data recorded by multibeam system. ~~These~~ The backscatter water column data ~~does not~~ show ~~the substantial absence of~~ amplitude anomalies ~~and therefore massive~~ above the mud volcano excluding significant paroxysmal fluid~~escapes /~~mud escape from the seafloor. Location tracks of the acoustic sections are shown in Fig. 4.

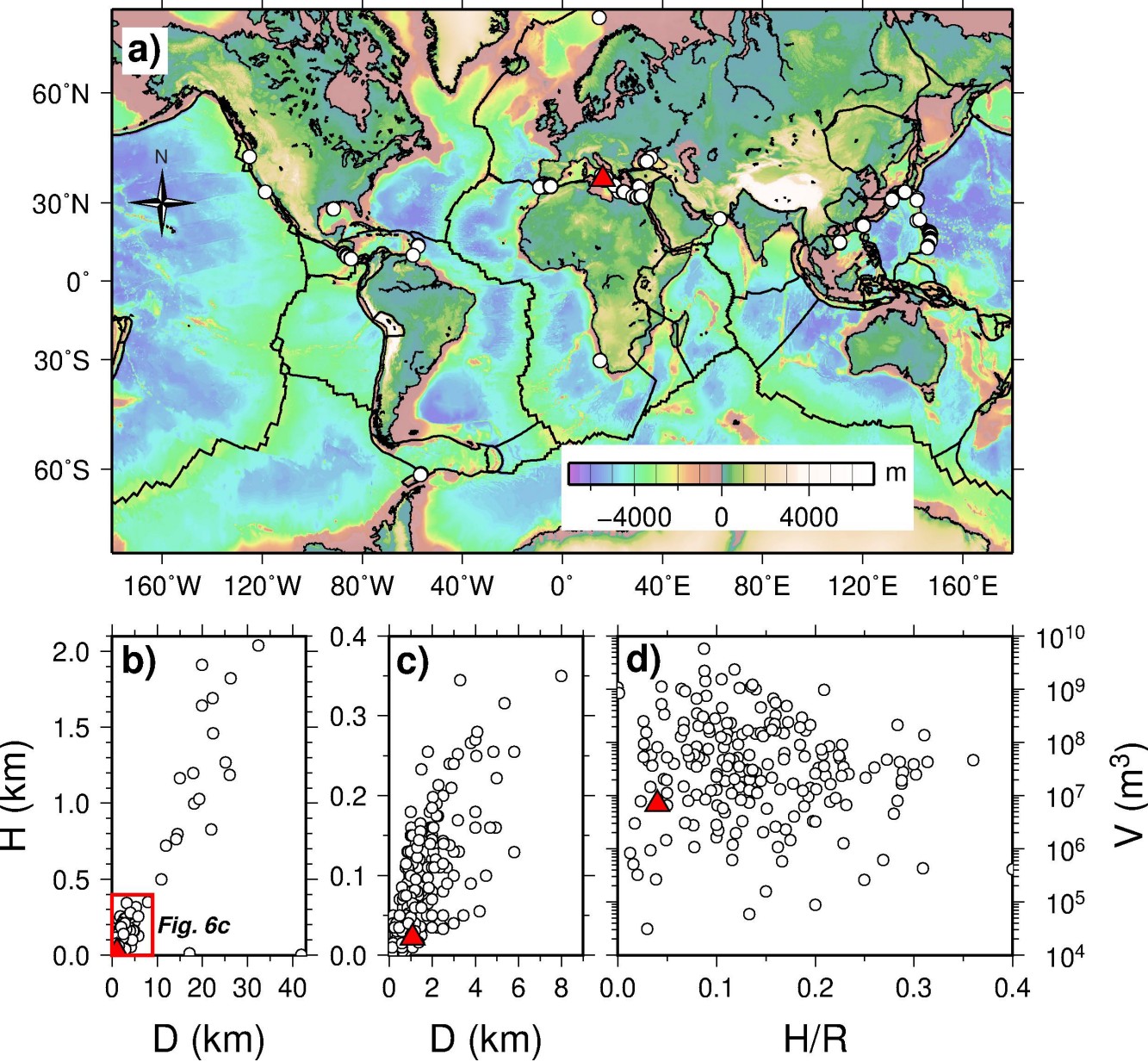

**Figure 6.** (a) Selection of worldwide distribution of mud volcanoes (white dots) from the dataset reported by Kioka and Ashi (2015). In particular, 232 volcanoes with available mean diameter ~~D~~ $D$ and height ~~H~~ $H$ are shown. The red triangle is the BMV location. (b) Compilation of mud volcanoes diameter ~~D~~ $D$ vs. height ~~H~~ $H$, showing an approximately linear trend between increasing mud volcano diameters $(D)$ and heights $(H)$, excluding some exceptions. Along this trend, the BMV (red triangle) stands close the lower values. (c) Close-up view of a portion of the diagram in b. (d) Compilation of volumes ~~V~~ $V$ of mud volcanoes vs. ~~H/R~~ $H/R$ (where ~~R~~ $R$ is the radius of the volcano base), showing that the ~~H/R~~ $H/R$ ratio is $\leq 0.4$ for all mud volcanoes, whereas there is a scatter distribution of mud volcano volumes, mostly ranging in the ~~106-109~~ $10^6$-$10^9$ m$^3$ interval. The BMV volume (red triangle) corresponds to 6.9 ~~x106~~ $x10^6$ m$^3$.

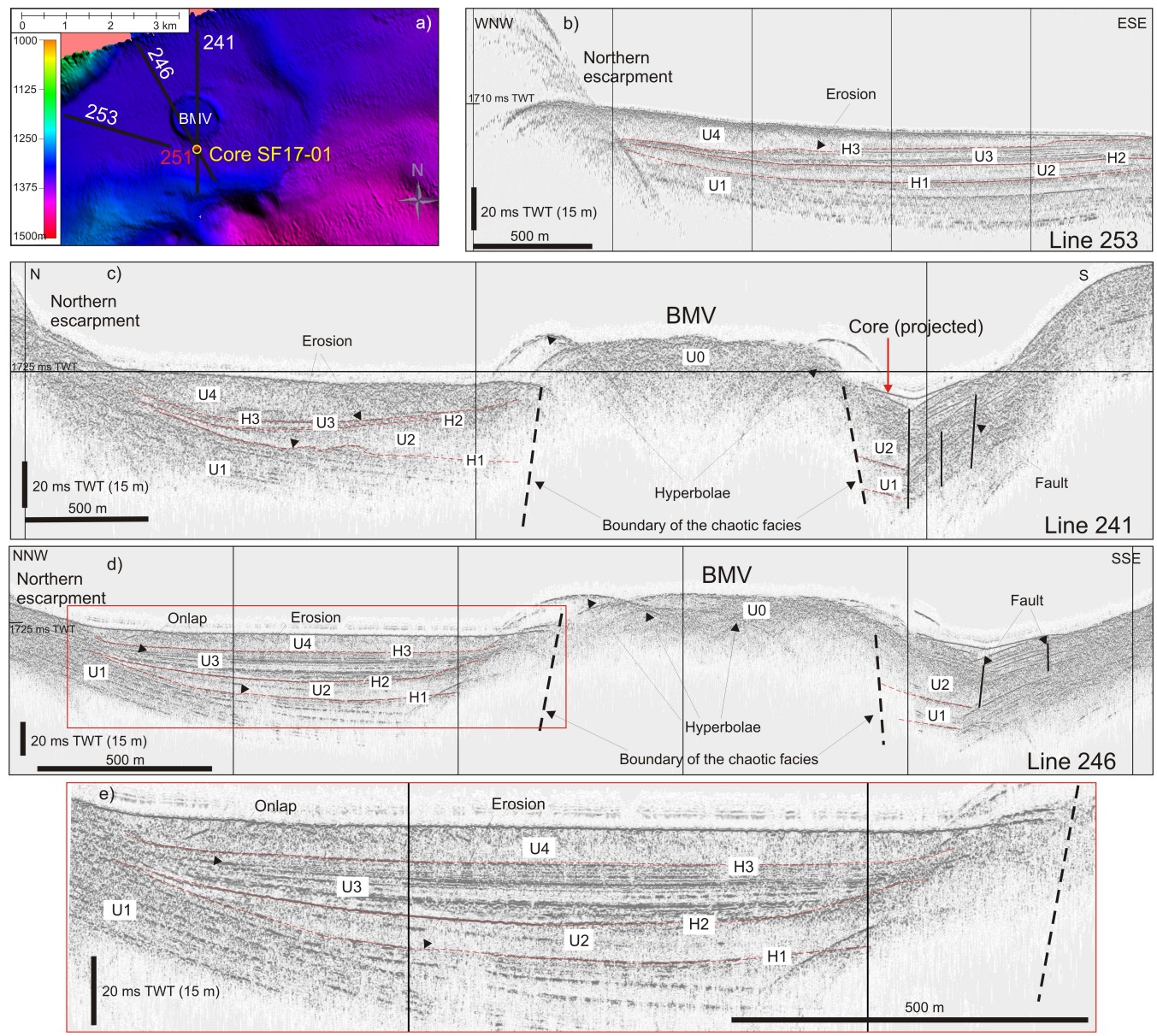

**Figure 7.** (a) Location map of three single-channel chirp profiles and one gravity core in the BMV area. The red dot (numbered 251) below the location of the gravity core indicates the location of the single-channel chirp profile 251 shown in Fig. 9. (b) Line 253. (c) Line 241. (d) Line 246. (e) Enlargement of Line 246 (see the red rectangle in Fig. 7d). Chirp profiles show main seismic units (U0-U4) and bounding horizons identified in the shallow subsurface. The high-resolution non-interpreted version of this figure is available in the Supplement (Fig. S1).

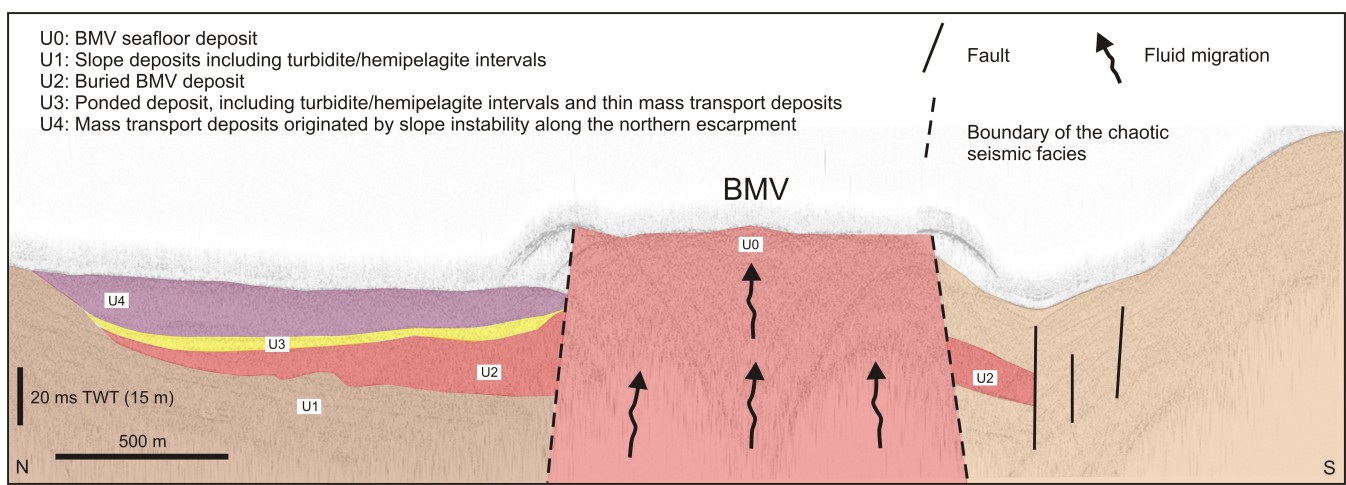

**Figure 8.** Interpretation of the chirp profile 241 shown in Fig. 7 (see text for details).

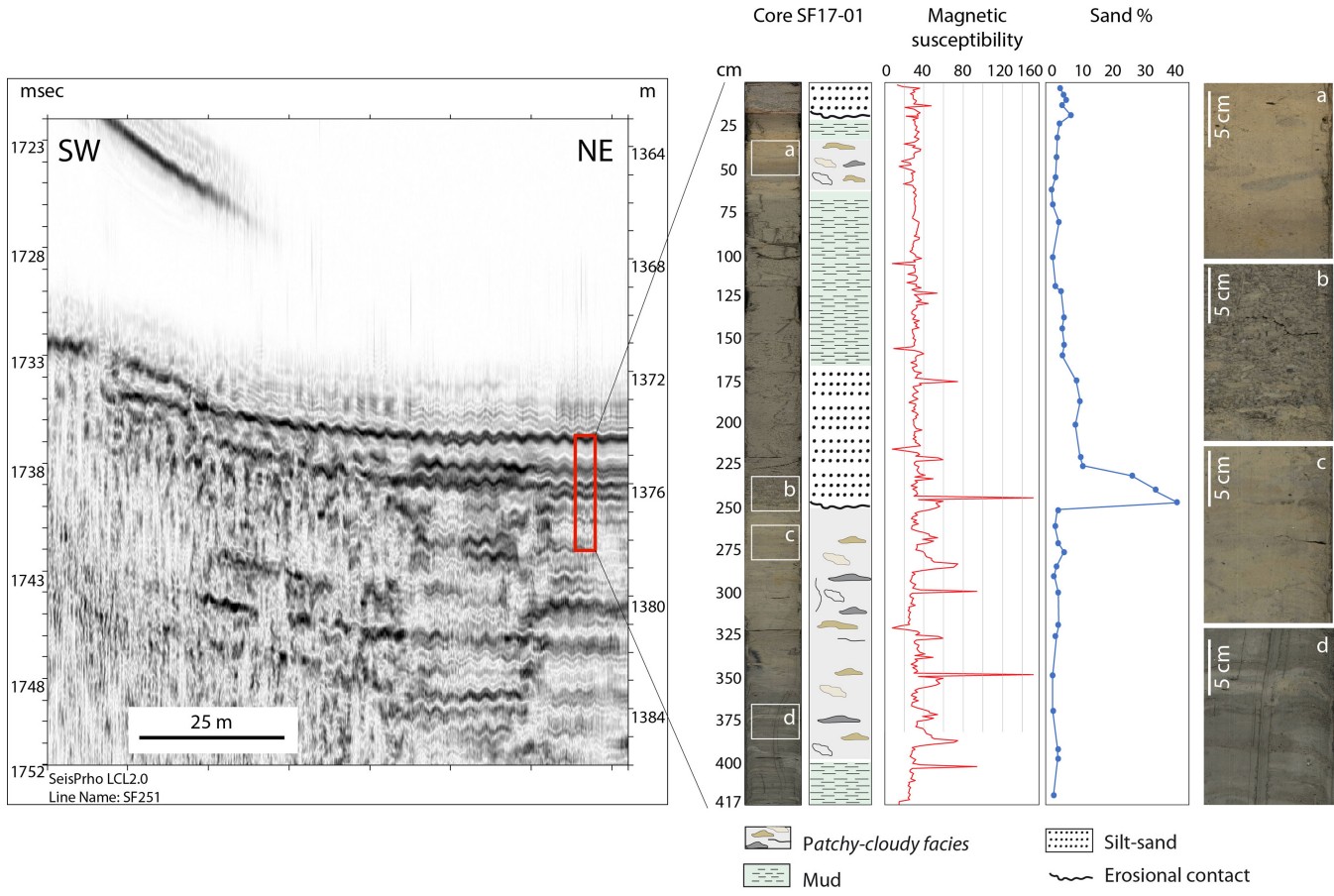

**Figure 9.** Correlation between the ~~SF17-01 gravity core~~ 251 chirp profile (~~right~~left) and the ~~251 chirp profile~~ SF17-01 gravity core (right). See locations in Fig. 7. Left: 251 chirp profile collected during coring operations. Right: photograph, lithological log, high-resolution magnetic susceptibility, and sand content of the SF17-01 gravity core. Magnetic susceptibility is rather constant throughout the core with the exception of some peaks in the lower part of the core, where the patchy-cloudy facies (*sensu* Staffini et al., 1993) is present. A peak in magnetic susceptibility marks the abrupt increase in sand content at the base of the resedimented deposit between 75 and 250 cm. a, b, c, and d represent close-ups of different core units: a, c, and d are patchy/cloudy facies whereas b is the base of the resedimented unit.

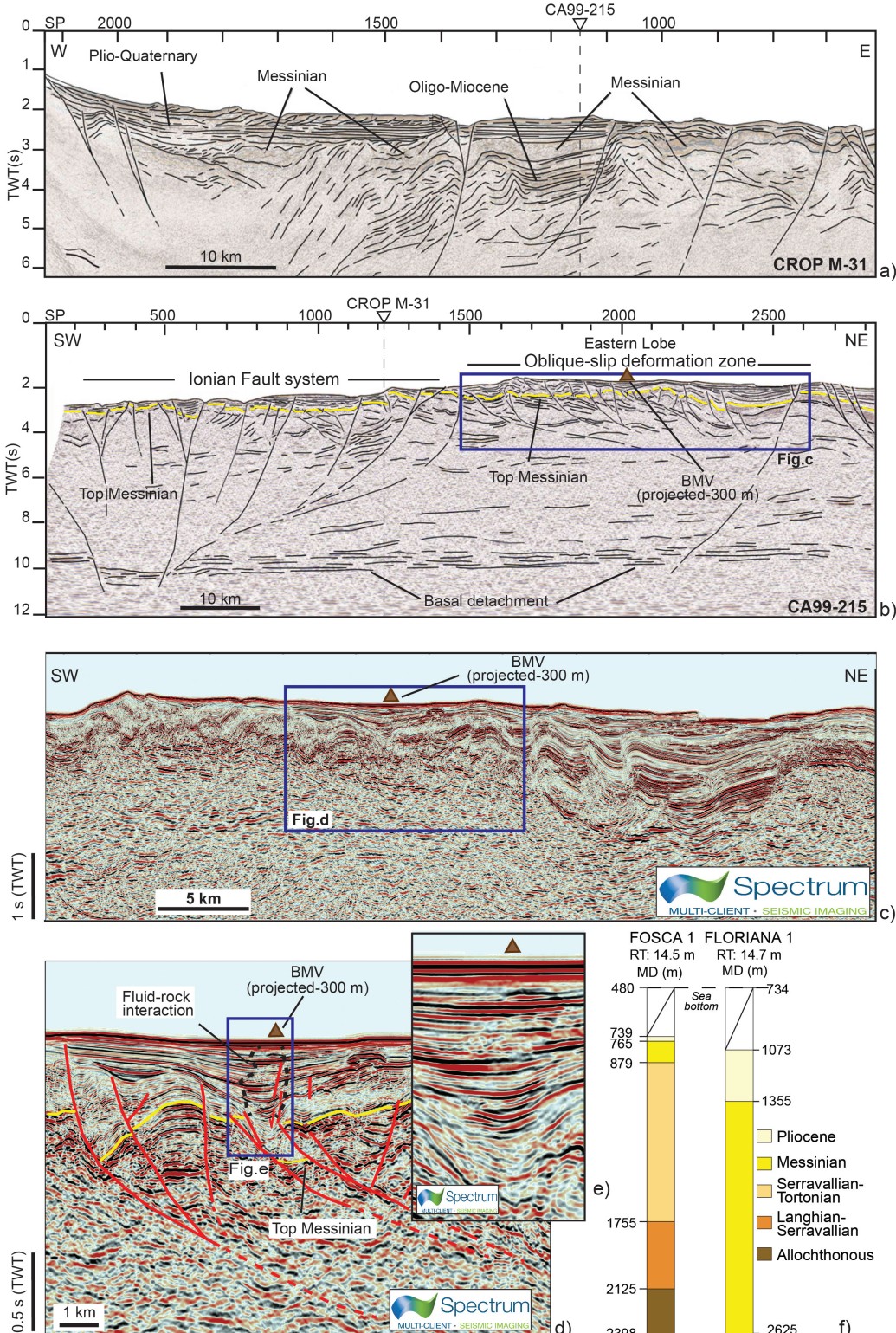

**Figure 10.** (a) CROP M-31 seismic reflection profiles (track in Fig. 1) in the version interpreted by Polonia et al. (2016), showing the complex structural setting and the main interpreted deposits. The CROP M-31 profile crosses the CA99-215 profile at s.p. 1150. (b) CA99-215 seismic reflection profiles (track in Fig. 1) in the version interpreted by Polonia et al. (2016). The top of the Messinian deposits is reported in yellow.

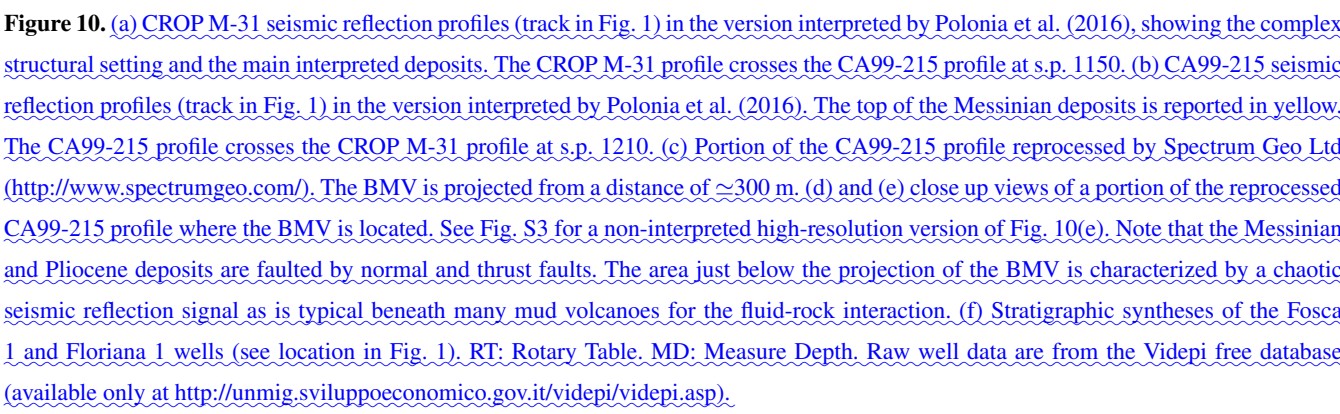

**Figure 10.** (a) CROP M-31 seismic reflection profiles (track in Fig. 1) in the version interpreted by Polonia et al. (2016), showing the complex structural setting and the main interpreted deposits. The CROP M-31 profile crosses the CA99-215 profile at s.p. 1150. (b) CA99-215 seismic reflection profiles (track in Fig. 1) in the version interpreted by Polonia et al. (2016). The top of the Messinian deposits is reported in yellow. The CA99-215 profile crosses the CROP M-31 profile at s.p. 1210. (c) Portion of the CA99-215 profile reprocessed by Spectrum Geo Ltd (http://www.spectrumgeo.com/). The BMV is projected from a distance of $\simeq$300 m. (d) and (e) close up views of a portion of the reprocessed CA99-215 profile where the BMV is located. See Fig. S3 for a non-interpreted high-resolution version of Fig. 10(e). Note that the Messinian and Pliocene deposits are faulted by normal and thrust faults. The area just below the projection of the BMV is characterized by a chaotic seismic reflection signal as is typical beneath many mud volcanoes for the fluid-rock interaction. (f) Stratigraphic syntheses of the Fosca 1 and Floriana 1 wells (see location in Fig. 1). RT: Rotary Table. MD: Measure Depth. Raw well data are from the Videpi free database (available only at http://unmig.sviluppoeconomico.gov.it/videpi/videpi.asp).

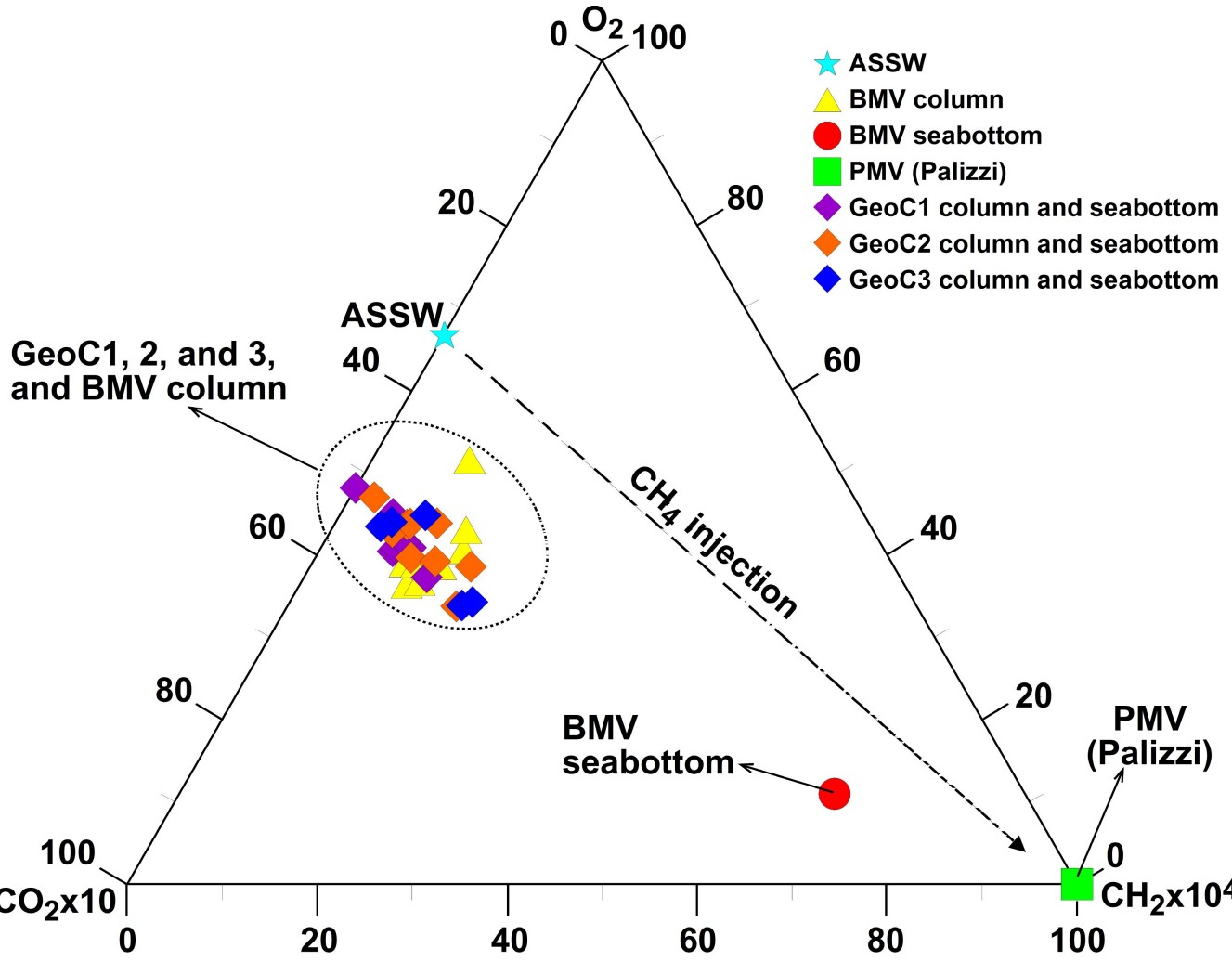

**Figure 11.** $CH_4$-$O_2$-$CO_2$ diagram. The ternary plot shows relative distribution of dissolved gases for the collected samples. Dissolved gases composition coming from the sampling site above the BMV are compared with the local air-saturated sea water (ASSW). The figure shows typical endogenic components ($CO_2$ and $CH_4$) versus the atmospheric component (here represented by $O_2$).

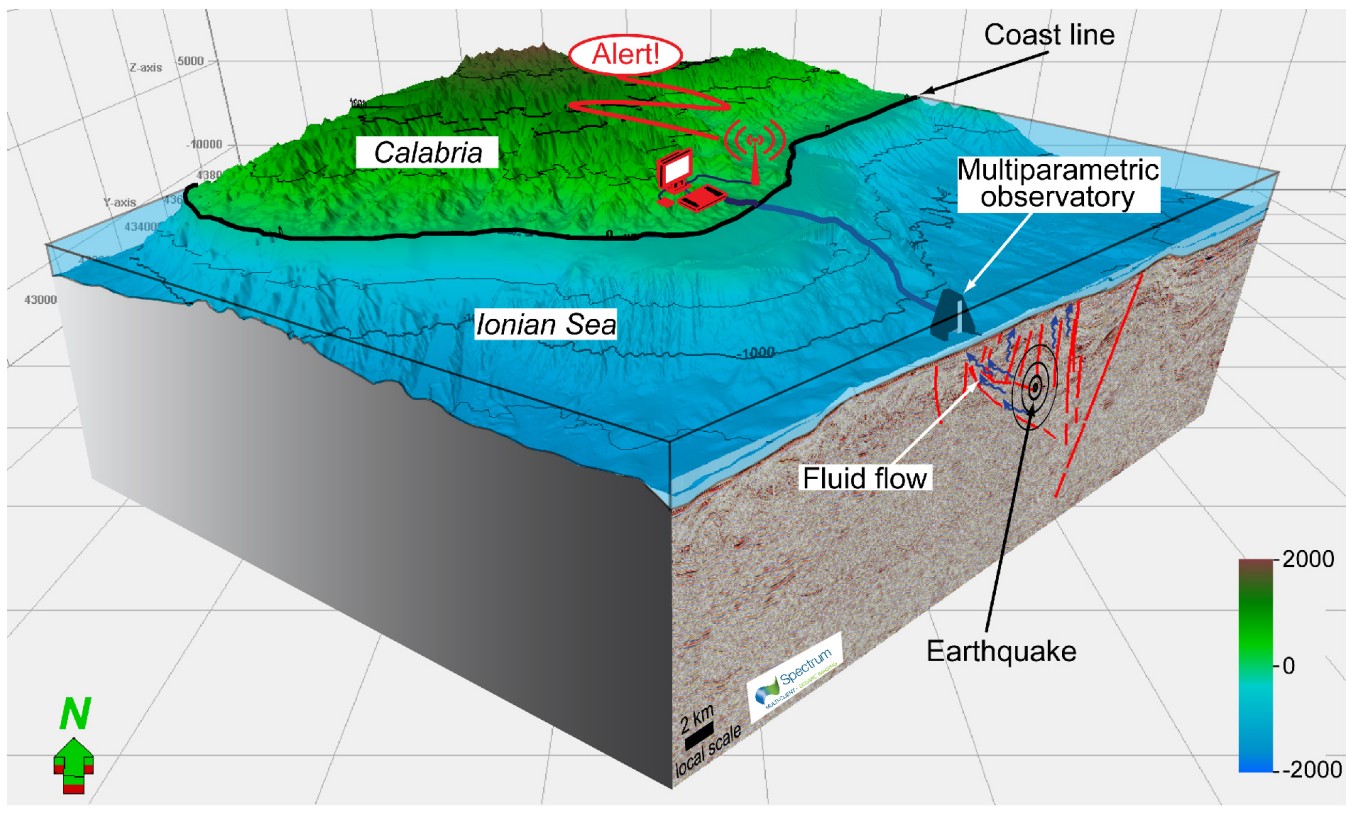

**Figure 12.** Conceptual scheme showing a future multiparametric cabled station on the BMV to geochemically tracking the seismic cycle of the underlying active faults and so to contribute to the mitigation of the seismic hazard.

| Sample | date | depth(m) | Latitude | Longitude | T°C | pH | Na(meq/L) | K(meq/L) | Mg(meq/L) | Ca(meq/L) | Cl(meq/L) | Br(meq/L) | $SO_4$(meq/L) | $HCO_3$(meq/L) |
|---|---|---|---|---|---|---|---|---|---|---|---|---|---|---|
| GeoC1 | 05.17.2017 | -1043 | 37°57'24.6" | 15°28'29.4" | 13.80 | 8.16 | 553 | 12.7 | 125 | 24.9 | 644 | 0.930 | 67.1 | 2.68 |
| GeoC2 | 05.16.2017 | -1649 | 37°43'10.8" | 15°26'40.8" | 13.79 | 8.37 | 552 | 12.7 | 125 | 25.2 | 637 | 0.999 | 66.5 | 2.68 |
| GeoC3 | 05.16.2017 | -2029 | 37°28'10.8" | 15°23'01.8" | 13.82 | 8.10 | 554 | 12.8 | 126 | 25.6 | 641 | bdl | 66.1 | 2.70 |
| BMV | 05.18.2017 | -1337 | 37°52'38.1" | 16°16'50.1" | 13.76 | 8.10 | 553 | 12.6 | 125 | 25.3 | 643 | 1.38 | 66.0 | 2.70 |
| | | -1000 | | | 13.79 | 8.09 | 551 | 12.6 | 125 | 25.5 | 633 | 1.04 | 63.5 | 2.72 |
| | | -500 | | | 14.25 | 8.16 | 555 | 12.6 | 125 | 25.5 | 645 | 1.04 | 65.8 | 2.70 |
| | | -200 | | | 14.88 | 8.06 | 555 | 12.8 | 126 | 25.5 | 643 | 1.75 | 65.9 | 2.66 |
| | | -50 | | | 15.91 | 8.07 | 552 | 12.7 | 125 | 25.5 | 638 | 1.09 | 66.0 | 2.71 |
| Palizzi PMV | 10.06.2003 | onland | 37°57'51.02" | 15°59'28.43" | na | na | 15.06 | 0.09 | 0.01 | 0.09 | 7.23 | bdl | 3.60 | na |

Table 1: Chemical composition of the sea water from the studied localities in the Ionian Sea (GeoC1, 2, and 3, and BMV). The chemical composition of the fluids sampled in a well adjacent to the Palizzi mud volcano (PMV) in the Calabria main land is also reported.

| Site | depth(m) | He(meqcc/L) | O$_2$(meqcc/L) | N$_2$(meqcc/L) | CH$_4$(meq/L) | CO$_2$(meqcc/L) | R/Ra | He/Ne |
|---|---|---|---|---|---|---|---|---|
| GeoC1 | 1000 | 7.14E-05 | 3.95 | 9.31 | 1.37E-04 | 5.28E-01 | 0.94 | 0.30 |
| | 900 | 6.11E-05 | 3.97 | 9.64 | 7.62E-05 | 5.10E-01 | 1.07 | 0.27 |
| | 800 | 7.74E-05 | 3.75 | 8.85 | 7.23E-05 | 4.73E-01 | 0.94 | 0.27 |
| | 700 | 6.86E-05 | 3.95 | 9.30 | 9.14E-05 | 4.80E-01 | 1.01 | 0.24 |
| | 500 | 5.51E-05 | 4.16 | 9.45 | 6.86E-05 | 4.73E-01 | 0.93 | 0.27 |
| | 300 | 5.56E-05 | 4.10 | 9.40 | 8.38E-05 | 4.95E-01 | 1.00 | 0.24 |
| | 200 | 6.35E-05 | 4.58 | 9.73 | 5.61E-05 | 5.04E-01 | 0.88 | 0.27 |
| | 100 | 6.61E-05 | 4.41 | 8.90 | bdl | 4.75E-01 | 0.93 | 0.30 |
| GeoC2 | 1649 | 6.01E-05 | 3.93 | 9.35 | 2.07E-04 | 5.64E-01 | 0.77 | 0.28 |
| | 1500 | 5.84E-05 | 4.23 | 8.98 | 1.07E-04 | 5.37E-01 | 0.92 | 0.26 |
| | 1300 | na | 3.77 | 8.98 | 1.24E-04 | 4.62E-01 | na | na |
| | 1000 | na | 4.24 | 9.62 | 2.30E-05 | 4.56E-01 | na | na |
| | 700 | na | 4.04 | 9.99 | 1.77E-04 | 4.67E-01 | na | na |
| | 600 | na | 3.94 | 9.21 | 6.90E-05 | 4.39E-01 | na | na |
| | 500 | na | 3.79 | 8.79 | 5.82E-05 | 4.50E-01 | na | na |
| | 300 | na | 4.24 | 9.48 | 7.67E-05 | 4.67E-01 | na | na |
| | 100 | na | 4.38 | 9.10 | 1.07E-04 | 4.54E-01 | na | na |
| GeoC3 | 2029 | 8.73E-05 | 3.98 | 9.20 | 5.33E-05 | 4.54E-01 | 0.766 | 0.422 |
| | 1500 | 6.75E-05 | 3.51 | 9.85 | 1.90E-04 | 4.97E-01 | 0.950 | 0.290 |
| | 1000 | 7.17E-05 | 3.99 | 9.34 | 4.57E-05 | 4.74E-01 | 0.927 | 0.366 |
| | 500 | 7.13E-05 | 3.61 | 9.33 | 2.02E-04 | 4.90E-01 | 0.976 | 0.289 |
| | 100 | na | 4.54 | 9.30 | 9.14E-05 | 4.68E-01 | na | na |
| BMV | 1337 | 8.27E-05 | 3.99 | 9.12 | 2.51E-03 | 7.31E-01 | 0.73 | 0.398 |
| | 1200 | na | 3.94 | 9.33 | 1.22E-04 | 5.67E-01 | nd | na |
| | 1000 | 7.70E-05 | 4.08 | 9.90 | 1.45E-04 | 5.03E-01 | 0.92 | 0.256 |
| | 800 | na | 3.97 | 9.37 | 9.90E-05 | 5.22E-01 | na | na |
| | 700 | na | 3.76 | 8.88 | 1.07E-04 | 4.85E-01 | na | na |
| | 500 | na | 3.87 | 8.60 | 1.45E-04 | 4.23E-01 | na | na |
| | 300 | na | 3.97 | 9.75 | 1.35E-04 | 5.50E-01 | na | na |
| | 200 | na | 4.61 | 9.58 | 1.52E-04 | 4.59E-01 | na | na |
| | 100 | na | 6.47 | 1.59 | 1.29E-04 | 4.80E-01 | na | na |
| PMV | onland | 2.62E-03 | bdl | 22.28 | 6.87E+00 | 1.23E-01 | 1.60E-01 | 17 |
| ASSW | | 4.80E-05 | 4.80 | 9.60 | 1.00E-06 | 2.40E-01 | 1 | 0.267 |

Table 2: Dissolved gas composition and helium isotopic ratio of the sea water from the studied localities in the Ionian Sea (GeoC1, 2, and 3, and BMV). Data for the fluids sampled in a well adjacent to the Palizzi mud volcano (PMV) in the Calabria main land are also reported.