# Peer review of "The Bortoluzzi Mud Volcano (Ionian Sea, Italy) and its potential for tracking the seismic cycle of active faults"

_Solid Earth, 2018_

## Referee Comment (RC1) · Anonymous Referee #1 · 29 Jan 2019

The manuscript 'The Bortoluzzi Mud Volcano (Ionian Sea, Italy) and its potential for tracking the seismic cycle of active faults' presents a multi-parameter dataset of a mud volcano in the Ionian Sea. The authors consider this mud volcano as a promising location for installing a cabled multiparamteric station in order to study the relationship between the seismic cycle of underlying active faults and fluid expulsion. The data set is definitely interesting and interpretations seems to be sound (though not always easy to verify based on the presented data, see below); hence, I principally support publication of the manuscript but suggest major modifications, especially concerning the structure of the manuscript and the data presentation. In addition, the authors should elaborate more on the general significance of their study, or rather demonstrate

more clearly, why the BMV is representative and best suited for the suggested investigations. I am not an expert on geochemistry and cannot comment this section of the manuscript.

General comments: 1) I strongly suggest to change the structure of the manuscript. Separate 'Material and Methods' from the 'Results'. In addition, you mix results and discussion quite a bit. Introduce the methods first and then summarize the results. You state that you have a multi-parameter data set. Make use of this opportunity and try to combine the data sets already during presentation of the results. Morphology and subsurface features (seismic data) can be combined. The second major type of data are the geochemical data. The integrated presentation of the data would be the perfect basis for the discussion. The main points of the discussion should then be the (1) Origin and Activity, and (2) the potential to use the BMV as location to investigate the relationship between the seismic cycle of underlying active faults and fluid expulsion.

2) It remains unclear why you selected the BMV for the investigations. How does it compare or differ to other mud volcanoes in this area. Mud volcanoes in the area haven been investigated before. Not much information is given on previous investigations of mud volcanoes in the Ionian Sea.

3) This previous point is closely related to the general relevance of your investigation. What is special compared to previous study? How does your investigation contribute to the general understanding of mud volcanos? Why is the BMV more than a case study?

4) The conclusions are mainly a list of your most important results, which are already listed in the abstract. A conclusion should tell the reader what she can or he could do with the newly acquired knowledge. Answer the question "So what?".

5) Several of your statements cannot be verified based on the presented figures. I am aware that you cannot show every detail but some enlargements/details in order to support your statements would be useful (see specific comments)

6) The geochemical analyses of the BMV is based on one cast above the BMV. I am not a geochemist but I wonder how representative this cast is?

7) Annotations of several figures are very small and difficult to read.

8) Avoid extensive referencing. References are always a selection; you cannot mention all, especially for a topic, where quite a lot of work has already been published. I suggest not using more than three references for one statement.

9) I am not a native English speaker but I have the impression that some (little) language polishing is needed. I have not made any specific comments on this point.

Specific comments:

Page 1, Line 14/15: Conclusions are drawn . . . Give specific conclusions. Do not only announce them.

Page 2, Line 5/6. This is an example for extensive referencing.

Page 2, Lines 8/9: Another example for excessive referencing. Not mentioned any more in the following. Check for the entire text.

Page 2, 14/15: 27 Mt methane a-1 . . .Is this the only estimate? I assume that estimates cover a wide range of numbers.

Page 2, Line 15/16. Calabrian arc is not marked on Fig. 1.

Page 2, Line 29/30: It seems that you do not use the OBS/OBH data in the manuscript. Why do you mention them here? Alternatively, why do you not use them for the manuscript?

Page 3, Line 4/5: Be more specific. Name some of the potential precursors.

Page 3, Line 26: Mark Aeolian Islands on Fig. 1

Page 3, Line 31: Mark escarpments on Figure 1. If the figure is getting too crowded, an additional figure showing all the locations mentioned in the text may be useful.

Page 5, Lines 6-13: You mentioned OBS/OBH deployments in the introduction. Why are these data not included? Local seismicity would be interesting. Or, are they still on the sea floor?

Page 5, Line 15. What is the extend of the BMV area? Is it the area shown on Fig. 1? Clarify.

Page 5, Line 15, 16. You explicitly state that the focal mechanisms are from earthquakes that are < 4.5 in magnitude. This implies that there are also larger earthquakes. What is about larger earthquakes?

Page 6, Lines 1-4. Processing details should be moved to the supplement or deleted, especially as the data have already been presented elsewhere (Bosman et al., 2015)

Page 6, Line 5/6: change to 'with a lateral resolution varying between 10 m for water depths shallower than 1000 m to 25 m at greater depth'.

Page 6, Lines 6-12: Delete. No need to include such details. You do not need to show that you know procedures for calibrating a multibeam system.

Page 6, Line 17, 18: This sentence reads very strange. First, you state that 22° is a maximum. Then you continue that it is even up to 28°. I assume that the 28° is related to the 35 km long escarpment. Maybe split to two sentences, though I am not sure what you mean. Suggestion: 'The upper part of the continental slope is characterized by a very steep slope (about 15°) that reaches a maximum of 22°. Even larger slope gradients of up to 28° are found along the NE-SW orientated main escarpments, which are up to 35 km long.

Page 6, Line 24/25. Difficult to follow. In your Fig. 4a, you show one rim. In this sentence, you mention two rims. This is confusing.

Page 6, Line 32 – Page 7, Line 11: More discussion. If you include the BMV in this discussion, you should mention the volume during the description. It also remains unclear if you calculate the volume of the BMV based on the given formula of if you use

exact volumes based on your data.

Page 7, Line 17. I strongly suggest converting the TWT to depth and resolution in meters using a constant velocity of 1500 m/s.

Page 7, Line 27-30: I like that you have unintepreted sections in the supplement. This give you the chance to add more interpretation to the interpreted sections. E.g., H1 is easy to identify at the location where you label H1. I find it difficult to trace H1 from these locations to the sides. In addition, erosion at H1 is difficult to see. Mark this clearly or show enlargements to support this statement.

Page 7, Line 31 to Page 8/Line 10: Similar comment as before. Mark lower and upper boundary more clearly. Mark features described in the text on the figures more clearly (Erosion, onlaps and so on). Make it easier for the reader to follow your description. In addition, I strongly suggest using depth/thickness of units in meters (possibly in addition to numbers in TWT).

Page 8, Lines 13 to 30: This is interpretation and I would suggest moving this to the discussion section. Sometimes, there are good reasons to leave this type of interpretation with the data description but the general structure of the chapters (methods, results, interpretation) is challenging the approach of an integrative interpretation of your multi-parameter data set.

Page 8, Lines 21-24: Do you have a preferred interpretation based on your data. Delete the figure numbers of the Evans et al. paper. It is sufficient just to reference the paper.

Page 9, Line 9. How sure can you be about the location of the core. Did you use a positioning system (e.g. USBL-System) during coring, or is it the surface location of the vessel? The chirp profiles show large lateral variability at the projected coring location. How sure can you be that you really penetrated the units imaged at the projected location of the core?

Page 9, Line 9 to 23: This is again mixing results and interpretation/discussion, even

more than before. E.g., you describe the patchy/cloudy facies when discussing it. I strongly suggest to give a good description of the core first and discuss it afterwards (here or even better in the discussion section).

Page 9, Lines 26-31: Give the basic acquisition parameters (such as source frequency in order to estimate the resolution). You give very detailed information for the other systems but almost no information is given for the seismic system.

Page 10/11, Results: It is again a mixture between description and discussion.

Page 10. Line 13/14: The Ionia fault system is labelled from SP200 – 1400 on the CROP M-31 line (Fig. 10b), which contrasts the statement in the text (SP 1100 -1400) Clarify.

Page 10, Lines 24/25. Mark the flower structures. It is clear on the western side of the profile but not obvious for the eastern side.

Page 11, Lines 4-8. I do not think that I see the slightly transparent seismic reflection signal beneath the BMV. I may see some small amplitude variations but they seem to occur along the entire profile. In addition, I also see some areas with enhanced reflection amplitudes beneath the BMV (e.g. in the transparent unit directly beneath the seafloor reflector in Fig. 10e). Strong amplitudes also indicate rock/fluid indication. A better description and a better visualization is required in order to believe your statement (indication for fluid-rock interaction).

Page 11, Line 12 – Page 12, Line 10: Cannot judge this part. Seems to be quite long for me. Isotopic numbers are not superscript (in the entire text).

Page 12, Line 17. Change 'comparable' to 'similar'. Everything is comparable.

Page 12, Line 18, 19: This statement seems to be wrong. The Br content for GeoC2 is 0.990, and 1.04 in 500 and 1000 m depth for the BMV. This s less than 11° higher. The highest Brome content is found in 200 m depth at BMV. This value is very unlikely to be related to the BMF because it is far above the BMV. How reliable is the statement, that

the Brome content is really higher. Can you make this statement on one single cast?

Page 12, Line 23, 24. Not clear where you see the decrease in oxygen content. Be more specific.

Page 12, Line 27. 0.73 (not 0.71 as in the text) is given for the $CO_2$ content in the table.

Page 13, Line 7 to Line 18. Delete. This section is not containing relevant information.

Page 13, Line 20-26.As mentioned above, some parts of the result section would be located here much better. Otherwise, this section is just a repetition of the results. As mentioned above, the semi-transparent seismic facies is not evident for me.

Page 13, Line 17-19: As mentioned above. How representative is a single sample? I believe that higher $CH_4$ content is significant but convince the reader.

Page 13, Line 28. See comments concerning the semi-transparent facies above.

Page 15, Lines 17-20. Delete. No relevant information.

Page 15, Line 29. Mention how long before the earthquake the pore pressure changes have been measured.

Page 15, Line 22 - Page 16, Line 15. These are nice examples but this chapter reads as earthquake forecasting can easily be done by geochemical monitoring. Add a few sentences that the story is not that easy.

Page 16, Line 16-33: Principally, I agree but I am not convinced that the BMV is really the best place just because it is best characterized by the most extensive data set. I assume that there are some mud volcanoes with proven ongoing activity close by (escape of fluids, mud flows etc.). Why not using another mud volcano? What is special about the BMV?

Page 17, Lines 1-11: Delete here. Include in a rewritten conclusion.

Page 17, Conclusion. See general comments. Answer the question "So what?"

[Figure]

Fig. 1: Nice figure with a lot of information. Try to add additional geographic locations and features mentioned in the text. Split to two figures if the figure is getting too crowded. I suggest deleting the earthquake locations from the figure because they are shown much better in Figure 2 and you only show the earthquakes in the area of Fig. 2. The grey dashed line (CA99-215 line) is very difficult to see.

Fig. 2: What is the difference between the solid and the blue dashed line? It seems for me that the dashed blue line is not needed.

Fig. 3: Figure should be larger. Add to figure caption 'Location of the maps is shown on Fig. 2a'.

Fig. 4: Figure should be larger. Annotations/Numbers are very small and difficult to read. Add distance scale to maps. Moat not labelled on the SE part of Profile a a'.

Fig. 4: Distance scale is missing on maps.

Fig. 5. Mark location of profile on Fig. 4. Add along track scale. Figure caption is confusing. It is not clear that 'absence' relates to both statements. Now it reads as the absence of amplitude anomalies indicates massive fluid escapes. Modify.

Fig. 6: Exponents are not in superscript for the volumes.

Fig7: See comments above. Add more interpretation and/or close-ups. Add absolute depth for the y-axis. I suggest using meters instead of TWT.

Fig. 8: Nice figure. Add absolute depth on Y-Axis. Would use meters instead of TWT.

Fig. 9: I have serious problems to correlate the enlargements of the core section with the photo of the entire core. E.g., section b should include the erosional contact. I can see a clear colour change at this interval in the image of the entire core but I cannot see this colour change or the erosional contact in the enlargement b. Enlargement b looks very homogenous for me. The same is the case for enlargement d, which also show clear colour changes. Some sedimentary layers seems to be down bended on the

left-hand side of the image of the entire core, which is not visible on the enlargement. Annotations are very small and almost impossible to read.

Figure 10. Nice figure. See comments made above (especially concerning the slightly transparent seismic facies).

Fig. 11: OK

Fig. 12: Nice figure but nor really discussed in the text. Discuss the outline of your monitoring tools in more detail.

Good luck with the revisions.

---

## Author Comment (AC1) · 30 Jan 2019

With this comment, we would like to acknowledge that the seismic reflection profile appearing in Figs. 10c-e, 12, and S3 was reprocessed by Spectrum Geo Ltd (www.spectrumgeo.com). For mere formal error, the Spectrum logo was not displayed in these figures. Therefore, new PDF files for both the main manuscript and the supporting material have just been uploaded by the Editorial Staff of Solid Earth and are now available in the Interactive Discussion. These new files show the correct version of Figs. 10c-e, 12, and S3 (i.e., with the Spectrum logo). These figures are the only items changed as regards the previous version. We apologize for the inconvenience.

[Figure]

Andrea Billi and co-authors

---

## Referee Comment (RC2) · Anonymous Referee #2 · 4 Mar 2019

Summary

The author presents an examination of the here named Bortoluzzi Mud Volcano (BMV) with the aim of demonstrating its potential for tracking the seismic cycle of active faults. The analysis is undertaken through the integration of numerous geological, geochemical and geophysical data. The primary conclusion is that the BMV could present a suitable site for installation of a cable submarine multiparametric station to study the potential relationship between subsurface vertical fluid migration and the seismic cycles of active faults. The manuscript offers an impressive integration and analysis of various data types which are well presented in the results section, however, some

work is needed in the figures and figure captions to make the data clearer for the reader. The premise of understanding how fluid migration and seismic faults are associated as a way to mitigate natural hazards is a very interesting and relevant question. However, the mechanisms behind how seismicity specifically results in fluid and mud expulsion are currently almost entirely absent in the manuscript. The nature of the conduit, whether exploitative through faults or self-generative in the form of hydraulic fracture pipes, is poorly described or debated and the figures (seismic specifically) are not clear in their presentation of the faults that have propagated through the salt and act as leakage pathways. Why did any faults propagating upward not detach in the salt and if any faults did propagate through the salt why have they not annealed over time due to the mechanical behavior of salt? If there is to be a relationship between fluid and mud remobilization, seismic cycles and underlying faults then the discussion must first present a balanced argument that clearly demonstrates that the migratory pathway is exploitative. Only then can the authors build a stronger argument for the conclusion that the BMV of all offshore mud volcanoes presents the best site for future monitoring of fluid expulsion associated with seismic cycles. It is my opinion that the core premise of the manuscript is good, the data and results are strong. I recommend this paper for publication but only after major revision to the points raised above and also in the comments below.

Comments and recommendations

The first paragraph is a very generic introduction for a mud volcano paper. A core aspect of the paper is the relationship between mud volcanoes (or focused fluid flow in general) and seismic cycles. I recommend that the authors focus more on known relationships between mud volcanoes and seismicity and how seismicity can trigger mud volcanism. Is this through a mechanism of seismic pumping or sheering and overpressuring of a source layer.

Avoid weak or unsupported statements e.g. Page 2 line 33 starting 'We believe, in fact..' and Page 2 lines 18-20, which is an important statement that is uncited, so reads

as conjecture.

While it is admirable and scholarly how extensive the use of citations is, in many instances it is beyond what is necessary and in fact breaks up and detracts from the main text with sometimes two to three lines of citations. Please wherever possible reduce the citations throughout.

While the BMV has been described in prior studies there are a couple of problems with immediately naming the structure the BMV by the start of the methods and results: firstly, you are already imposing an interpretation before laying out the evidence that ultimately leads to the diagnostic that this is a MV; Second, one of your main conclusions is that the BMV is in fact an MV, but the impact of this conclusion is completely removed by imposing the conclusion prior to presenting the evidence. I recommend refraining from calling this structure an MV and building your argument through the results, ending with a summary of the diagnostics (both your own and those already published) that allow you to name this structure the BMV.

I recommend modifying the structure of the method and data and results into a single method and data section and a separate results section. While I appreciate this must have been a conscious stylistic decision, due to the fact you are presenting information from so many different data types, constantly jumping between the two makes the reading experience disjointed and impedes real integration of all the different data.

Interpretation is often mixed with results. Perhaps it would help the reader to have an interpretation section at the end of the results. For the author this would avoid mixing observations and interpretation and would present an opportunity to more seamlessly integrate more than one data type in an interpretation. This would also make the manuscript more enjoyable and understandable for the reader.

The discussion in broken up into two parts. The first part is the origin and activity of the BMV. I enjoyed the section on the present activity of the BMV, which used backscattering and geochemical analysis to demonstrate that the mud volcano is currently quiescent but provides a conduit for dewatering deeper layers. However, my recommendations for the rest of this section are as follows: 1) the first part of the discussion should first demonstrate that the BMV is in fact a mud volcano; 2) At present the morphology section doesn't pose a question to be answered but rather states what was observed from the results. The geometry of a mud volcano is typically governed by two main factors, the viscosity of the extruding mud and the geometry of the connecting conduit. The scatter you see in the graph of height vs diameter is primarily related to this. Consider, what does the morphology (such as height and slope angle) tell us about the either the type of conduit or the fluidity of the extruded mud? Is this a mud cone or mud pie? What does the evidence of palaeo flows tell us about the episodicity of this mud volcano and what governs the changes in pressure that give this episodicity? These are simple questions that when answered will elevate the first part of the discussion. Finally, in the drainage process section, what is the origin of the fluid and mud? Do you have any constrains on the age/origin of the extruded mud from either seismic observations of depletion zones or core samples? Samples of hypersaline water are indicative of a fluid source at least as deep as the Messinian evaporites. However, it is currently unclear if the mud is also pre-salt in origin. This is important for our understanding of the process for how the mud has been mobilized, e.g. fluidization vs liquefaction.

I would recommend having a native English speaker read over the manuscript prior to any resubmission. Some sentences are unnecessarily long and would benefit from being broken down or made more concise. One key point per sentence with the subject of the sentence at the start.

Annotations in figures are sometimes a little lacking, with text size often too small, certain annotated lines too thin or a colour that is hard to see and basic annotations including north arrows, colour bars and scale bars sometimes missing (examples given below).

Specific comments

Page 1, Line 8 – weak sentence with the use of probably. Whether the faults are seismically active or not is a fundamental part of the paper.

Page 1, Line 15-17 – once again this is a weak sentence with too much use of words including may 'contribute, potential and feasible, in favour'. Please be clearer, assertive and concise in what you want to say. This is supposed to be one of your main conclusions.

Page 2, Lines 12 – 15 – an example of a sentence that would benefit from being split in two. I suggest a full stop after the first set of citations and then a separate sentence above the amount of methane emitted from known mud volcanoes.

Page 2, Line 18-20 – This sentence needs citations, otherwise it is an unsupported statement.

Page 2, Line 24 – delete 'In other words,' at the start of the sentence. Start with 'The faults. . ..'.

Page 2, Line 22 – Weak start to sentence. Remove 'We believe,. . ..' And start with 'The activity of some of these. . ..'

Page 3, Line 9 – As you ascertained from the analysis of the BMVs geometry compared to other recoreded MVs, it is in fact a relatively small mud volcano. Therefore, please do not refer to it as a 'large mud volcano'.

Page 5, Line 20 – You state that seismic events were selected within a distance of 12 km from the transect track. Please explain in the methods why you chose a distance of specifically 12 km.

Page 6, Line 16 – Please annotate/highlight examples of the small scale mass-wasting features listed here in Fig. 3.

Page 6, Line 19 – Please highlight in Fig. 3 where 'this well-defined flat area of 26 km2' is.

Page 7, Line 3 – Can you please justify why you have selected this data set of 232 mud volcanoes?

Page 7, Line 4 – How have you defined the diameter and height of the BMV? This should be explained in the methods with reference to a figure showing these parameters. From the data currently presented it is not clear how you interpret the base of the MV in order to measure its height. An awareness of the imaging artefacts at the margin and beneath the MV should also be demonstrated. How do you interpret the conduit? These are often much narrower than the area of acoustic noise.

Section 3.4.2 – The seismic facies of the units is briefly described, but the acoustic character of the bounding reflections is not described. What is the polarity? What is the acoustic impedance contrast? Is there variability in acoustic character along the horizon? This information gives further insight into the velocity and origin of the sediments.

Page 9, Line 8 – Please ensure all heading and sub-headings are capitalized where appropriate. Please check for other examples.

Page 9, Line 16 – Is there a more technical term that can be used than cloudy facies. Is this term standard? I struggle to understand what you mean.

Page 14, Line 5 – Please reference the statement about an association between overpressure and the MSC. See Bertoni et al., 2015 and Al-Balushi et al., 2016.

Figure 1 – Please change the grey and grey dashed line colours to something that is brighter and stands out more. They are currently hard to see.

Figure 2 – Please add the north arrow.

Figure 3 – In Fig. 3b contours and contour numbers are at times almost impossible to see. Consider white for the text and thicker white lines for the contours. In Fig. 3a & 3b, more annotation and description in the caption of morphological features is needed. What are the detailed diagnostics from these maps for the BMV?

Figure 4 – This figure would benefit from further subdivision into sections a-e. In Fig. 4a, the text size is too small, the units are missing, the scale bar is missing (also in 4b), the lines in the map are too thin and 'Rim' is not labelled on the bathymetric profile. 4b needs more annotation. What is meant by 'breaking'? Other than being nice to look at without further annotation or comment in the caption it is not clear what the 3D perspective adds. Please use a different line track labelling than a-a' and b-b' because a and b are used for subdividing the figure.

Figure 5 – This figure needs to be completely reworked with much more interpretation added to both the figure and caption. At present it tell the reader, especially one who doesn't work often with this data type, very little. The caption for 5a is too sparse and 5b reads in a way that suggests that there is massive fluid expulsion, although I think the author wants to state the opposite. Also, please be careful of using terms like 'massive fluid expulsion'. Unless you can quantify what you mean by 'massive' I suggest avoiding.

Figure 6 – Please add the north arrow.

Figure 7 – The text sizes vary throughout the figures. Please be consistent. Please add to Fig 7a a scale bar, north arrow and colour bar. The chirp profiles in Figs. 7b-d are quite zoomed out and so it is difficult to track any horizons. You have an un-interpreted version of this figure in the supplementary material, so please add line interpretations for horizons and also a more zoomed in view of the stratigraphy. What do the dashed lined represent?

Figure 8 – The region of extrusion is shrouded by artefacts, so how have you defined the boundary of the MV? This boundary looks more like the margin of a zone of artefacts and doesn't truly depict the margins of the MV.

Figure 9 – In the caption, should chirp profile be labelled left rather than right? This figure would also benefit from being broken into subsections, each of which is described in the caption.

Figure 10 – The figure is well presented. Fig, 10d is too zoomed in to interpret.

Figure 11 – Good

Figure 12 – Nice figure. It would be good if it were integrated into the main body of text more as it is the final conceptual model.

———————————————————

---

## Author Comment (AC2) · 16 Apr 2019

Dear Editor,

Please find below our detailed responses to all Reviewers' requests.

We thank you and the reviewers very much for the constructive efforts done on our manuscript.

Sincerely Andrea Billi and co-authors

Reviewer 1

[Figure]

The manuscript 'The Bortoluzzi Mud Volcano (Ionian Sea, Italy) and its potential for tracking the seismic cycle of active faults' presents a multi-parameter dataset of a mud volcano in the Ionian Sea. The authors consider this mud volcano as a promising location for installing a cabled multiparamteric station in order to study the relationship between the seismic cycle of underlying active faults and fluid expulsion. The data set is definitely interesting and interpretations seems to be sound (though not always easy to verify based on the presented data, see below); hence, I principally support publication of the manuscript but suggest major modifications, especially concerning the structure of the manuscript and the data presentation. In addition, the authors should elaborate more on the general significance of their study, or rather demonstrate more clearly, why the BMV is representative and best suited for the suggested investigations. I am not an expert on geochemistry and cannot comment this section of the manuscript.

The manuscript has been entirely reorganized, separating methods and data (see section 3. Methods and Data), results (see section 4. Results), interpretation (see section 5. Interpretation), and discussion (see section 5. Discussion). Morever, the conclusion section has been largely re-written (see section 7. Conclusions) as requested by the reviewers. Some further insights into the mud volcano have been added (P. 3 L. 2-18, P. 19 L. 9-23); however, the main goal of this article is not to characterize the activity, evolution, and conduit of the mud volcano. The main goal, as stated at P. 3 L. 9-35, is: "Our aim in this article is not finding a link between earthquakes and the activity of mud volcanoes but to understand whether even a mud volcano that seems inactive or poorly-active can constitute an open and preferential pathway for possible fluid precursors of earthquakes. Whether these precursors truly manifest before an earthquake is a different task that has to be accomplished with a specific monitoring as we will explain in the discussion section. Our aim is nonetheless relevant for many geoscientists as noteworthy examples of earthquake geochemical precursors are documented in many previous works (Wakita et al., 1988; Igarashi et al., 1995; Claesson et al., 2004; Huang et al., 2012; Inan et al., 2012;

Skelton et al., 2014, 2019; Sano et al., 2016; Barberio et al., 2017; Petitta et al., 2018) and are very promising for the science of earthquake forecasting." And also: "As mentioned above, our main aim is not only to characterize the mud volcano but also to provide a contribution toward a potential and feasible future path for the use of these ubiquitous structures in favour of the monitoring and mitigation of natural hazards." We cannot be sure that the BMV is the BEST spot where installing a monitoring station and indeed this is not a point in our work. Surely, we can assert that: the BMV sits atop seismically active faults; geofluid circulation is active through the BMV; the geochemical signal provided by the BMV is stronger than those provided by nearby active fault zones; and the BMV is one of the best characterized mud volcanoes of the seismically-active Ionian area. These features altogether make the BMV a potentially suitable spot where installing a monitoring station (see Conclusions section).
* * *
General comments: 1) I strongly suggest to change the structure of the manuscript. Separate 'Material and Methods' from the 'Results'. In addition, you mix results and discussion quite a bit. Introduce the methods first and then summarize the results. You state that you have a multi-parameter data set. Make use of this opportunity and try to combine the data sets already during presentation of the results. Morphology and subsurface features (seismic data) can be combined. The second major type of data are the geochemical data. The integrated presentation of the data would be the perfect basis for the discussion. The main points of the discussion should then be the (1) Origin and Activity, and (2) the potential to use the BMV as location to investigate the relationship between the seismic cycle of underlying active faults and fluid expulsion.

The manuscript has been entirely reorganized, separating methods and data (see section 3. Methods and Data), results (see section 4. Results), interpretation (see section 5. Interpretation), and discussion (see section 5. Discussion). Morever, the conclusion section has been largely re-written (see section 7. Conclusions) as requested by the reviewers.

2) It remains unclear why you selected the BMV for the investigations. How does it compare or differ to other mud volcanoes in this area. Mud volcanoes in the area haven been investigated before. Not much information is given on previous investigations of mud volcanoes in the Ionian Sea.

The selection of the BMV is now explained at P. 3 L. 33-35. Our conclusions are not that the BMV is the best structure where installing a monitoring station (see Conclusions section). Not at all. We conclude that results from the BMV may help understanding other similar structures nearby and far away, where these other structures may be good as well for monitoring stations. We have no evidence to reject or suggest other structures. Our study is willing to be a pilot study for similar structures elsewhere. See also our previous response: We cannot be sure that the BMV is the BEST spot where installing a monitoring station and indeed this is not a point in our work. Surely, we can assert that: the BMV sits atop seismically active faults; geofluid circulation is active through the BMV; the geochemical signal provided by the BMV is stronger than those provided by nearby active fault zones; and the BMV is one of the best characterized mud volcanoes of the seismically-active Ionian area. These features altogether make the BMV a potentially suitable spot where installing a monitoring station.

3) This previous point is closely related to the general relevance of your investigation. What is special compared to previous study? How does your investigation contribute to the general understanding of mud volcanos? Why is the BMV more than a case study?

See our previous response and our new conclusion sections. We have no evidence to reject or suggest other mud volcanoes. One of the strength points of our study is the geochemistry of seawater columns above the BMV and above nearby active fault zones. The BMV shows evidence of geofluid circulation that seems instead absent or limited over the active fault zones

(see conclusions). This evidence may be true also for other mud volcanoes.

4) The conclusions are mainly a list of your most important results, which are already listed in the abstract. A conclusion should tell the reader what she can or he could do with the newly acquired knowledge. Answer the question "So what?".

Done. Conclusions entirely rewritten.

5) Several of your statements cannot be verified based on the presented figures. I am aware that you cannot show every detail but some enlargements/details in order to support your statements would be useful (see specific comments)

Most Figures have been improved following the in-dications provided by both Reviewers (see below).

6) The geochemical analyses of the BMV is based on one cast above the BMV. I am not a geochemist but I wonder how representative this cast is?

Obviously more casts would be more significant but renting a scientific vessel to these casts is a big and expensive issue for scientific institutions. Note, however, that the relevance of this cast is corroborated by the comparison with the other three casts realized during the same marine survey (Fig. 11 and related text)

7) Annotations of several figures are very small and difficult to read

Most figures have been improved and enlarged to make them more easily readable.

8) Avoid extensive referencing. References are always a selection; you cannot mention all, especially for a topic, where quite a lot of work has already been published. I suggest not using more than three references for one statement.

Following the Reviewers' indications, we have omitted the following references: Argnani and Bonazzi, 2005; Argnani et al., 2009; Barber, 1981; Barnard et al., 2015; Deville et al., 2003; Dimitrov, 2002; Higgins and Saunders, 1974; Leon et al., 2007; Limonov et al., 1996; Milkov, 2000; Rabaute and Chamot-Rooke, 2007; Robertson and Kopf, 1998
* * *
9) I am not a native English speaker but I have the impression that some (little) language polishing is needed. I have not made any specific comments on this point.

Done.      We   have   improved   the   English   as   much   as   we   could.
* * *
Specific comments: Page 1, Line 14/15: Conclusions are drawn : : : Give specific conclusions. Do not only announce them.

Done. Sentence modified (see abstract). ___________________________________________

Page 2, Line 5/6. This is an example for extensive referencing.

Following the Reviewers' indications, we have omitted the following references: Argnani and Bonazzi, 2005; Argnani et al., 2009; Barber, 1981; Barnard et al., 2015; Deville et al., 2003; Dimitrov, 2002; Higgins and Saunders, 1974; Leon et al., 2007; Limonov et al., 1996; Milkov, 2000; Rabaute and Chamot-Rooke, 2007; Robertson and Kopf, 1998
* * *
Page 2, Lines 8/9: Another example for excessive referencing. Not mentioned any more in the following. Check for the entire text.

Following the Reviewers' indications, we have omitted the following references: Argnani and Bonazzi, 2005; Argnani et al., 2009; Barber, 1981; Barnard et al., 2015; Deville et al., 2003; Dimitrov, 2002; Higgins and Saunders, 1974; Leon et al., 2007; Limonov et al., 1996; Milkov,

2000; Rabaute and Chamot-Rooke, 2007; Robertson and Kopf, 1998

Page 2, 14/15: 27 Mt methane a-1 : : :Is this the only estimate? I assume that estimates cover a wide range of numbers.

As far as we know, there are not many estimates. We know this one that is an authoritative one.

Page 2, Line 15/16. Calabrian arc is not marked on Fig. 1.

Done. The Calabrian arc is emphasized in Figure 1

Page 2, Line 29/30: It seems that you do not use the OBS/OBH data in the manuscript. Why do you mention them here? Alternatively, why do you not use them for the manuscript?

Explanation at P. 2 L. 27-30.

Page 3, Line 4/5: Be more specific. Name some of the potential precursors.

Done, see top of P. 3.

Page 3, Line 26: Mark Aeolian Islands on Fig. 1

Done. The Aeolian Islands is emphasized in Figure 1

Page 3, Line 31: Mark escarpments on Figure 1. If the figure is getting too crowded, an additional figure showing all the locations mentioned in the text may be useful.

Done, see Figure 1. We enlarged the dimensions of the Figure 1 for a better comprehension.

Page 5, Lines 6-13: You mentioned OBS/OBH deployments in the introduction. Why are these data not included? Local seismicity would be interesting. Or, are they still on the sea floor?

Explanation at P. 2 L. 27-30. _______________________________________________

Page 5, Line 15. What is the extend of the BMV area? Is it the area shown on Fig. 1? Clarify

Explained at P. 5. _________________________________________________________

Page 5, Line 15, 16. You explicitly state that the focal mechanisms are from earthquakes that are < 4.5 in magnitude. This implies that there are also larger earthquakes. What is about larger earthquakes?

Explanation given in Section 3.2. ____________________________________________

Page 6, Lines 1-4. Processing details should be moved to the supplement or deleted, especially as the data have already been presented elsewhere (Bosman et al., 2015)

Omitted, see Method section. _______________________________________________

Page 6, Line 5/6: change to 'with a lateral resolution varying between 10 m for water depths shallower than 1000 m to 25 m at greater depth'.

Omitted, see Method section. _______________________________________________

Page 6, Lines 6-12: Delete. No need to include such details. You do not need to show that you know procedures for calibrating a multibeam system.

Omitted, see Method section. _______________________________________________

Page 6, Line 17, 18: This sentence reads very strange. First, you state that 22_ is a maximum. Then you continue that it is even up to 28_. I assume that the 28_ is related to the 35 km long escarpment. Maybe split to two sentences, though I am not sure what you mean. Suggestion: 'The upper part of the continental slope is characterized

by a very steep slope (about 15_) that reaches a maximum of 22_. Even larger slope gradients of up to 28_ are found along the NE-SW orientated main escarpments, which are up to 35 km long.

Rephrased, see Section 4.2.1. ___________________________________________________

Page 6, Line 24/25. Difficult to follow. In your Fig. 4a, you show one rim. In this sentence, you mention two rims. This is confusing.

Rephrased in Section 4.2.1. ___________________________________________________

Page 6, Line 32 – Page 7, Line 11: More discussion. If you include the BMV in this discussion, you should mention the volume during the description. It also remains unclear if you calculate the volume of the BMV based on the given formula of if you use exact volumes based on your data.

We have better explained in the text (see Section 4.2.1) that we used the given formula to compute the BMV volume using height and diameter (morphometric features) through bathymetric sections reported in Fig. 4. ___________________________________________________

Page 7, Line 17. I strongly suggest converting the TWT to depth and resolution in meters using a constant velocity of 1500 m/s.

Done. See P. 18 L. 25. ___________________________________________________

Page 7, Line 27-30: I like that you have unintepreted sections in the supplement. This give you the chance to add more interpretation to the interpreted sections. E.g., H1 is easy to identify at the location where you label H1. I find it difficult to trace H1 from these locations to the sides. In addition, erosion at H1 is difficult to see. Mark this clearly or show enlargements to support this statement.

Done. We modified Figure 7 following the reviewer's comment. We interpreted the principal horizons and we indicated the erosional features.

Page 7, Line 31 to Page 8/Line 10: Similar comment as before. Mark lower and upper boundary more clearly. Mark features described in the text on the figures more clearly (Erosion, onlaps and so on). Make it easier for the reader to follow your description. In addition, I strongly suggest using depth/thickness of units in meters (possibly in addition to numbers in TWT).

Done. We modified Figure 7 following the reviewer's comment. We interpreted the principal horizons, we indicated the features described in the text, and we added the scale bar in meters.

Page 8, Lines 13 to 30: This is interpretation and I would suggest moving this to the discussion section. Sometimes, there are good reasons to leave this type of interpretation with the data description but the general structure of the chapters (methods, results, interpretation) is challenging the approach of an integrative interpretation of your multi-parameter data set.

Done. See the Interpretation section (Section 5).

Page 8, Lines 21-24: Do you have a preferred interpretation based on your data. Delete the figure numbers of the Evans et al. paper. It is sufficient just to reference the paper.

Omitted.

Page 9, Line 9. How sure can you be about the location of the core. Did you use a positioning system (e.g. USBL-System) during coring, or is it the surface location of the vessel? The chirp profiles show large lateral variability at the projected coring location. How sure can you be that you really penetrated the units imaged at the projected location of the core?

The positioning of the core is obtained through the Differential GPS positioning of the ship (accuracy 0.5 m) (Section 3.5). During coring, the position-
ing of the vessel is managed by dedicated hydrographic software providing
a radius of a few meters. Finally the horizontal scale of the seismic pro-
file was wrong in the previous version and has now been replaced correctly.

Page 9, Line 9 to 23: This is again mixing results and interpretation/discussion, even
more than before. E.g., you describe the patchy/cloudy facies when discussing it. I
strongly suggest to give a good description of the core first and discuss it afterwards
(here or even better in the discussion section).

Done. See separation of method-data and results.

Page 9, Lines 26-31: Give the basic acquisition parameters (such as source frequency
in order to estimate the resolution). You give very detailed information for the other
systems but almost no information is given for the seismic system.

Done in the text (see Sections 3.4 and 3.6).

Page 10/11, Results: It is again a mixture between description and discussion.

Done. See separation of method-data and results.

Page 10. Line 13/14: The Ionia fault system is labelled from SP200 – 1400 on the
CROP M-31 line (Fig. 10b), which contrasts the statement in the text (SP 1100 -1400)
Clarify.

Done. See top of P. 14.

Page 10, Lines 24/25. Mark the flower structures. It is clear on the western side of the
profile but not obvious for the eastern side.

We modified the sentence as follow: Earthquakes focal mechanisms located in the study area (Figs. 1 and 2) show that faults are mostly characterized by strike-slip and transtensional kinematics defining a clear flower-like structure on the western side and an oblique-slip deformation zone on the eastern side (Fig. 10b).
* * *
Page 11, Lines 4-8. I do not think that I see the slightly transparent seismic reflection signal beneath the BMV. I may see some small amplitude variations but they seem to occur along the entire profile. In addition, I also see some areas with enhanced reflection amplitudes beneath the BMV (e.g. in the transparent unit directly beneath the seafloor reflector in Fig. 10e). Strong amplitudes also indicate rock/fluid indication. A better description and a better visualization is required in order to believe your statement (indication for fluid-rock interaction).

The reviewer is right. We properly modified the text describing the seismic evidences of fluid-rock interactions. We erased and/or replaced the term "transparent" with "chaotic" which is more suitable for the seismic pattern visible on the profile. We analyzed in text the strong amplitudes reflector beneath the seafloor. Furthermore, we changed Figure 10d introducing a bigger portion of the seismic line, in this way the chaotic signal beneath the mud volcano is visible. The chaotic signal is located in sediments otherwise characterized by reflectors showing high continuity.
* * *
Page 11, Line 12 – Page 12, Line 10: Cannot judge this part. Seems to be quite long for me. Isotopic numbers are not superscript (in the entire text).

Done. _______________________________________________________________________

Page 12, Line 17. Change 'comparable' to 'similar'. Everything is comparable.

Done. _______________________________________________________________________

Page 12, Line 18, 19: This statement seems to be wrong. The Br content for GeoC2 is

0.990, and 1.04 in 500 and 1000 m depth for the BMV. This s less than 11_ higher. The highest Brome content is found in 200 m depth at BMV. This value is very unlikely to be related to the BMF because it is far above the BMV. How reliable is the statement, that the Brome content is really higher. Can you make this statement on one single cast?

We agree with the reviewer. We rephrased this state-ment omitting the part dealing with Br as poorly relevant. ________________________________________________________________________

Page 12, Line 23, 24. Not clear where you see the decrease in oxygen content. Be more specific.

Explained in section 4.6. ________________________________________________________________________

Page 12, Line 27. 0.73 (not 0.71 as in the text) is given for the $CO_2$ content in the table.

Thanks, corrected. ________________________________________________________________________

Page 13, Line 7 to Line 18. Delete. This section is not containing relevant information.

Done. ________________________________________________________________________

Page 13, Line 20-26.As mentioned above, some parts of the result section would be located here much better. Otherwise, this section is just a repetition of the results. As mentioned above, the semi-transparent seismic facies is not evident for me.

Done. ________________________________________________________________________

Page 13, Line 17-19: As mentioned above. How representative is a single sample? I believe that higher $CH_4$ content is significant but convince the reader.

See our previous responses. We have no further data. It is not so trivial to rent a scientific vessel and monitor a mud volcano. ________________________________________________________________________

Page 13, Line 28. See comments concerning the semi-transparent facies above.

We report here the answer related to this comment: We properly modified the text describing the seismic evidences of fluid-rock interactions. We erased and/or replaced the term "transparent" with "chaotic" which is more suitable for the seismic pattern visible on the profile. We analyzed in text the strong amplitudes reflector beneath the seafloor. Furthermore, we changed Figure 10d introducing a bigger portion of the seismic line, in this way the chaotic signal beneath the mud volcano is visible. The chaotic signal is located in sediments otherwise characterized by reflectors showing high continuity.
* * *
Page 15, Lines 17-20. Delete. No relevant information.

Although a synthesis may appear as redundant, we believe that it is useful to focus the main result. We would like to leave these few lines of synthesis.
* * *
Page 15, Line 29. Mention how long before the earthquake the pore pressure changes have been measured.

Done. _____________________________________________________________________________

Page 15, Line 22 - Page 16, Line 15. These are nice examples but this chapter reads as earthquake forecasting can easily be done by geochemical monitoring. Add a few sentences that the story is not that easy.

Done. See P. 22 L. 15-17. ___________________________________________________________

Page 16, Line 16-33: Principally, I agree but I am not convinced that the BMV is really the best place just because it is best characterized by the most extensive data set. I assume that there are some mud volcanoes with proven ongoing activity close by (escape of fluids, mud flows etc.). Why not using another mud volcano? What is special about the BMV?

We agree with the Reviewer. We do not state that the BMV is the best place for a monitoring station. We state the following (Section 6.2.3): "Collectively, the
geochemical, geophysical, and geologic data presented in this paper show that the BMV, likewise other onshore monitoring stations previously realized (e.g., Claesson et al., 2004; Skelton et al., 2014; Barberio et al., 2017; Huang et al., 2017), could be a proper site where installing a cabled submarine multiparametric station (Fig. 12) to study possible relationships between the seismic cycle of the underlying active faults and geofluid emissions ... In the case of the BMV and other mud volcanoes, dissolved gases such as $CO_2$ and $CH_4$ may rather easily be monitored by submarine devices (Annunziatellis et al., 2009; Roberts et al., 2017). In particular for the seismically-active Ionian Sea, many other existing mud volcanoes (Gutscher et al., 2017; Loher et al., 2018) may host a monitoring station, but the BMV is so far the one characterized by the largest geological, geophysical, and geochemical dataset and its location seems connected with seismically-active faults (Fig. 2)." In synthesis, the BMV (for the data collected) could be a proper site for a monitoring station but other similar structures may also be appropriate. So far, the BMV is the best studied mud volcano of the Ionian Sea, at least the area nearby Calabria and Sicily. See also our previous response: We cannot be sure that the BMV is the BEST spot where installing a monitoring station and indeed this is not a point in our work. Surely, we can assert that: the BMV sits atop seismically active faults; geofluid circulation is active through the BMV; the geochemical signal provided by the BMV is stronger than those provided by nearby active fault zones; and the BMV is one of the best characterized mud volcanoes of the seismically-active Ionian area. These features altogether make the BMV a potentially suitable spot where installing a monitoring station.
* * *
Page 17, Lines 1-11: Delete here. Include in a rewritten conclusion.

Conclusions entirely rewritten _________________________________________________

Page 17, Conclusion. See general comments. Answer the question "So what?"

Conclusions entirely rewritten _________________________________________________

[Figure]

Fig. 1: Nice figure with a lot of information. Try to add additional geographic locations and features mentioned in the text. Split to two figures if the figure is getting too crowded. I suggest deleting the earthquake locations from the figure because they are shown much better in Figure 2 and you only show the earthquakes in the area of Fig. 2. The grey dashed line (CA99-215 line) is very difficult to see.

We enlarged the dimensions of Figure 1 for a better comprehension. The grey dashed line has been replaced with a red brighter color.
* * *
Fig. 2: What is the difference between the solid and the blue dashed line? It seems for me that the dashed blue line is not needed.

The gray and dashed gray lines in Figure 2a are the seismic lines reported in Figure 1. The gray dashed is the seismic line reported in Figure 10a. The solid blue line represents the transect length, where the seismic events are projected, which does not correspond to the length of the seismic profile. We have replaced the colors of the seismic lines in Figure 2a as reported in Figure1.
* * *
Fig. 3: Figure should be larger. Add to figure caption 'Location of the maps is shown on Fig. 2a'.

We enlarged the figure and we added the text in the caption.
* * *
Fig. 4: Figure should be larger. Annotations/Numbers are very small and difficult to read. Add distance scale to maps. Moat not labelled on the SE part of Profile a a'.

Done. We enlarged the figure and the labels, we added the distance scale, we labelled the moats in the profiles where present.
* * *
Fig. 4: Distance scale is missing on maps.
Done. Please see the new version of Fig. 4.

Fig. 5. Mark location of profile on Fig. 4. Add along track scale. Figure caption is confusing. It is not clear that 'absence' relates to both statements. Now it reads as the absence of amplitude anomalies indicates massive fluid escapes. Modify.

The reviewer is right. We changed the caption of the figure and we modified the figure following the reviewer's comment.

Fig. 6: Exponents are not in superscript for the volumes.

We do not understand this comment. The exponents for the volumes are in superscript.

Fig7: See comments above. Add more interpretation and/or close-ups. Add absolute depth for the y-axis. I suggest using meters instead of TWT.

Done. Please see the new version of Figure 7. A close-up view of 246 chirp profile has been added.

Fig. 9: I have serious problems to correlate the enlargements of the core section with the photo of the entire core. E.g., section b should include the erosional contact. I can see a clear colour change at this interval in the image of the entire core but I cannot see this colour change or the erosional contact in the enlargement b. Enlargement b looks very homogenous for me. The same is the case for enlargement d, which also show clear colour changes. Some sedimentary layers seems to be down bended on the left-hand side of the image of the entire core, which is not visible on the enlargement. Annotations are very small and almost impossible to read.

We agree with the comments of Reviewer 1. We have now properly modified the figure. In particular, the enlargement "b" has been moved upward showing

the erosional contact in the lower portion. Furthermore, following the reviewer's comment on enlargement "d", we realized that there was a mistake in the location of the white polygons along the entire core. The white polygons are now properly located and, in this way, the enlargement "d" perfectly match the area defined by the white polygon along the entire core. We enlarged the labels.
* * *
Figure 10. Nice figure. See comments made above (especially concerning the slightly transparent seismic facies).

Thank        you.            We        answered        and        properly        modified        text        and        figure        following        the        reviewer's        comments.
* * *
Fig. 11: OK Fig. 12: Nice figure but nor really discussed in the text. Discuss the outline of your monitoring tools in more detail. Good luck with the revisions.

This manuscript is not focused on the installation and tools of monitoring stations. This figure serves solely to provide an outlook of future possibilities for the readership. We do not intend and we are not able to discuss monitoring tools or installation procedures.
* * *
Reviewer 2

Summary The author presents an examination of the here named Bortoluzzi Mud Volcano (BMV) with the aim of demonstrating its potential for tracking the seismic cycle of active faults. The analysis is undertaken through the integration of numerous geological, geochemical and geophysical data. The primary conclusion is that the BMV could present a suitable site for installation of a cable submarine multiparametric station to study the potential relationship between subsurface vertical fluid migration and the seismic cycles of active faults. The manuscript offers an impressive integration and analysis of various data types which are well presented in the results section, however,

some work is needed in the figures and figure captions to make the data clearer for the reader. The premise of understanding how fluid migration and seismic faults are associated as a way to mitigate natural hazards is a very interesting and relevant question. However, the mechanisms behind how seismicity specifically results in fluid and mud expulsion are currently almost entirely absent in the manuscript. The nature of the conduit, whether exploitative through faults or self-generative in the form of hydraulic fracture pipes, is poorly described or debated and the figures (seismic specifically) are not clear in their presentation of the faults that have propagated through the salt and act as leakage pathways. Why did any faults propagating upward not detach in the salt and if any faults did propagate through the salt why have they not annealed over time due to the mechanical behavior of salt? If there is to be a relationship between fluid and mud remobilization, seismic cycles and underlying faults then the discussion must first present a balanced argument that clearly demonstrates that the migratory pathway is exploitative. Only then can the authors build a stronger argument for the conclusion that the BMV of all offshore mud volcanoes presents the best site for future monitoring of fluid expulsion associated with seismic cycles. It is my opinion that the core premise of the manuscript is good, the data and results are strong. I recommend this paper for publication but only after major revision to the points raised above and also in the comments below.

The manuscript has been improved following the indications by both Reviewers. In particular, we have improved the separation between methods, data, results, and interpretation. We have also entirely rewritten the conclusions section. Although most questions posed by Reviewer2 are surely very interesting, they cannot be substantially answered with our data and for many submarine mud volcanoes. The rationale of this paper is as follows: (1) a possible mud volcano (submarine) was signaled in the previous literature off Calabria (in a very seismic region) based solely on bathimetric evidence; (2) we surveyed this structure and collected geological, geophysical, and geochemical data; (3) thanks to these data, we confirmed and corroborated the hypothesis that the studied structure is truly a mud volcano, which

sits atop seismically-active faults and has its inner conduit (through the volcano) somehow open with ascending fluids; (4) compared with nearby active fault zones, the conduit through the BMV seems to be a more efficient fluid carrier than the active fault zones (based on geochemical data); (5) outlook: what this structure might be used for? Given many recent studies/results on geochemical precursors of earthquakes, the BMV may be a future site for hydrogeochemical/seismic monitoring of potential seismic precursors. Although it would be surely interesting, we have not enough evidence to infer or speculate over the nature of the conduit or over details concerning the faults beneath the BMV. Concerning the faults, in particular, we can only speculate over what we can see on the seismic reflection profiles. The related resolution does not allow further inferences. We recognize that in the previous version we used the term conduit in a misleading way. We have no data to speculate over the mud conduit of the BMV except for what stated at P.19 L. 15-25. We changed conduit into fluid pathways when talking about geofluids ascending through the BMV.
* * *
Comments and recommendations The first paragraph is a very generic introduction for a mud volcano paper. A core aspect of the paper is the relationship between mud volcanoes (or focused fluid flow in general) and seismic cycles. I recommend that the authors focus more on known relationships between mud volcanoes and seismicity and how seismicity can trigger mud volcanism. Is this through a mechanism of seismic pumping or sheering and overpressuring of a source layer.

Done. See Introduction section. ___________________________________________________

Avoid weak or unsupported statements e.g. Page 2 line 33 starting 'We believe, in fact..' and Page 2 lines 18-20, which is an important statement that is uncited, so reads as conjecture.

Omitted. ___________________________________________________________________________

While it is admirable and scholarly how extensive the use of citations is, in many instances it is beyond what is necessary and in fact breaks up and detracts from the main text with sometimes two to three lines of citations. Please wherever possible reduce the citations throughout.

Following the Reviewers' indications, we have omitted the following references: Argnani and Bonazzi, 2005; Argnani et al., 2009; Barber, 1981; Barnard et al., 2015; Deville et al., 2003; Dimitrov, 2002; Higgins and Saunders, 1974; Leon et al., 2007; Limonov et al., 1996; Milkov, 2000; Rabaute and Chamot-Rooke, 2007; Robertson and Kopf, 1998
* * *
While the BMV has been described in prior studies there are a couple of problems with immediately naming the structure the BMV by the start of the methods and results: firstly, you are already imposing an interpretation before laying out the evidence that ultimately leads to the diagnostic that this is a MV; Second, one of your main conclusions is that the BMV is in fact an MV, but the impact of this conclusion is completely removed by imposing the conclusion prior to presenting the evidence. I recommend refraining from calling this structure an MV and building your argument through the results, ending with a summary of the diagnostics (both your own and those already published) that allow you to name this structure the BMV.

The rationale is a bit different. This structure (BMV) was already signaled as a mud volcano (based on bathymetric data) by Gutscher et al. (2017) and Loher et al. (2018). So there is no reason to present it as a new discovery at the end of the manuscript. We corroborate this interpretation (mud volcano by Gutscher et al., 2017, and Loher et al., 2018) with many other data and propose an outlook. As mentioned previously, the rationale of this paper is as follows: (1) a possible mud volcano (submarine) was signaled in the previous literature off Calabria (in a very seismic region) based solely on bathimetric evidence; (2) we surveyed this structure and collected geological, geophysical, and geochemical data; (3) thanks to these data, we confirmed and corroborated the hypothesis that the studied structure is truly a mud volcano, which sits atop

seismically-active faults and has its inner conduit (through the volcano) somehow open with ascending fluids; (4) compared with nearby active fault zones, the conduit through the BMV seems to be a more efficient fluid carrier than the active fault zones (based on geochemical data); (5) outlook: what this structure might be used for? Given many recent studies/results on geochemical precursors of earthquakes, the BMV may be a future site for hydrogeochemical/seismic monitoring of potential seismic precursors.
* * *
I recommend modifying the structure of the method and data and results into a single method and data section and a separate results section. While I appreciate this must have been a conscious stylistic decision, due to the fact you are presenting information from so many different data types, constantly jumping between the two makes the reading experience disjointed and impedes real integration of all the different data.

The manuscript has been entirely reorganized, separating methods and data (see section 3. Methods and Data), results (see section 4. Results), interpretation (see section 5. Interpretation), and discussion (see section 5. Discussion). Morever, the conclusion section has been largely re-written (see section 7. Conclusions) as requested by the reviewers.
* * *
Interpretation is often mixed with results. Perhaps it would help the reader to have an interpretation section at the end of the results. For the author this would avoid mixing observations and interpretation and would present an opportunity to more seamlessly integrate more than one data type in an interpretation. This would also make the manuscript more enjoyable and understandable for the reader.

Done. See interpretation (see section 5. Interpretation) and discussion (see section 6. Discussion) sections.
* * *
The discussion in broken up into two parts. The first part is the origin and activity of the

BMV. I enjoyed the section on the present activity of the BMV, which used backscattering and geochemical analysis to demonstrate that the mud volcano is currently quiescent but provides a conduit for dewatering deeper layers. However, my recommendations for the rest of this section are as follows: 1) the first part of the discussion should first demonstrate that the BMV is in fact a mud volcano; 2) At present the morphology section doesn't pose a question to be answered but rather states what was observed from the results. The geometry of a mud volcano is typically governed by two main factors, the viscosity of the extruding mud and the geometry of the connecting conduit. The scatter you see in the graph of height vs diameter is primarily related to this. Consider, what does the morphology (such as height and slope angle) tell us about the either the type of conduit or the fluidity of the extruded mud? Is this a mud cone or mud pie? What does the evidence of palaeo flows tell us about the episodicity of this mud volcano and what governs the changes in pressure that give this episodicity? These are simple questions that when answered will elevate the first part of the discussion.

Explained better in the text (section 6.1.2). However we cannot infer any dynamics of the mud volcano from the available data, because of the resolution of the seismic imaging and of the BMV location with respect to the seismic lines (BMV, projected-300 m). From the morphological analysis, the BMV could be interpreted as a pie-type mud volcano. However, looking at the slope values highlighted by the acquired multibeam data, the $< 5°$ angle criterion proposed by Kopf (2002) is not satisfied, being the BMV slope values $> 10°$ (Fig. 4). The paleoflows and dimension of the conduit are not visible in the seismic profiles, as, for example, the Christmas-tree structures described by Somoza et al., (2003) Mar. Geol. We can only infer from previous literature (e.g., Kioka and Ashi 2015, GRL) that dimensions and computed volume suggest a polygenethic behavior of the BMV so that we argue that the main fluid pathways have been possibly utilized several times to increase the volume.

Finally, in the drainage process section, what is the origin of the fluid and mud? Do

you have any constrains on the age/origin of the extruded mud from either seismic observations of depletion zones or core samples? Samples of hypersaline water are indicative of a fluid source at least as deep as the Messinian evaporites. However, it is currently unclear if the mud is also pre-salt in origin. This is important for our understanding of the process for how the mud has been mobilized, e.g. fluidization vs liquefaction.

I understand the points raised by the Reviewer, but the aim and rationale of our manuscript are different. We have no data to infer the age of the mud volcano and the mechanisms of mud mobilizations. Our aim and rationale are different and reported below. Aim: "Based on these new data integrated with previous ones, we interpret the recently-discovered mud volcano within the framework of the (seismically) active accretionary prism of Calabria and eventually provide some implications for future marine researches and monitoring of the seismic cycle in marine areas. As mentioned above, our main aim is not only to characterize the mud volcano but also to provide a contribution toward a potential and feasible future path for the use of these ubiquitous structures in favour of the mitigation of natural hazards."

Rationale: (1) a possible mud volcano (submarine) was signaled in the previous literature off Calabria (in a very seismic region) based solely on bathimetric evidence; (2) we surveyed this structure and collected geological, geophysical, and geochemical data; (3) thanks to these data, we confirmed and corroborated the hypothesis that the studied structure is truly a mud volcano, which sits atop seismically-active faults and has its inner conduit (through the volcano) somehow open with ascending fluids; (4) compared with nearby active fault zones, the conduit through the BMV seems to be a more efficient fluid carrier than the active fault zones (based on geochemical data); (5) outlook: what this structure might be used for? Given many recent studies/results on geochemical precursors of earthquakes, the BMV may be a future site for hydrogeochemical/seismic monitoring of potential seismic precursors.

I would recommend having a native English speaker read over the manuscript prior to any resubmission. Some sentences are unnecessarily long and would benefit from being broken down or made more concise. One key point per sentence with the subject of the sentence at the start.

We have improved the English as much as possible.
* * *
Annotations in figures are sometimes a little lacking, with text size often too small, certain annotated lines too thin or a colour that is hard to see and basic annotations including north arrows, colour bars and scale bars sometimes missing (examples given below).

We properly modified several figures following the reviewer's comment.
* * *
Specific comments Page 1, Line 8 – weak sentence with the use of probably. Whether the faults are seismically active or not is a fundamental part of the paper.

"Probably" omitted. ________________________________________________________________

Page 1, Line 15-17 – once again this is a weak sentence with too much use of words including may 'contribute, potential and feasible, in favour'. Please be clearer, assertive and concise in what you want to say. This is supposed to be one of your main conclusions.

Done. The sentence is now more assertive.
* * *
Page 2, Lines 12 – 15 – an example of a sentence that would benefit from being split in two. I suggest a full stop after the first set of citations and then a separate sentence above the amount of methane emitted from known mud volcanoes.

Done. ______________________________________________________________________

Page 2, Line 18-20 – This sentence needs citations, otherwise it is an unsupported statement.

Done. _______________________________________________________________________

Page 2, Line 24 – delete 'In other words,' at the start of the sentence. Start with 'The faults: : :.'.

Done. _______________________________________________________________________

Page 2, Line 22 – Weak start to sentence. Remove 'We believe,: : :.' And start with 'The activity of some of these: : :.'

Done. _______________________________________________________________________

Page 3, Line 9 – As you ascertained from the analysis of the BMVs geometry compared to other recoreded MVs, it is in fact a relatively small mud volcano. Therefore, please do not refer to it as a 'large mud volcano'.

Large is now omitted. _______________________________________________________________________

Page 5, Line 20 – You state that seismic events were selected within a distance of 12 km from the transect track. Please explain in the methods why you chose a distance of specifically 12 km.

We added that the chosen distance is arbitrary (P. 8 L. 29). A longer projection distance would be unsuitable. _______________________________________________________________________

Page 6, Line 16 – Please annotate/highlight examples of the small scale mass-wasting features listed here in Fig. 3.

Done. Please see the new version of Fig. 3 and its caption. _______________________________________________________________________

Page 6, Line 19 – Please highlight in Fig. 3 where 'this well-defined flat area of 26 km2'

is.

Done. Please see the new version of Fig. 3 and its caption.
* * *
Page 7, Line 3 – Can you please justify why you have selected this data set of 232 mud volcanoes?

We used the supplementary information by Kioka and Ashi 2015, GRL which is composed by 258 mud volcanoes. We found that 26 elements have a mean diameter set to NaN so that we consider 232 volcanoes with available mean diameter and height to compute volumes, by making use of the proposed formula (see explanation in Section 4.2.1).
* * *
Page 7, Line 4 – How have you defined the diameter and height of the BMV? This should be explained in the methods with reference to a figure showing these parameters. From the data currently presented it is not clear how you interpret the base of the MV in order to measure its height. An awareness of the imaging artefacts at the margin and beneath the MV should also be demonstrated. How do you interpret the conduit? These are often much narrower than the area of acoustic noise.

Explained in the text. BMV diameter and height were estimated by high resolution multibeam bathymetry data and values (vertical and horizontal scale in meters) are reported in Fig. 4. The height is measured on morphological basis from the foot of the BMV detected by multibeam data. No data below the BMV were used to interpret the internal feature of the mud volcano, because the penetration of the seismic signal on the BMV is poor or absent (see Scetion 4.2.1)
* * *
Section 3.4.2 – The seismic facies of the units is briefly described, but the acoustic character of the bounding reflections is not described. What is the polarity? What

is the acoustic impedance contrast? Is there variability in acoustic character along the horizon? This information gives further insight into the velocity and origin of the sediments.

Thank you for this suggestion but unfortunately we have little information on acoustic character of the horizons at present. Actually, we have not quantitative information on polarity and acoustic impedance. However, we evaluated amplitude and continuity of seismic reflections following the reviewer's comment. Please, see new version of Section 3.4.2.
* * *
Page 9, Line 8 – Please ensure all heading and sub-headings are capitalized where appropriate. Please check for other examples.

Done. _____________________________________________________________________________

Page 9, Line 16 – Is there a more technical term that can be used than cloudy facies. Is this term standard? I struggle to understand what you mean.

The patchy/cloudy facies is now explained at P. 13 L. 1-12.
* * *
Page 14, Line 5 – Please reference the statement about an association between overpressure and the MSC. See Bertoni et al., 2015 and Al-Balushi et al., 2016.

Done, P. 19 L. 28. _________________________________________________________________

Figure 1 – Please change the grey and grey dashed line colours to something that is brighter and stands out more. They are currently hard to see.

The grey and dashed gray lines have been replaced with ones with brighter colors in Figure 1, white and red respectively.
* * *
Figure 2 – Please add the north arrow.

Done. Figure 2 has been properly modified _______________________________________

Figure 3 – In Fig. 3b contours and contour numbers are at times almost impossible to see. Consider white for the text and thicker white lines for the contours. In Fig. 3a & 3b, more annotation and description in the caption of morphological features is needed. What are the detailed diagnostics from these maps for the BMV?

Done. We changed the colors of the map and the contour numbers. We added annotations of morphological features. Please see the new version of Fig. 3 and its caption.
* * *
Figure 4 – This figure would benefit from further subdivision into sections a-e. In Fig. 4a, the text size is too small, the units are missing, the scale bar is missing (also in 4b), the lines in the map are too thin and 'Rim' is not labelled on the bathymetric profile. 4b needs more annotation. What is meant by 'breaking'? Other than being nice to look at without further annotation or comment in the caption it is not clear what the 3D perspective adds. Please use a different line track labelling than a-a' and b-b' because a and b are used for subdividing the figure.

Done. We properly modified the figure after the reviewer's comments. Please see the new version of Fig. 4. We changed "breaking" with "breaching".
* * *
Figure 5 – This figure needs to be completely reworked with much more interpretation added to both the figure and caption. At present it tell the reader, especially one who doesn't work often with this data type, very little. The caption for 5a is too sparse and 5b reads in a way that suggests that there is massive fluid expulsion, although I think the author wants to state the opposite. Also, please be careful of using terms like 'massive fluid expulsion'. Unless you can quantify what you mean by 'massive' I suggest avoiding.

The     reviewer     is     right.         We     changed     the     caption     of     the     figure and we modified the figure following the reviewer's comment.

Figure 6 – Please add the north arrow.

Done. Figure 6 has been properly modified

Figure 7 – The text sizes vary throughout the figures. Please be consistent. Please add to Fig 7a a scale bar, north arrow and colour bar. The chirp profiles in Figs. 7b-d are quite zoomed out and so it is difficult to track any horizons. You have an un-interpreted version of this figure in the supplementary material, so please add line interpretations for horizons and also a more zoomed in view of the stratigraphy. What do the dashed lined represent?

Done. Please see the new version of Figure 7. A close-up view of 246 chirp profile has been also added.

Figure 8 – The region of extrusion is shrouded by artefacts, so how have you defined the boundary of the MV? This boundary looks more like the margin of a zone of artefacts and doesn't truly depict the margins of the MV.

We agree with the reviewer. We specified in Figures 7 and 8 that the boundaries represent the margin of the chaotic (not-penetrated) seismic facies.

Figure 9 – In the caption, should chirp profile be labelled left rather than right? This figure would also benefit from being broken into subsections, each of which is described in the caption.

The reviewer is right. We modified the caption of Figure 9 as suggested. Regarding the reviewer's suggestion to broke the Figure in two, we acknowledge that it could be useful but, at the same time, having the chirp close to core it could help the reader to understand immediately the message of

the figure. Anyway, we modified the figure in order to make it homogenous (e.g., we deleted the black rectangles around the chirp and the core).
* * *
Figure 10 – The figure is well presented. Fig, 10d is too zoomed in to interpret.

We modified the figure changing Figure 10d. _________________________________________

Figure 11 – Good Figure 12 – Nice figure. It would be good if it were integrated into the main body of text more as it is the final conceptual model.

The figure is properly discussed in the Discussion section.
* * *
Thanks a lot

Sincerely Andrea Billi and Co-Authors

Please also note the supplement to this comment:
https://www.solid-earth-discuss.net/se-2018-118/se-2018-118-AC2-supplement.pdf